# Exploration vs Exploitation: Rethinking RLVR through Clipping, Entropy, and Spurious Reward

**Peter Chen**[1] **Xiaopeng Li**[2] **Ziniu Li**[2] **Wotao Yin**[3] **Xi Chen**[4] **Tianyi Lin**[1]

[1]Columbia  [2]CUHK SZ  [3]DAMO, Alibaba US  [4]NYU Stern

## Abstract

This paper examines the exploration-exploitation trade-off in reinforcement learning with verifiable rewards (RLVR), a framework for improving the reasoning of Large Language Models (LLMs). Recent studies suggest that RLVR can elicit strong mathematical reasoning in LLMs through two seemingly paradoxical mechanisms: *spurious rewards*, which suppress exploitation by rewarding outcomes unrelated to the ground truth, and *entropy minimization*, which suppresses exploration by pushing the model toward more confident and deterministic outputs, highlighting a puzzling dynamic: both discouraging exploitation and discouraging exploration improve reasoning performance, yet the underlying principles that reconcile these effects remain poorly understood. We focus on two fundamental questions: (i) how policy entropy relates to performance, and (ii) whether spurious rewards yield gains, potentially through the interplay of clipping bias and model contamination. Our results show that clipping bias under spurious rewards reduces policy entropy, leading to more confident and deterministic outputs, while entropy minimization alone is insufficient for improvement. We further propose a reward-misalignment model explaining why spurious rewards can enhance performance beyond contaminated settings. Our findings clarify the mechanisms behind spurious-reward benefits and provide principles for more effective RLVR training.

## 1 Introduction

The recent emergence of Large AI Reasoning Models (e.g., Kimi-K2, OpenAI-o1, and DeepSeek-R1 (Kimi, 2025; Jaech et al., 2024; Guo et al., 2025)) has been driven by reinforcement learning with verifiable rewards (RLVR) (Li et al., 2025c). In RLVR, a verifier compares the model's rollout against a deterministic ground-truth solution, especially in mathematics and other STEM domains, providing outcome rewards. This verifiability has enabled models to achieve competitive and human-level performance on challenging benchmarks, such as the International Mathematical Olympiad (Huang & Yang, 2025). Among RLVR methods, Group Relative Policy Optimization (GRPO) (Shao et al., 2024) has become particularly popular due to its computational simplicity and memory efficiency.

In reinforcement learning, the exploration-exploitation trade-off is framed within a Markov decision process with per-step or shaped rewards. Exploration is typically promoted through stochastic policies or explicit bonus terms for underexplored actions (e.g., entropy regularization), while exploitation reinforces high-return actions via accurate value estimation. RLVR for LLMs departs from this paradigm in three respects: (i) rewards are outcome-level, extremely sparse, and verifiable only at the end of long rollouts, rendering all intermediate token-level actions reward-equivalent; (ii) exploration unfolds in sequence space and is governed by decoding temperature rather than state-local bonuses; and (iii) policy updates rely on ratio clipping with group-normalized advantages, making them more sensitive to importance ratios and relative ranks than to absolute reward values.

These properties give RLVR a distinctive exploration–exploitation regime. In classical RL, spurious rewards, which are misaligned with the true outcome reward (e.g., random noise), would be expected to hinder exploitation by injecting randomness that encourages suboptimal actions. Yet in RLVR, they have been observed to improve performance in `Qwen-Math` models (Shao et al., 2025), a

phenomenon attributed to upper-clipping bias that disproportionately amplifies high-prior responses, consistent with contamination effects reported on `MATH500` (Wu et al., 2025). Conversely, entropy minimization, which reduces policy entropy to yield more deterministic, high-confidence rollouts, has been widely adopted in RLVR and empirically linked to consistent gains (Zhang et al., 2025b; Zhao et al., 2025b; Cui et al., 2025; Fu et al., 2025). Notably, Agarwal et al. (2025) and Gao et al. (2025) directly optimize entropy as an objective and report substantial improvements even without verifiable feedback. These findings point to an RLVR-specific paradox: discouraging exploitation through spurious rewards and discouraging exploration through entropy minimization can both enhance validation accuracy, underscoring learning dynamics that depart from classical RL intuitions.

In this paper, we investigate how clipping, policy entropy, and spurious (random) rewards jointly shape model performance in RLVR. We show, both theoretically and empirically, that under random rewards, which discourage exploitation, clipping bias alone provides no meaningful learning signal and cannot directly improve performance. Instead, we establish a direct connection between clipping and policy entropy: clipping reduces entropy and drives the policy toward more deterministic, higher-confidence outputs, thereby inducing an entropy-minimization effect. Importantly, reduced entropy by itself does not guarantee performance gains. To clarify when spurious rewards can be beneficial, we introduce a simple reward-misalignment model. Our analysis overturns the prevailing view that improvements under spurious rewards are limited to potentially contaminated `Qwen-Math` models; similar gains also arise in the `Llama` and `QwQ` families, revealing a more nuanced exploration-exploitation dynamic that cannot be explained by contamination alone.

**Contributions.** We focus on two fundamental questions: (i) how policy entropy relates to performance, and (ii) whether spurious rewards yield gains, potentially through the interplay of clipping bias and model contamination. Our contributions can be summarized as follows:

1. We advance the theoretical foundations of RLVR by deriving explicit bounds on clipping bias and showing, under spurious rewards, this bias does not constitute a meaningful learning signal. To capture its effect more precisely, we introduce a novel one-step policy-entropy shift formulation, which establishes a deterministic link between clipping and policy entropy: clipping systematically reduces entropy and drives the policy toward more deterministic, higher-confidence rollouts.

2. We conduct extensive experiments across multiple model families (`Qwen-Math`, `Llama`, `QwQ`) and sizes (7B, 8B, 32B), including both base and distilled variants. These results reconcile conflicting reports in the literature, demonstrating that performance improvements under spurious rewards are robust and not tied to any single model or dataset.

3. We show that these gains cannot be attributed to clipping bias or to causal effects of policy entropy, thereby overturning the prevailing view that improvements under spurious rewards are confined to potentially contaminated `Qwen-Math` models. Instead, our findings reveal a broader and more nuanced exploration-exploitation dynamic unique to RLVR.

## 2 PRELIMINARIES AND TECHNICAL BACKGROUND

### 2.1 GROUP RELATIVE POLICY OPTIMIZATION

RLVR assigns a binary outcome-based reward $\mathbf{r}(\mathbf{x}, \mathbf{y})$ to a sampled response $\mathbf{y}$ from prompt $\mathbf{x}$ by comparing it against the ground-truth answer $\mathbf{y}^\star$. To learn an optimized policy via these reward, policy gradient methods (Williams, 1992; Sutton & Barto, 1998) aim to maximize

$$J(\theta) = \mathbb{E}_{\mathbf{x}\sim\rho, \mathbf{y}\sim\pi_\theta(\cdot|\mathbf{x})}[\mathbf{r}(\mathbf{x}, \mathbf{y})],$$

where $\rho$ is the prompt distribution and $\pi_\theta$ denotes the LLM policy. The parameter update at each iteration is $\theta \leftarrow \theta + \eta\nabla_\theta J(\theta)$. In practice, the trajectories are generated by an older policy $\pi_{\theta_{old}}$, but we wish to estimate the gradient at current policy $\pi_\theta$. By using the importance sampling technique with per-token ratio $r_t(\theta) = \frac{\pi_\theta(\mathbf{y}_t|\mathbf{h}_t)}{\pi_{old}(\mathbf{y}_t|\mathbf{h}_t)}$, it can be rewritten as

$$J(\theta) = \mathbb{E}_{\mathbf{x}\sim\rho, \mathbf{y}\sim\pi_{\theta_{old}}(\cdot|\mathbf{x})}\left[\sum_{t=1}^{|\mathbf{y}|} r_t(\theta)\mathbf{r}(\mathbf{h}_t, \mathbf{y}_t)\right],$$

where $\mathbf{y}_t$ is the $t$-th token of $\mathbf{y}$, which has $|\mathbf{y}|$ tokens in total, and $\mathbf{h}_t := \{\mathbf{x}, \mathbf{y}_{<t}\}$ with $\mathbf{h}_1 = \mathbf{x}$. Importance sampling might suffer from large variance when $\pi_\theta$ drifts from $\pi_{\theta_{\text{old}}}$. To stabilize training, we optimize the clipped surrogate objective as follows,

$$J(\theta) = \mathbb{E}_{\mathbf{x}\sim\rho, \mathbf{y}\sim\pi_{\theta_{\text{old}}}(\cdot|\mathbf{x})} \left[ \sum_{t=1}^{|\mathbf{y}|} \min\left\{ r_t(\theta)\mathbf{r}(\mathbf{h}_t, \mathbf{y}_t), \texttt{clip}[r_t(\theta), 1-\varepsilon, 1+\varepsilon)\mathbf{r}(\mathbf{h}_t, \mathbf{y}_t)] \right\} \right].$$

In this context, GRPO (Shao et al., 2024) and its variants (Yu et al., 2025; Liu et al., 2025b; Chu et al., 2025; Zhang et al., 2025a; Chen et al., 2025c; Li et al., 2025b) estimate policy gradients using groups of samples. For each prompt $\mathbf{x}$, GRPO draws a set $\{\mathbf{y}^{(i)}\}_{i=1}^G$ from $\pi_{\theta_{\text{old}}}$. We denote $\mathbf{y}_t^{(i)}$ as the $t$-th token of $i$-th sample $\mathbf{y}^{(i)}$ and $\mathbf{h}_t^{(i)} := \{\mathbf{x}, \mathbf{y}_{<t}^{(i)}\}$ and optimize the clipped objective as follows,

$$J(\theta) = \mathbb{E}_{\mathbf{x}\sim\rho, \{\mathbf{y}^{(i)}\}_{i=1}^G \sim \pi_{\theta_{\text{old}}}(\cdot|\mathbf{x})} \left[ \frac{1}{G}\sum_{i=1}^{G} \sum_{t=1}^{|\mathbf{y}^{(i)}|} \min\left\{ r_t^{(i)}(\theta)A_i, \texttt{clip}(r_t^{(i)}(\theta), 1-\varepsilon, 1+\varepsilon)A_i \right\} \right],$$

where $r_t^{(i)}(\theta) = \frac{\pi_\theta(\mathbf{y}_t^{(i)}|\mathbf{h}_t^{(i)})}{\pi_{\text{old}}(\mathbf{y}_t^{(i)}|\mathbf{h}_t^{(i)})}$, $\varepsilon \in (0, 1)$ is a hyper-parameter and the advantage $A_i := A(\mathbf{x}, \mathbf{y}^{(i)})$ is computed from the group rewards as follows,

$$A(\mathbf{x}, \mathbf{y}^{(i)}) = \frac{\mathbf{r}(\mathbf{x}, \mathbf{y}^{(i)}) - \texttt{mean}(\{\mathbf{r}(\mathbf{x}, \mathbf{y}^{(1)}), \ldots, \mathbf{r}(\mathbf{x}, \mathbf{y}^{(G)})\})}{\texttt{std}(\{\mathbf{r}(\mathbf{x}, \mathbf{y}^{(1)}), \ldots, \mathbf{r}(\mathbf{x}, \mathbf{y}^{(G)})\})}, \tag{1}$$

with $\mathbf{r}(\mathbf{x}, \mathbf{y}^{(i)}) = 1$ if $\mathbf{y}^{(i)}$ matches the ground-truth final answer and $\mathbf{r}(\mathbf{x}, \mathbf{y}^{(i)}) = 0$ otherwise.

**Remark 2.1.** *Under the GRPO update, the token-level advantage equals the response-level advantage $A_i$ and is independent of token index $t$.*

**Policy update.** Following Cui et al. (2025), we use the softmax policy update framework, and one typical iteration amounts to one-step exponentiation update with $G$ rollouts $\{\mathbf{y}^{(i)}\}_{i=1}^G$ as follows,

$$\pi_\theta(a \mid \mathbf{h}) = \frac{\pi_{\theta_{\text{old}}}(a|\mathbf{h})\exp(\eta\tilde{A}(\mathbf{h},a))}{\sum_{a'\in\mathcal{V}} \pi_{\theta_{\text{old}}}(a'|\mathbf{h})\exp(\eta\tilde{A}(\mathbf{h},a'))} \propto \pi_{\theta_{\text{old}}}(a \mid \mathbf{h})\exp(\eta\tilde{A}(\mathbf{h},a)), \tag{2}$$

where $\eta > 0$ is the step size and the advantage of an arbitrary token $a \in \mathcal{V}$ is given by

$$\tilde{A}(\mathbf{h}, a) = \frac{1}{G}\sum_{i=1}^{G}\sum_{t=1}^{|\mathbf{y}^{(i)}|} \left( \frac{\mathbf{1}\{\mathbf{h}_t^{(i)}=\mathbf{h}, \mathbf{y}_t^{(i)}=a\}}{\pi_{\text{old}}(a|\mathbf{h})} \right) A_i. \tag{3}$$

Throughout the paper, we assume there exists $\pi_{\min} > 0$ such that $\pi_{\text{old}}(a \mid \mathbf{h}) \geq \pi_{\min}$ for all $(\mathbf{h}, a)$. For the ease of presentation, we abbreviated $\pi_\theta$ and $\pi_{\theta_{\text{old}}}$ as $\pi_{\text{new}}$ and $\pi_{\text{old}}$ in the subsequent analysis. Building upon Eq. (2), we derive the following reparameterization for token-level importance ratio, with its proof presented in Appendix C.1.

**Lemma 2.2.** *Suppose that $\eta \geq 0$. Then, we have*

$$\log(\pi_{\text{new}}(a \mid \mathbf{h})) - \log(\pi_{\text{old}}(a \mid \mathbf{h})) - \eta(\tilde{A}(\mathbf{h}, a) - \mu(\mathbf{h})) + \frac{\eta^2}{2}\sigma^2(\mathbf{h}) \leq C\eta^3,$$

*where $\mu(\mathbf{h}) = \mathbb{E}_{a\sim\pi_{\text{old}}(\cdot|\mathbf{h})}[\tilde{A}(\mathbf{h}, a)]$, $\sigma^2(\mathbf{h}) = \text{Var}_{a\sim\pi_{\text{old}}(\cdot|\mathbf{h})}[\tilde{A}(\mathbf{h}, a)]$ and $C = \frac{1}{36\sqrt{3}(\pi_{\min})^3}$ does not depend on $\eta$. Equivalently, we have $\log(r(\mathbf{h}, a)) - \eta(\tilde{A}(\mathbf{h}, a) - \mu(\mathbf{h})) + \frac{\eta^2}{2}\sigma^2(\mathbf{h}) \leq C\eta^3$. As a consequence, under the standardized setting with $\mu(\mathbf{h}) = 0$ and $\sigma^2(\mathbf{h}) = 1$, we have*

$$\log(r(\mathbf{h}, a)) - \eta\tilde{A}(\mathbf{h}, a) + \frac{\eta^2}{2} \leq C\eta^3. \tag{4}$$

## 2.2 SPURIOUS REWARD FOR RLVR

Spurious reward arises whenever the feedback signal is misaligned with the ground truth reward. A random reward is a canonical example of such misalignment. In the context of RLVR, we formalize this notion as follows,

**Definition 2.3** (Random reward). We consider the binary reward $\mathbf{r}(\mathbf{x}, \mathbf{y}^{(i)})$ in Eq. (1). A *random reward* is a feedback signal independent of $(\mathbf{x}, \mathbf{y}^{(i)})$ and follows that $\mathbf{r}(\mathbf{x}, \mathbf{y}^{(i)}) \sim \text{Bernoulli}(\frac{1}{2})$, i.e., $\Pr(\mathbf{r}(\mathbf{x}, \mathbf{y}^{(i)}) = 1) = \Pr(\mathbf{r}(\mathbf{x}, \mathbf{y}^{(i)}) = 0) = \frac{1}{2}$.

Based on Definition 2.3, we obtain the following lemma for the GRPO advantage mechanism. These properties form the foundation for our subsequent analysis. The proofs are deferred to Appendix C.1.

**Lemma 2.4.** *Fixing a group size $G \geq 2$ and denoting $\mathbf{r}_i := \mathbf{r}(\mathbf{x}, \mathbf{y}^{(i)})$ and $A_i := A(\mathbf{x}, \mathbf{y}^{(i)})$ where $\{\mathbf{r}_i\}_{i=1}^G$ are a group of random rewards, we define $\bar{\mathbf{r}} = \frac{1}{G} \sum_{i=1}^G \mathbf{r}_i$, $\mathbf{S_r} = \sqrt{\frac{1}{G} \sum_{j=1}^G (\mathbf{r}_i - \bar{\mathbf{r}})^2}$, and $A_i = \frac{\mathbf{r}_i - \bar{\mathbf{r}}}{\mathbf{S_r}}$. Then, the following statements hold: (i) $A_i$ is symmetrically distributed around 0 and thus $\mathbb{E}[A_i^{2k-1}] = 0$ for all $k \in \mathbb{N}^+$; (ii) $|A_i| \leq \sqrt{G} - 1/\sqrt{G}$; (iii) $E[|A_i|] = \frac{2}{G2^G} \sum_{K=1}^{G-1} \binom{G}{K} \sqrt{K(G-K)}$; for all integers $k \geq 2$, $\mathbb{E}[|A_i|^k] \geq 1 - 2^{1-G}$.*

We examine several empirical findings related to random rewards. Notably, Shao et al. (2025) reports striking performance gains on `MATH500` for the `Qwen-Math` family when models are fine-tuned using the random reward defined in Definition 2.3. However, similarly large improvements are not observed for several other model families. Wu et al. (2025) likewise find substantial contamination in `Qwen-Math` on the `MATH500` validation benchmark, hypothesizing that the apparent gains under random reward largely stem from reinforcing memorized or contaminated trajectories. In particular, Shao et al. (2025) attributes these gains to the PPO-style upper-clipping bias, formalized as follows,

**Remark 2.5** (Upper-clipping bias). *The upper clipping enforces $r_t^{(i)}(\theta) = \frac{\pi_{\text{new}}(\mathbf{y}_t^{(i)}|\mathbf{h}_t^{(i)})}{\pi_{\text{old}}(\mathbf{y}_t^{(i)}|\mathbf{h}_t^{(i)})} \leq 1 + \varepsilon$ and implies that $\pi_{\text{new}}(\mathbf{y}_t^{(i)} \mid \mathbf{h}_t^{(i)}) \leq (1 + \varepsilon)\pi_{\text{old}}(\mathbf{y}_t^{(i)} \mid \mathbf{h}_t^{(i)})$. Equivalently, we have*

$$\Delta_{\max}(\mathbf{y}_t^{(i)}) = \pi_{\text{new}}(\mathbf{y}_t^{(i)} \mid \mathbf{h}_t^{(i)}) - \pi_{\text{old}}(\mathbf{y}_t^{(i)} \mid \mathbf{h}_t^{(i)}) \leq \varepsilon \pi_{\text{old}}(\mathbf{y}_t^{(i)} \mid \mathbf{h}_t^{(i)}).$$

*If $\pi_{\text{old}}(\mathbf{y}_t^{(i)} \mid \mathbf{h}_t^{(i)}) \geq \pi_{\text{old}}(\mathbf{y}_{t'}^{(i)} \mid \mathbf{h}_{t'}^{(i)})$ and the upper clipping are active for both tokens, we have $\Delta_{\max}(\mathbf{y}_t^{(i)}) \geq \Delta_{\max}(\mathbf{y}_{t'}^{(i)})$.*

The above interpretation indicates that upper clipping permits larger absolute increases for tokens that already have relatively high probability, whereas low-probability tokens reach the clipping threshold much earlier. This asymmetry can preferentially amplify high-prior responses, potentially exploiting latent knowledge rather than fostering new reasoning ability.

However, Oertell et al. (2025) challenge this interpretation, arguing that the reported gains arise from algorithmic heuristics and evaluation artifacts; in their experiments, random-reward fine-tuning does not consistently improve reasoning and can even degrade it. These conflicting findings highlight how little is currently understood about RLVR learning dynamics and motivate two central questions: (i) *Can random rewards improve model performance, and under what conditions*? (ii) *Does clipping bias provide a meaningful learning signal, and if not, what role does it actually play*? Following prior work, our empirical analysis also focuses primarily on `MATH500`. We further discuss the broader implications of random-reward training for general reinforcement learning settings in Appendix A.

## 2.3 LLM Policy Entropy

Policy entropy $\mathcal{H}(\pi_\theta)$ quantifies the diversity of a policy's action distribution. A high-entropy policy allocates probability more evenly across actions, producing a wider variety of sampled responses, whereas a low-entropy policy concentrates probability on a small subset of actions, resulting in more deterministic behavior (Li et al., 2025a).

**Definition 2.6** (Policy entropy). *For any given policy $\pi_\theta$, its entropy over a rollout trajectory space $\mathbf{y} \in \mathcal{Y}$ given prompt $\mathbf{x}$ can be defined as follows:*

$$\mathcal{H}(\pi_\theta) = -\mathbb{E}_{\mathbf{y} \sim \pi_\theta(\cdot|\mathbf{x})}[\log(\pi_\theta(\mathbf{y} \mid \mathbf{x}))] = -\sum_{\mathbf{y} \in \mathcal{Y}} \pi_\theta(\mathbf{y} \mid \mathbf{x}) \log(\pi_\theta(\mathbf{y} \mid \mathbf{x})).$$

Recent works in RLVR has begun to examine how policy entropy influences model performance. A common perspective emphasizes avoiding "entropy collapse" to prevent premature convergence to a low-diversity, suboptimal policy (Yu et al., 2025). At the token level, Wang et al. (2025b) similarly highlight the importance of minority high-entropy tokens for effective reasoning. Yet several studies report the opposite pattern: reducing entropy can be beneficial. Agarwal et al. (2025) explicitly optimize an entropy-minimization objective and observe performance improvements, and Cui et al. (2025) even propose a monotonic relationship in which lower entropy yields better

performance. These conflicting findings raise a second fundamental question: (iii) *Is there a direct causal relationship between policy entropy and policy performance?*

Beyond empirical observations, Cui et al. (2025) provide a theoretical analysis by deriving the following estimate of the one-step change in policy entropy:

$$\mathcal{H}(\pi_{\text{new}}) - \mathcal{H}(\pi_{\text{old}}) \approx -\text{Cov}_{\mathbf{y} \sim \pi_\theta(\cdot|\mathbf{x})}(\log(\pi_{\text{old}}(\mathbf{y} \mid \mathbf{x})), A(\mathbf{x}, \mathbf{y})). \tag{5}$$

Intuitively, if the reward is positively correlated with the rollout probability, meaning high-probability responses tend to receive reward 1 while low-probability responses receive reward 0, the policy becomes more peaked, leading to a decrease in entropy. Conversely, if low-probability responses receive reward 1 and high-probability responses receive reward 0, the policy is pushed toward a flatter distribution, increasing its entropy. However, we emphasize that the approximation in Eq. (5) does not apply for analyzing RLVR with random rewards.

**Remark 2.7.** *Under random rewards, because $A(\mathbf{x}, \mathbf{y})$ is independent of $\pi_{\text{old}}(\mathbf{y} \mid \mathbf{x})$ and has zero mean, substituting into Eq. (5) yields $\mathcal{H}(\pi_{\text{new}}) - \mathcal{H}(\pi_{\text{old}}) = 0$ (see Appendix C.3 for details). This implies that policy entropy should remain constant throughout training. However, this prediction contradicts our empirical observations, which exhibit a clear interaction between clipping and entropy dynamics. The discrepancy arises because Eq. (5) (i) retains only first-order terms in the policy expansion, ignoring higher-order contributions, and most importantly and (ii) assumes an unclipped formulation. Our theoretical results in §4.1 provide a more complete picture of how clipping interacts with and modulates policy entropy.*

## 3 CLIPPING AND MODEL PERFORMANCE

We provide a rigorous analysis of the upper-clipping bias from Remark 2.5. Indeed, we derive explicit bounds on the magnitude of the clipping bias and describe its effect on the learning signal. We further validate our theoretical findings with extensive empirical experiments.

### 3.1 THEORETICAL RESULTS

We begin by decomposing the upper-clipping surrogate into two components: the raw term $N_t$, corresponding to the unclipped surrogate, and the clipping-correction term $N_t^{\text{clip}}$.

**Definition 3.1.** Suppose a rollout $\mathbf{y}$ of length $L$ is sampled from a prompt $\mathbf{x}$ and the clip ratio is $\varepsilon \in (0, 1)$. For simplicity, we denote the token-level ratio $r(\mathbf{h}_t, \mathbf{y}_t)$ as $r_t$. Then, we define the clipped token-level ratio as $\bar{r}_t = \text{clip}(r_t, 1 - \varepsilon, 1 + \varepsilon) = \max\{\min\{r_t, 1 + \varepsilon\}, 1 - \varepsilon\}$, the raw surrogate as $N_t = r_t A(\mathbf{x}, \mathbf{y})$, the clipping-correction surrogate as $N_t^{\text{clip}} = \bar{r}_t A(\mathbf{x}, \mathbf{y})$, and the upper activation indicator $I_t^+ := \mathbf{1}_{\{r_t > 1+\varepsilon\}}$. The corresponding total *upper clipping correction* $C_{\text{tot}}^+$ is defined as

$$C_{\text{tot}}^+ = \sum_{t=1}^{L} (N_t^{\text{clip}} - N_t) I_t^+ = \sum_{t=1}^{L} (\bar{r}_t - r_t) I_t^+ A(\mathbf{x}, \mathbf{y}).$$

For simplicity, we omit the superscript $i$ since it can be applied to any sample of the response group. The following theorem provides an upper bound on $\mathbb{E}[|C_{\text{tot}}^+|]$; its proof is deferred to Appendix C.2.

**Theorem 3.2.** *Let a prompt $\mathbf{x}$ have a response group of size $G$, each rollout has length $L$, and the clip ratio is $\varepsilon \in (0, 1)$. For any rollout $\mathbf{y}$, write $A := A(\mathbf{x}, \mathbf{y})$. Denote $p_+ := \mathbb{E}[I_t^+]$ and $D_t^+ := (\bar{r}_t - r_t) I_t^+$ such that $C_{\text{tot}}^+ = \sum_{t=1}^{L} D_t^+ A$. Then, for all $\eta > 0$, we have*

$$\mathbb{E}[|C_{\text{tot}}^+|] \leq M\sqrt{2p_+ L R_\eta^{\max} \phi(R_\eta^{\max})} + ML\Delta_\eta^+ \min\left\{\sqrt{p_+}, \frac{\phi(R_\eta^{\max})}{\phi(1+\varepsilon)}\right\}, \tag{6}$$

*where $R_\eta^{\max} = e^{\eta/\pi_{\min}}$, $M = \sqrt{G-1}$, $\phi(u) = u \log u - u + 1$, and $\Delta_\eta^+ = (R_\eta^{\max} - 1 - \varepsilon)_+$. For sufficiently small $\eta$, we have $\mathbb{E}[|C_{\text{tot}}^+|] \leq c_1 \eta \sqrt{L} + \min\{c_2 \eta \sqrt{p}L, c_3 \eta^3 L\}$ where $c_1 = M\sqrt{2e}\pi_{\min}^{-1}$, $c_2 = M(e-1)\pi_{\min}^{-1}$, and $c_3 = M(e-1)\phi(1+\varepsilon)^{-1}\pi_{\min}^{-3}$.*

**Remark 3.3.** *Theorem 3.2 shows that the upper bound on the total clipping-correction term depends on the (empirical) expected token-level activation rate $p$: larger $p$ would bring more clipping correction. $p$ varies across model families but can be directly monitored during training. This motivates a general, model-agnostic framework for analyzing clipping effects – one that applies uniformly across architectures by expressing all bounds in terms of the observable activation rate $p$.*

To quantify the effect of clipping, we establish the following bound relating the magnitude of the raw surrogate sum to the total clipping correction. For the proof, please refer to Appendix C.2.

**Theorem 3.4** (Law of Clipping). *Under the same settings as Theorem 3.2, we define the raw surrogate sum $N_{\text{raw}} = \sum_{t=1}^{L} r_t A$. Then, for all $\eta > 0$, we have*

$$\mathbb{E}[|N_{\text{raw}}|] \geq L\mathbb{E}[|A|]e^{-C\eta^2} \geq L\mathbb{E}[|A|](1 - C\eta^2), \tag{7}$$

*where $C = \frac{1}{8\pi_{\min}^2}$. Furthermore, we have*

$$\frac{\mathbb{E}[|N_{\text{raw}}|]}{\mathbb{E}[|C_{\text{tot}}^+|]} \geq \frac{\mathbb{E}[|A|](1 - C\eta^2)}{L^{-1/2}M\sqrt{2p_+ R_\eta^{\max}\phi(R_\eta^{\max})} + M\Delta_\eta^+ \min\left\{\sqrt{p_+}, \frac{\phi(R_\eta^{\max})}{\phi(1+\varepsilon)}\right\}}.$$

*In addition, under practical hyperparameter settings, we have $\mathbb{E}[|N_{\text{raw}}|] \gg \mathbb{E}[|C_{\text{tot}}^+|]$. A quantitative evaluation using the parameters from our actual training setup is given in Corollary 3.6.*

## 3.2 MODEL-SPECIFIC EVALUATION

Following the hyperparameter configuration of Shao et al. (2025), we train `Qwen2.5-Math-7B` on the DeepScaleR dataset (Luo et al., 2025) using random rewards drawn from Bernoulli($\frac{1}{2}$). The training setup uses a batch size of 128, group size of 16, decoding temperature 1.0, clipping ratio 0.2, learning rate $5 \times 10^{-7}$, and KL coefficient 0.

We run multiple consecutive experiments with and without clipping using the `verl` framework (Sheng et al., 2025). The resulting training trajectories on the `MATH500` validation set, together with the clipping activation fraction over training, are shown in Figure 1. We adopt the default training prompt from `verl`, which instructs the model to enclose its final answer in a box for verifier validation (see Appendix A for further discussion). Notably, for `Qwen2.5-Math-7B`, the clipping activation rate is substantially lower than what is typically observed in other base models:

**Remark 3.5.** *Empirically, the clipping activation ratio is usually below $1\%$ for general GRPO training. For specific `Qwen2.5-Math-7B` training, the clipping activation ratio never exceeds $0.2\%$, with expected activation probability $\mathbb{E}[I_t] \approx 0.001$.*

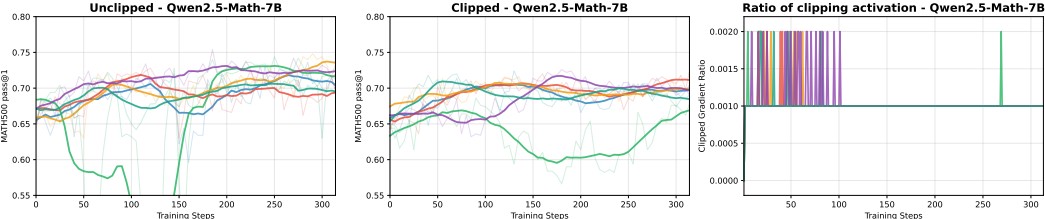

Figure 1: Independent trials over `Qwen2.5-Math-7B` on the `MATH500` validation set. For performance validation subpanels (Left & Middle), each color represents a different run; the bold line shows the smoothed trajectory, and the faint line of the same color shows the corresponding raw individual run. All later figures follow the same plotting convention. Unclipped training (**Left**); clipped training (**Middle**); and clipping activation ratio during training (**Right**).

As shown in Figure 1, enabling clipping can lead to a decline in validation performance, whereas disabling clipping often results in improvement. These findings suggest that upper clipping bias is not the mechanism driving the observed gains under random rewards. To illustrate this point, we provide a numerical instantiation of Theorem 3.4 using the training hyperparameters of `Qwen-Math`:

**Corollary 3.6.** *Suppose that $\eta = 5 \times 10^{-7}$, $\varepsilon = 0.2$, $p_+ = 0.001$, $G = 16$, $L = 4096$, and $\pi_{\min} = 10^{-6}$, then by Theorem 2.4, $M = 3.75$ and $\mathbb{E}[|A|] \approx 0.967$; by Theorem 3.2, $R_\eta^{\max} \approx 1.649$, $\phi(R_\eta^{\max}) \approx 0.176$, and $\Delta_\eta^+ \approx 0.449$. Thus, Theorem 3.4 implies $C = 1.25 \times 10^{11}$ and*

$$\frac{\mathbb{E}[|N_{\text{raw}}|]}{\mathbb{E}[|C_{\text{tot}}^+|]} \geq \frac{\mathbb{E}[|A|](1 - C\eta^2)}{L^{-1/2}M\sqrt{2p_+ R_\eta^{\max}\phi(R_\eta^{\max})} + M\Delta_\eta^+ \min\left\{\sqrt{p_+}, \frac{\phi(R_\eta^{\max})}{\phi(1+\varepsilon)}\right\}} \approx 17.15.$$

*This confirms that $\mathbb{E}[|N_{\text{raw}}|] \gg \mathbb{E}[|C_{\text{tot}}^+|]$ in magnitude for hyperparameters used in practice.*

**Remark 3.7.** *As a consequence of Corollary 3.6, the upper-clipping bias fails to provide meaningful learning signal towards the gradient, even under contaminated model and benchmark. This result is supported by our empirical observations and theoretical justifications. We present further ablation analysis over clipping threshold and group size in Appendix B. Nonetheless, even though clipping does not directly correlated to performance, §4 shows that it still has a causal effect on policy entropy under random rewards, shaping the structure of the outcomes without enhancing learning.*

## 4 CLIPPING AND POLICY ENTROPY

We provide two theoretical results describing policy entropy under *unclipped* and *clipped* training (§4.1). As discussed in §2.3, the approximation in Eq. (5) from Cui et al. (2025) becomes inaccurate when clipping or random rewards are present. Our analysis incorporates both clipping and initial policy skewness, yielding a more precise characterization of entropy dynamics. In §4.2, we validate these results through extensive experiments and targeted case studies. In §4.3, we interpret clipping as a mechanism that implicitly reduces entropy and caution – supported by empirical evidence – against conflating entropy reduction with improved performance.

### 4.1 ONE-STEP POLICY ENTROPY CHANGE UNDER RANDOM REWARDS

We analyze entropy dynamics under unclipped and clipped training in Theorem 4.1 and Theorem 4.3, and hope these results motivate new ways to modulate entropy using spurious-reward setups alongside explicit entropy regularization.

We first present the unclipped-training dynamics in Theorem 4.1, with detailed statement and proof in Appendix C.3, where we also identify the conditions that permit entropy growth.

**Theorem 4.1.** *With update in Eq. (2) and $c_G := (1 - 2^{1-G})/G$, for all $\eta > 0$,*

$$\mathbb{E}[\mathcal{H}(\pi_{\text{new}}) - \mathcal{H}(\pi_{\text{old}})] = -c_G \Phi(\pi_{\text{old}})\eta^2 + \mathbb{E}[R(\eta)],$$

*where $\Phi$ measures the skewness of $\pi_{\text{old}}$ and $|R(\eta)| = \mathcal{O}(\eta^4)$ for small $\eta$.*

**Remark 4.2.** *Theorem 4.1 shows that the one-step entropy change under unclipped training depends critically on the initial policy distribution; indeed, more skewed policies can exhibit entropy increases during training. As a concrete example, we consider a two-armed policy $\pi_{\text{old}} = (\beta, 1 - \beta)$ for some $\beta \in (0, 1)$. In this case, using the definition of $\Phi$, one can compute $\Phi(\pi_{\text{old}}) = 1 + (1 - 2\beta)\log(\frac{\beta}{1-\beta})$. Moreover, $\Phi(\pi_{\text{old}}) \geq 0$ if and only if $\beta \in [0.176, 0.824]$. Thus, up to $\mathcal{O}(\eta^2)$ term, entropy decreases in expectation when $\beta \in [0.176, 0.824]$ (a less skewed policy) and increases when $\beta > 0.824$ or $\beta < 0.176$ (a more skewed policy). Figure 9 illustrates this behavior: for a less-skewed initialization (Figure 9, Left), spurious rewards do not increase entropy under unclipped training, whereas with a sufficiently skewed initialization (Figure 9, Right), entropy increases over training. This is also consistent with the entropy growth observed in our experiments (Figure 2, Left).*

**Actual policy $\Phi(\pi)$ evaluation.** Apart from the two-armed example in Remark 4.2, we further evaluate $\Phi(\pi)$ for the actual `Qwen-Math-7B` policy in Figure 8, which helps readers better perceive the policy skewness and its corresponding $\Phi(\pi)$ associated with entropy increases during training. For detailed setup and results, please refer to Appendix B.

Next, we analyze training dynamics with upper clipping in Theorem 4.3; detailed statements, proof and entropy decay verification under practical parameters are deferred to Appendix C.3.

**Theorem 4.3.** *Define $C_i := \{A_i > 0, r(\mathbf{y}^{(i)}) > 1 + \epsilon\}$. Let $\rho := \mathbb{P}(C_1)$ and $\delta := \mathbb{E}[r(\mathbf{y}^{(1)}) - (1 + \epsilon) \mid C_1]$. Then for $\eta > 0$ small enough and any $p \in (\pi_{\min}, 1)$,*

$$\mathbb{E}[\mathcal{H}(\pi_{\text{new}}) - \mathcal{H}(\pi_{\text{old}})] \leq -c_G \Phi(\pi_{\text{old}})\eta^2 + \mathbb{E}[R(\eta)] + c(p)G\left(\rho\delta_{\text{eff}} - \frac{X_{\max}}{2}(G - 1)p\right),$$

*where $c_G$, $\Phi$ and $R(\eta)$ are defined as the same as in Theorem 4.1; for other constants, see Appendix C.3.*

**Remark 4.4.** *As shown in Figure 2 (Middle), our experiments confirm that policy entropy consistently decreases over time under random rewards. In contrast, disabling clipping leads to entropy increasing during training (Figure 2, Left). Existing approaches to counter early-stage entropy collapse rely on*

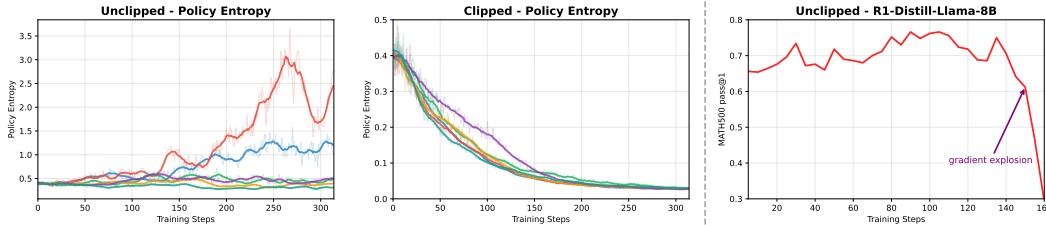

Figure 2: Policy entropy evolution of `Qwen2.5-Math-7B` under random-reward training, with results for unclipped training (**Left**) and clipped training (**Middle**); Unclipped training with `R1-Distill-Llama-8B`, an example that leads to the gradient explosion (**Right**).

*regularization techniques that merely slow the decay (Wang et al., 2025a; Yao et al., 2025; Zheng et al., 2025; Cheng et al., 2025). Our finding that one can* actively increase *policy entropy – while also improving validation performance – suggests a complementary strategy: using spurious-reward setups to more effectively preserve and modulate entropy. This highlights a promising direction for combining true and spurious rewards to better balance exploration and exploitation in RLVR.*

### 4.2 EMPIRICAL EVALUATION

Figure 2 (Left & Middle) shows that, under random rewards, disabling clipping can cause policy entropy to *increase* over training, reflecting progressively greater exploration. In contrast, enabling clipping constrains this behavior and leads to a monotonic decrease in entropy. This pattern highlights that clipping functions primarily as a form of *regularization*: by capping per-token likelihood ratios, it effectively reduces the update step size and prevents the policy from drifting too far from its previous distribution. Beyond its regularization effect, clipping also fulfills its original purpose of preventing gradient explosion, thereby adding further training stability.

When gradient magnitudes grow large, clipping protects the optimization process by preventing abrupt, destabilizing updates. Without clipping, this safeguard disappears: the optimizer may take oversized steps that inject excessive exploration and destabilize training. Thus, clipping does not introduce additional learning signals; its primary function is to maintain optimization stability by enforcing a local trust region. Models with sufficiently large single-step gradient norms can collapse entirely. A failure case is shown in Figure 2 (Right): training `R1-Distill-Llama-8B` without clipping initially raises the `MATH500` validation accuracy from $65.6\%$ to $76.6\%$ within 100 steps, but around step 150 the gradients explode, causing a sharp drop in performance. For comparison, the clipped-training counterpart for `R1-Distill-Llama-8B` is shown in Figure 4 (Middle).

### 4.3 POLICY ENTROPY AND MODEL PERFORMANCE

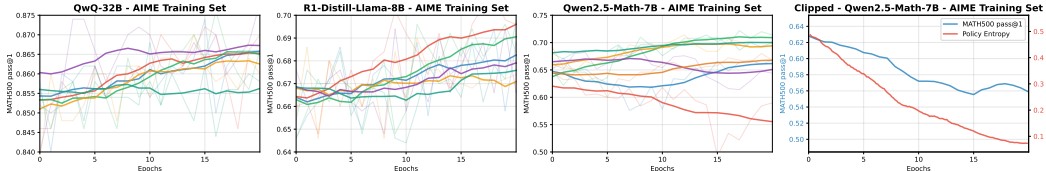

Figure 3: Results on AIME training set on `QwQ-32B` (**Left**), `R1-Distill-Llama-8B` (**Middle-L**), `Qwen2.5-Math-7B` (**Middle-R**). With one specific example that shows entropy minimization would lead to sub-optimal policy under noisier and more difficult training environment (**Right**).

Figure 2 shows that both higher and lower entropy can achieve improved performance. In practice, higher entropy reflects stronger *exploration*: the policy is flatter and thus more capable of discovering new trajectories. Lower entropy corresponds to greater confidence, with the policy becoming more concentrated on a small set of trajectories; in RLVR, such concentration may also correlate with better performance. However, this connection is not guaranteed: convergence to a highly skewed, low-entropy policy does *not* necessarily improve accuracy, as demonstrated in Figure 3 (Right).

This suggests that methods explicitly minimizing policy entropy should be applied with caution. Additional evidence from unclipped training under the same setup is provided in Appendix B.

Under random rewards, clipping acts as an implicit entropy minimization mechanism, pushing the policy toward a more peaked distribution that concentrates probability mass on a small set of trajectories. Whether this effect is beneficial depends on the model's initial policy distribution and the difficulty of the training data. For a strong model on an easy dataset, the policy is already concentrated on correct trajectories; additional concentration can be sufficient and may appear advantageous. We provide a simple theoretical explanation of this phenomenon in §5.

However, as the training data becomes difficult, the policy may place most of its probability mass on *incorrect* trajectories. This produces the noisy rollouts and unstable updates, often driving the model toward an incorrect low-entropy solution. To illustrate, for Qwen2.5-Math-7B, we replace the milder DeepScaleR curriculum with the harder AIME Past series. As shown in Figure 3 (Middle-R), after 20 training epochs (with the same hyperparameters as in Figure 2), the trajectory resembles a random walk with little meaningful improvement in validation accuracy. In contrast, stronger QwQ-32B and R1-Distill-Llama-8B models (rollout length 8192, with all other settings identical to the 7B configuration) trained on AIME dataset exhibits steady early-epoch gains (Figure 3, Left & Middle-L). These results indicate that the effectiveness of entropy minimization is regime-dependent: for strong models on easier data, it can further concentrate mass on correct trajectories, whereas for weaker models or harder data, it may reinforce incorrect modes and stall, or degrade performance. Thus, entropy minimization mechanisms (including clipping under random rewards) can be interpreted as regularization rather than universally beneficial learning signals.

## 5 REWARD MISALIGNMENT: WHO CAN BENEFIT FROM RANDOM REWARDS?

From empirical observations in this and prior work, we note two consistent patterns under random-reward training. First, in line with Shao et al. (2025), weaker models tend to improve less—and importantly, *model strength is dataset-dependent*: a model that performs well on an easier benchmark may struggle on a harder one. Second, as baseline accuracy increases (e.g., approaching 70%), training dynamics become noticeably smoother, whereas models starting around 50% accuracy exhibit substantially more oscillation. To explain when and why a model may improve under random rewards, we analyze the phenomenon through the lens of *reward misalignment*. As a warm-up, we introduce a simple probabilistic model that captures this mechanism in the binary outcome-reward (ORM) setting, converting the observed behavior into a tractable misalignment analysis.

For a prompt $\mathbf{x}$, draw $G$ rollouts $\{\mathbf{y}^{(1)}, \ldots, \mathbf{y}^{(G)}\}$ from current policy $\pi_\theta$. Partition the indices into correct and incorrect sets $\mathcal{C}, \mathcal{I} \subseteq \{1, \ldots, G\}$ with $|\mathcal{C}| = n_c$, $|\mathcal{I}| = n_i$, and $n_c + n_i = G$. We analyze two label errors: (i) *False positives (FP)*: $\mathbf{r}_j = 1$ for $j \in \mathcal{I}$ (an incorrect rollout is rewarded); (ii) *False negatives (FN)*: $\mathbf{r}_k = 0$ for $k \in \mathcal{C}$ (a correct rollout is not rewarded). Specifically, we aim to explain: (i) why validation curves fluctuate less when accuracy is high but become noticeably unstable when accuracy is low, and (ii) why stronger models are more likely to improve under random rewards. Our starting point is to formalize *reward misalignment*: the loss of advantage mass that should have been assigned to correct rollouts but is instead diverted due to random reward mislabeling.

**Definition 5.1** (Correct-response advantage loss). Let $\{\mathbf{r}_j\}_{j=1}^G$ be i.i.d. with $\mathbf{r}_j \sim \text{Bernoulli}(\frac{1}{2})$ for all $j$, independent of correctness. We define the event counts $f := \sum_{j \in \mathcal{I}} \mathbf{1}\{\mathbf{r}_j = 1\}$ and $g := \sum_{k \in \mathcal{C}} \mathbf{1}\{\mathbf{r}_k = 0\}$, and let $T := \sum_{j=1}^G \mathbf{r}_j = f + (n_c - g)$ be the total number of $+1$ rewards. We write $\bar{\mathbf{r}} := \frac{T}{G}$ for the group-averaged reward. The class-wise centered reward sum over $\mathcal{C}$ is $\Sigma_{\mathcal{C}}(f, g) := \sum_{k \in \mathcal{C}} (\mathbf{r}_k - \bar{\mathbf{r}}) = (n_c - g) - \frac{n_c T}{G}$. As an "ideal" reference with no mislabels ($f = g = 0$), we have $\Sigma_{\mathcal{C}}^{\text{ideal}} = \sum_{k \in \mathcal{C}} (1 - \frac{n_c}{G}) = n_c(1 - \frac{n_c}{G})$. Finally, we define the *damage* (advantage loss) as

$$\Delta(f, g) := \Sigma_{\mathcal{C}}^{\text{ideal}} - \Sigma_{\mathcal{C}}(f, g). \tag{8}$$

**Proposition 5.2.** *For any $n_c, n_i \geq 1$ and $G = n_c + n_i$, let $f \sim \text{Binomial}(n_i, \frac{1}{2})$, $g \sim \text{Binomial}(n_c, \frac{1}{2})$ be independent, and $\Delta := \Delta(f, g)$ be defined in Eq. (8). Under i.i.d. $\text{Bernoulli}(\frac{1}{2})$ rewards, we have*

$$\mathbb{E}[\Delta] = \frac{n_c(G - n_c)}{G}, \quad \text{Var}(\Delta) = \frac{n_c(G - n_c)}{4G}. \tag{9}$$

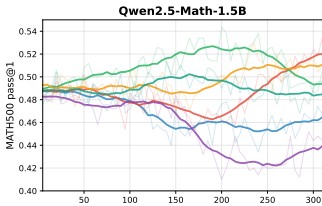 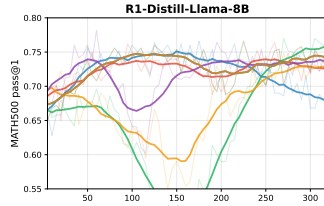 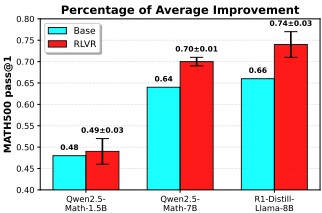

Figure 4: Results of `Qwen2.5-Math-1.5B` under clipped training (**Left**); results of R1-Distill-Llama-8B under clipped training (**Middle**); percentage improvement (averaged over six independent runs) for different models under the same training and validation setup (**Right**).

The expected damage decreases as the number of correct rollouts $n_c$ increases, and its variance likewise shrinks with $n_c$, explaining why the stronger models exhibit more stable validation curves. The largest fluctuations occur near the symmetric regime $n_c \approx n_i$. This is consistent with our empirical results in Figure 1. We further refine this characterization by decomposing the damage into conditional means in Theorem 5.3. The proofs are given in Appendix C.4.

**Theorem 5.3.** *Let $f \sim \text{Binomial}(n_i, \frac{1}{2})$ and $g \sim \text{Binomial}(n_c, \frac{1}{2})$ be independent, and let $\Delta$ be defined in Eq. (8). For policy with more correct rollouts ($n_c > n_i$), we have*

$$\mathbb{E}[\Delta \mathbf{1}_{\{f>g\}}] \leq \mathbb{E}[\Delta \mathbf{1}_{\{g>f\}}].$$

*As $n_c$ increases on $\left[\frac{G}{2}, G\right]$, we have $\mathbb{E}[\Delta \mathbf{1}_{\{f>g\}}]$ constitutes a strictly smaller fraction of $\mathbb{E}[\Delta]$.*

Theorem 5.3 refines Theorem 5.2. As the overall damage $\mathbb{E}[\Delta]$ decreases with $n_c$, the *composition* of that (shrinking) damage shifts: for stronger models (those with $n_c > n_i$), FN-dominated regions ($g > f$) contribute a larger share than FP-dominated regions ($f > g$), and the FP-dominated portion decreases monotonically. Practically, this means that training stronger models on datasets where $n_c > n_i$ incurs less total misalignment damage—particularly fewer FP misallocations—and is therefore more likely to yield improvements under random rewards. This effect persists even beyond contaminated-reward settings. We further corroborate this trend through experiments on a stronger distilled `Llama` model and a weaker `Qwen-Math` model, with results shown in Figure 4.

As reported by Shao et al. (2025), base `Llama` models reliably degrade during random-reward training across trials. Under the reward-misalignment perspective, stronger models should benefit more and are thus more likely to improve. We test this by evaluating a stronger distilled `Llama` variant, whose base and teacher models both exhibit contamination on `MATH500`. As shown in Figure 4 (Middle), using a rollout length of 8192 tokens and matching all other hyperparameters to the `Qwen-Math` configuration, we observe improvements comparable to those in Figure 2. In contrast, the weaker and potentially contaminated `Qwen-Math` model (Figure 4, Left) fails to achieve similar gains. These results indicate that validation-set contamination does not account for the improvements under random rewards, nor is the effect specific to `Qwen-Math`. Figure 4 (Right) summarizes the percentage improvements across the model results in Figure 1 (Left) and Figure 4 (Left and Middle).

## 6 CONCLUSION

We now revisit the three guiding questions posed in §2: (i) Can random rewards improve model performance, and under what conditions? (ii) Does clipping bias provide a meaningful learning signal, and if not, what purpose does it serve? (iii) Is there a direct causal relationship between policy entropy and policy performance?

First of all, random rewards **can** improve model performance. As shown in §5, the benefits depend on model strength: stronger models are more likely to realize gains from random reward, whereas weaker models become unstable when trained on harder datasets. Second, clipping bias does **not** supply a useful signal (§3.1); instead, its function is to regulate policy entropy in the presence of spurious training signals (§4.1). Finally, as demonstrated in §4.2 and §4.3, policy entropy and performance do not exhibit a deterministic causal relationship: entropy decreases may accompany performance collapse, while entropy increases may coincide with improvements. Overall, our theoretical and empirical analyses disentangle the complex interplay between exploration and exploitation in RLVR, offering a principled foundation for future work to understand the alignment dynamics.

ACKNOWLEDGMENT

We sincerely appreciate Buzz High Performance Computing (https://www.buzzhpc.ai, info@buzzhpc.ai) for providing computational resources and support for this work.

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

# A RELATED WORKS

We provide a technical review clarifying the differences in experimental setups and summarizing insights from recent advances in RLVR for LLM post-training.

**Spurious reward in classical RL.** We provide broader context on how prior work in reinforcement learning for classical settings has leveraged spurious rewards to facilitate training. Spurious reward signals are closely linked to challenges in generalization (Zhang et al., 2018; Koch et al., 2021; Langosco Di Langosco et al., 2022). While these works illustrate deliberate uses of such signals, spurious rewards may also arise unintentionally, leading to reward misspecification and reward hacking (Pan et al., 2022); similar reward hacking has been documented in Tien et al. (2023). A second relevant thread traces back to *potential-based reward shaping* (PBRS) (Ng et al., 1999), which introduces additional or misaligned rewards in principled ways to preserve the optimality of desired behaviors. More recently, *Random Network Distillation* (RND) (Burda et al., 2019) emerged as a leading exploration mechanism, with subsequent extensions such as Ma et al. (2025). In this same context, numerous other works propose reward signals (often spurious with respect to the true task objective) that encourage an agent to explore the state space in ways that eventually uncover genuine rewards (Pathak et al., 2017; Zhang et al., 2021; Wang et al., 2023; Li et al., 2024a). Spurious rewards have thus played a substantial role in improving exploration. One prominent theoretical foundation is *posterior sampling* (Russo & Van Roy, 2014), which motivates exploration through uncertainty and has been generalized to broader RL settings (Chen et al., 2024b; Xu et al., 2025).

**Spurious reward in RLVR.** We now turn to recent works that study spurious rewards in RLVR. Although these works report broadly similar empirical phenomena, their experimental configurations differ in important ways. In Shao et al. (2025), the prompt omits the standard `Qwen`-style instruction to place the final answer in a box. As they note, `Qwen-Math` is highly sensitive to prompt formatting, and such differences can substantially shift baseline performance. In contrast, our experiments follow the default `Qwen` prompt used in `verl` (Sheng et al., 2025), which explicitly instructs the model to place the final answer in a boxed expression—mirroring the RLVR verifier in `verl`, which extracts the boxed answer for scoring and reward assignment.

Apart from this prompt choice, we match all other hyperparameters in Shao et al. (2025). Oertell et al. (2025), however, adopt a markedly different configuration: (i) a rollout-length cap of 1024 tokens (well below the 4096-token context window of `Qwen-Math`), (ii) a different training dataset (`MATH` (Hendrycks et al., 2021) instead of DeepScaleR (Luo et al., 2025)), (iii) a significantly smaller learning rate ($1 \times 10^{-7}$ versus $5 \times 10^{-7}$ in Shao et al. (2025)), and (iv) a reduced batch size (64 versus 128). The smaller learning rate changes the effective update magnitude, and the smaller batch size yields noisier estimates of the stochastic reward distribution. Given these differences, the empirical results reported across prior works are not directly comparable.

**Contamination.** We further comment on potential contamination in the `Qwen2.5-Math` models. As reported by Wu et al. (2025), contamination in `Qwen-Math` has been observed only on validation sets (e.g., `MATH500`). Beyond these findings, the official `Qwen2.5-Math` technical report (Yang et al., 2024, Table 1) also acknowledges possible contamination arising from the close similarity between the training and validation sets of the `MATH` dataset (Hendrycks et al., 2021). Their training corpus comprises two components: (i) CoT data synthesis, which includes GSM8K, MATH, and NuminaMath (Yang et al., 2024, §3.1.1), and (ii) a tool-integrated reasoning dataset containing GSM8K, MATH, CollegeMath, NuminaMath, MuggleMath, and DotaMath (Yang et al., 2024, §3.1.2). In contrast, our experiments employ the `DeepScaleR` training set, which consists exclusively of selected questions from AMC, AIME, Omni-Math, and Still (Luo et al., 2025). None of these datasets appear in the training sources listed for `Qwen2.5-Math`. Therefore, we believe that our training data does not overlap with the datasets used to train `Qwen2.5-Math` and is thus not contaminated.

**LLM entropy.** Agarwal et al. (2025) demonstrate that token-level entropy minimization can substantially improve LLM reasoning without verifiable feedback, arguing that reduced entropy increases model confidence and reveals latent reasoning capability. This mechanism parallels *clipped training under random rewards*, where updates primarily modulate entropy rather than exploit informative rewards. However, we show that entropy minimization alone may drive the policy toward

low-entropy yet suboptimal solutions; hence, entropy should be viewed as a stabilizing regularizer rather than a replacement for genuine RLVR signals.

Related work explores entropy through the lens of self-confidence. In particular, Prabhudesai et al. (2025) use low-entropy rollouts as implicit rewards, achieving gains across diverse benchmarks, while Gao et al. (2025) show that even a single unlabeled example can improve reasoning via entropy reduction. Methods such as EMPO (Zhang et al., 2025b) and Zhao et al. (2025b) similarly enhance performance in unsupervised settings by amplifying model confidence. van Niekerk et al. (2025) further construct preference datasets from confidence scores, achieving RLHF-level improvements without human feedback. In this context, Cui et al. (2025) propose a simple but influential empirical relationship between policy entropy $\mathcal{H}$ and model performance $R$, fit across extensive experiments:

$$R = -a \exp\left(\mathcal{H}\right) + b, \qquad a > 0.$$

This relation suggests that performance increases monotonically as entropy decreases but plateaus once entropy collapses too early. Intuitively, when a model overemphasizes certain tokens, its output distribution becomes overconfident and loses exploratory capacity, leading to a performance ceiling.

To mitigate early-stage entropy collapse, several works propose alternative strategies. Shen (2025) analyze why entropy regularization suffers from limited benefit in RLVR training for LLMs by attributing it to the vast response space and the sparsity of optimal outputs, and then introduce an adaptive entropy-control method using a clamped entropy bonus with automatically tuned coefficients. Song et al. (2025) show that ORM yields induces sharp reductions in output entropy and diversity (as shown by lower pass@n scores), and propose outcome-level entropy bonuses to counteract it. Prior works (Wang et al., 2025a; Yao et al., 2025; Zheng et al., 2025; Cheng et al., 2025) also develop additional techniques for controlling entropy during RLVR training.

**Reinforcement learning for LLM.**   Proximal policy optimization (PPO) (Schulman et al., 2017) has become standard for reward-based policy updates in LLM training and remains a core component of RLHF. However, since PPO requires loading and maintaining four separate models during training, it is computationally and memory intensive. This has motivated the development of lighter-weight and adapted policy-gradient updates (Li et al., 2024b; Ahmadian et al., 2024; Shao et al., 2024; Guo et al., 2025). In parallel, advances in verifiable reward construction (Cobbe et al., 2021; Uesato et al., 2022; Zelikman et al., 2022; Singh et al., 2023; Hosseini et al., 2024; Lightman et al., 2024; Wang et al., 2024; Luo et al., 2024; Setlur et al., 2025; Zhang et al., 2025c) have enabled reinforcement learning to substantially improve LLM reasoning, particularly in mathematical problem solving. Beyond training algorithms, recent work also explores post-processing and collaborative strategies to strengthen reasoning performance. Kay et al. (2025) and Zhao et al. (2025a) propose consensus-based and answer-aggregation methods within multi-model frameworks. Chen et al. (2025a) introduce a self-questioning paradigm for iterative refinement, while Park et al. (2025) develop an online multi-agent collaborative reinforcement learning framework.

**Offline alignment.**   *Direct alignment* methods (Rafailov et al., 2023) provide a simple and stable offline alternative to RLHF. Extensions to DPO include broader ranking objectives (Dong et al., 2023; Yuan et al., 2023; Song et al., 2024; Chen et al., 2024a; Liu et al., 2025a) and lightweight variants that remove the reference model (Hong et al., 2024; Meng et al., 2024). Since DPO avoids reward model, limited human preference data becomes a key bottleneck; recent work addresses this by generating additional preference pairs via SFT policies (Zhao et al., 2023; Liu et al., 2024a). The framework has also been generalized to token-level MDPs (Rafailov et al., 2024) and broader RL settings (Azar et al., 2024). Complementary approaches incorporate online human feedback to reduce distribution shift and improve reasoning (Dong et al., 2024; Xiong et al., 2024; Pang et al., 2024). Another line studies *unintentional alignment* and proposes remedies (Pal et al., 2024; Tajwar et al., 2024; Liu et al., 2024b; Xiao et al., 2024; Yuan et al., 2025; Razin et al., 2025; Chen et al., 2025b). For example, Razin et al. (2025) filter noisy preference pairs using CHES similarity, while Chen et al. (2025b) show that combining comparison oracles with DPO mitigates unintended alignment effects.

# B   ADDITIONAL EXPERIMENTAL RESULTS

We begin with a high-level overview of the experimental setup, followed by a comprehensive presentation of results in both the main text and the appendix. Our experiments are organized around two

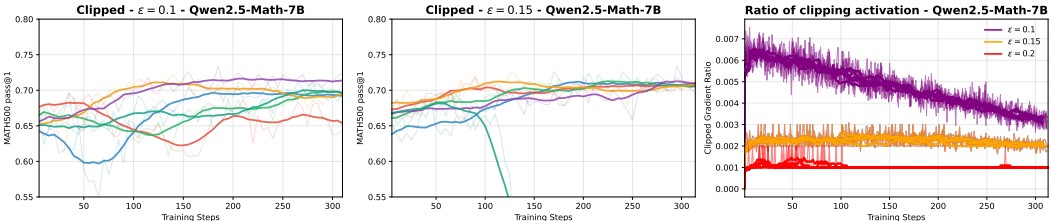

Figure 5: All experiments follow the same setup as Figure 1, varying the threshold $\varepsilon$ with six independent runs for each setting: trials with clipping ratio $\varepsilon = 0.1$ (**Left**); trials with clipping ratio $\varepsilon = 0.15$ (**Middle**); and the ratio of clipping activations across $\varepsilon \in \{0.2, 0.15, 0.1\}$ (**Right**).

objectives: (i) characterizing the interplay between clipping, policy entropy, and performance under spurious rewards, and (ii) assessing whether the observed benefits of spurious rewards generalize beyond `Qwen-Math` to a broader range of model families.

For the first objective, we focus on `Qwen-Math-7B` and provide a controlled setting for examining how clipping and policy entropy affect model performance under spurious rewards. This choice is supported by previous empirical findings and practical considerations: `Qwen-Math-7B` has a moderate parameter count and a relatively short 4K context window, stabilizing training and reducing exposure to issues such as gradient explosion. This stability is crucial since clipping is commonly used to prevent gradient explosion in larger models with longer chain-of-thought rollouts. As shown in Figure 2 (**Right**) and discussed in §4.2, removing clipping on a stronger model with longer rollouts can cause catastrophic training collapse, making it difficult to disentangle the core relationship between clipping, entropy, and performance. Indeed, one key motivation for applying the clipping in GRPO originates from the need to stabilize training for the `DeepSeek-R1-671B` model. For the second objective, we additionally evaluate two non-contaminated model families, `Llama` and `QwQ`, for which no contamination has been reported in the community, to demonstrate that the benefits and behaviors of spurious rewards extend beyond `Qwen-Math` and reflect broader RLVR learning.

In §3.2, we examine how clipping affects model performance by comparing training with and without clipping. In §4.2, we validate our theoretical findings on the relationship between clipping and policy entropy. For consistency, these experiments use `Qwen-Math-7B` trained on the DeepScaleR dataset. Then, in §4.3 and this section, we investigate the interaction between entropy and performance on the more challenging AIME training set, again evaluating both clipped and unclipped training. We find that policy entropy is not directly related to performance improvements, and that models gain significantly less from random rewards when their baseline performance is reduced by dataset difficulty. This stands in contrast to stronger `Llama` and `QwQ` models, which continue to benefit from random rewards even on harder tasks. Finally, in §5, we proceed to a broader spectrum of model strengths, showing that stronger models are more likely to benefit from random reward signals.

**Ablation analysis.** We ablate the GRPO group size $G$. Larger groups ($G = 16$) yield more balanced binary rewards, while smaller groups increase the likelihood of extreme reward-misalignment events, such as entire groups receiving reward 0 despite containing correct rollouts, or reward 1 despite containing incorrect ones. Thus, reducing $G$ inherently amplifies instability from the reward-misalignment perspective. As shown in Figure 6, using a smaller group size ($G = 8$) allows most runs to improve, but leads to higher variance and less stable learning dynamics throughout training.

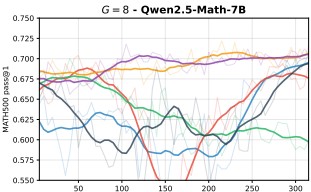

Figure 6: Smaller group size.

We further analyze the effect of varying the clipping ratio threshold $\varepsilon$. Indeed, Figure 1 examined the cases $\varepsilon = 0.2$ and $\varepsilon = \infty$ (no clipping), showing that relaxing the clipping threshold does not degrade performance. However, this does not yet confirm robustness under stricter clipping. To address this, we report additional results for $\varepsilon \in \{0.15, 0.1\}$ in Figure 5. Across these settings, we observe behavior consistent with Figure 1: (i) some runs fail to improve, as predicted by our probabilistic reward-misalignment framework; and (ii) successful runs converge to roughly 70% validation accuracy regardless of the clipping strength. As discussed in §4.1, clipping primarily influences policy entropy. Among the improving trials, stricter clipping tends to reduce variance

across seeds, reflecting a more deterministic policy toward convergence. Taken together, these results indicate that our findings remain robust under different choices of $\varepsilon$.

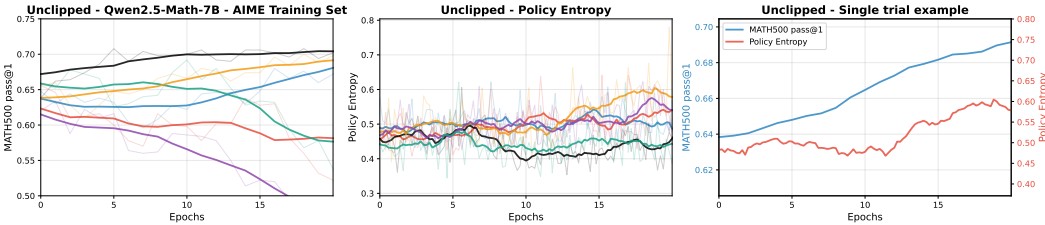

Figure 7: Unclipped `Qwen2.5-Math-7B` on the hard AIME dataset: independent runs following from the setup in Figure 3 (**Left**); corresponding policy entropy dynamics during unclipped training (**Middle**); joint evolution of model performance and policy entropy for an example trial (**Right**).

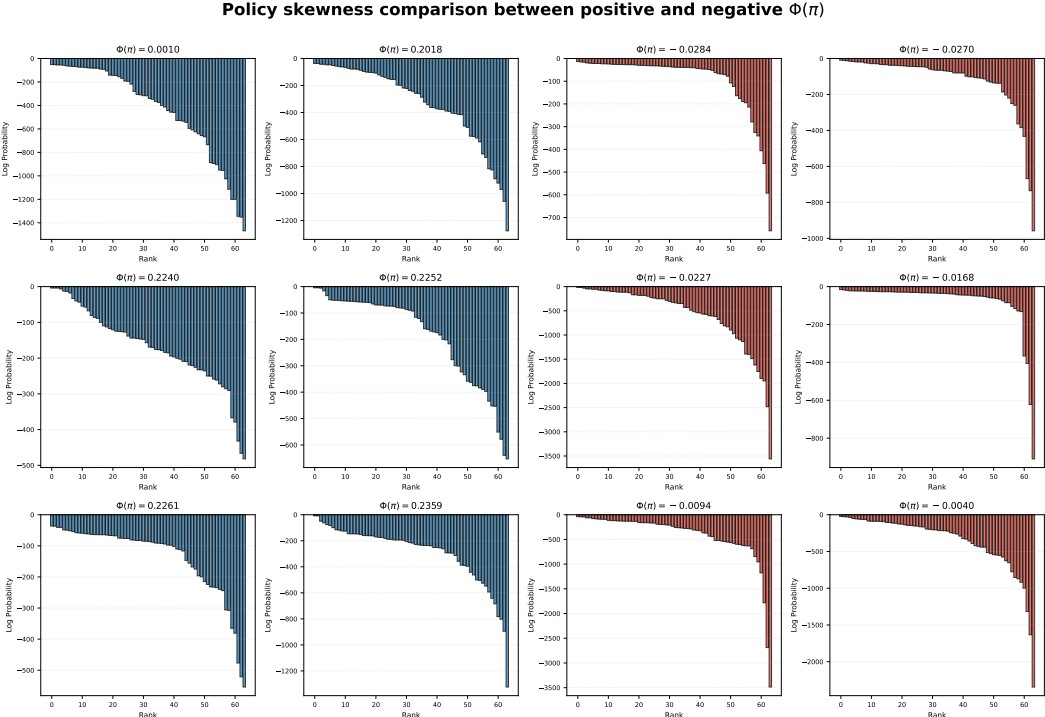

Figure 8: Visualization of policy action distributions over 12 prompt $\mathbf{x}_i$. Each subplot displays the sorted log-probability of $\pi(\mathbf{y} \mid \mathbf{x}_i)$ for 64 sampled responses from each prompt $\mathbf{x}_i$. Columns 1-2 (blue) correspond to prompts $\mathbf{x}_i$ with $\Phi(\pi(\cdot \mid \mathbf{x}_i)) > 0$, while Columns 3-4 (orange) correspond to prompts with $\Phi(\pi(\cdot \mid \mathbf{x}_i)) < 0$. As discussed in Theorem 4.2, the entropy increase under unclipped training can occur only for the skewed one shown in Columns 3-4.

**Unclipped training.** In Figure 3, we present clipped training results for `Qwen2.5-Math-7B` on the AIME dataset, where clipping induces entropy collapse as demonstrated by Theorem 4.3. Although entropy decreases, performance also degrades, indicating that lower entropy is not reliably associated with better model performance. This raises an important question: what happens in the complementary regime where entropy *increases* under random reward? To answer this, we conduct additional experiments and report the results in Figure 7. Across independent seeds shown in Figure 7 (Left), the behavior remains qualitatively similar to the clipped case in Figure 3: some runs improve, others degrade, and overall learning dynamics appear stochastic. In Figure 7 (Middle), we empirically confirm the predicted entropy increase under unclipped training. Among the improving runs, Figure 7 (Right) provides a representative example in which performance improves even as entropy increases. These results answers our question: there is **no** direct causal relationship between policy entropy

and model performance. Both clipped and unclipped experiments support that, as the model's initial performance on a dataset decreases, its likelihood of benefiting from random rewards diminishes.

**Policy skewness.** We empirically evaluate the skewness measure $\Phi(\pi)$ introduced in Remark 4.2 on the actual `Qwen-Math-7B` policy. Recall that under unclipped training, entropy can increase after a single update only when $\Phi(\pi) < 0$. Since the policy induces different action distributions $\pi(a \mid \mathbf{x})$ for different input questions $\mathbf{x}$, we estimate skewness across questions by sampling the first 500 examples from the DeepScaleR training set (Luo et al., 2025). For each question $\mathbf{x}$, we generate 64 responses $\mathbf{y}$ from the policy using the same sampling and decoding hyperparameters as in §3.2 and compute an empirical estimate of $\Phi(\pi(\cdot \mid \mathbf{x}))$. We visualize selected prompts $\mathbf{x}_i$ along with their corresponding skewness values in Figure 8, providing a clearer picture of how `Qwen-Math-7B` behaves across the dataset. Among 500 sampled questions, 358 ones satisfy $\Phi(\pi(\cdot \mid \mathbf{x}_i)) < 0$, which is consistent with the observed entropy increases for the unclipped training.

## C  THEORETICAL ANALYSIS

**Setup.** We model the next-token generation using a softmax at each history. Let $\mathcal{V}$ be the vocabulary and $\mathbf{h}_t = (\mathbf{x}, \mathbf{y}_{<t})$ be the history. For each prompt $\mathbf{x} \in \mathcal{X}$ and response $\mathbf{a} = (a_1, \dots, a_L) \in \mathcal{V}^L$ where $a_t \in \mathcal{V}$, we have

$$\pi_\theta(\mathbf{a} \mid \mathbf{x}) = \prod_{t=1}^{L} \pi_{\theta_t}(a_t \mid \mathbf{h}_t), \quad \text{where } \pi_{\theta_t}(a_t \mid \mathbf{h}_t) = \frac{\exp(\theta_{t,\mathbf{h}_t,a_t})}{\sum_{a' \in \mathcal{V}} \exp(\theta_{t,\mathbf{h}_t,a'})},$$

where $\theta = (\theta_1^\top, \dots, \theta_L^\top)^\top$, and $\theta_t \in \mathbb{R}^{|\mathcal{X}||\mathcal{V}|^t}$ for all $t = 1, \dots, L$.

Given trajectories drawn from $\pi_{\text{old}}$, we define the per-token ratio $r_t^{(i)}(\theta) = \frac{\pi_\theta(\mathbf{y}_t^{(i)} \mid \mathbf{h}_t^{(i)})}{\pi_{\text{old}}(\mathbf{y}_t^{(i)} \mid \mathbf{h}_t^{(i)})}$. For a group $\{\mathbf{y}^{(i)}\}_{i=1}^{G} \sim \pi_{\text{old}}(\cdot \mid \mathbf{x})$ and the corresponding outcome-reward advantages $\{A_i\}_{i=1}^{G}$, the empirical per-history advantage used in the policy update is

$$\tilde{A}(\mathbf{h}, a) = \frac{1}{G} \sum_{i=1}^{G} \sum_{t=1}^{L} \left( \frac{\mathbf{1}\{\mathbf{h}_t^{(i)} = \mathbf{h}, \mathbf{y}_t^{(i)} = a\}}{\pi_{\text{old}}(a \mid \mathbf{h})} \right) A_i.$$

This can be derived from Eq. (3) using $L = |\mathbf{y}^{(i)}|$ for all $i = 1, 2, \dots, G$.

Following Williams (1992) and Li et al. (2024b), we define the clipped surrogate loss with per-token ratios without adding separate length normalization terms as follow,

$$J(\theta) = \mathbb{E}_{\mathbf{x} \sim \rho, \{\mathbf{y}^{(i)}\}_{i=1}^{G} \sim \pi_{\theta_{\text{old}}}(\cdot \mid \mathbf{x})} \left[ \frac{1}{G} \sum_{i=1}^{G} \sum_{t=1}^{L} \min \left\{ r_t^{(i)}(\theta) A_i, \texttt{clip}(r_t^{(i)}(\theta), 1 - \varepsilon, 1 + \varepsilon) A_i \right\} \right],$$

Without clipping, the surrogate loss reduces to

$$J(\theta) = \mathbb{E}_{\mathbf{x} \sim \rho, \{\mathbf{y}^{(i)}\}_{i=1}^{G} \sim \pi_{\theta_{\text{old}}}(\cdot \mid \mathbf{x})} \left[ \frac{1}{G} \sum_{i=1}^{G} \sum_{t=1}^{L} r_t^{(i)}(\theta) A_i \right].$$

We derive the closed-form token-level update for optimizing the unclipped surrogate loss with a forward KL penalty to $\pi_{\text{old}}$ as follows. To begin with, notice that

$$\hat{J}(\theta) = \frac{1}{G} \sum_{i=1}^{G} \sum_{t=1}^{L} r_t^{(i)}(\theta) A_i = \sum_{\mathbf{h}} \sum_{a \in \mathcal{V}} \pi_\theta(a \mid \mathbf{h}) \underbrace{\left[ \frac{1}{G} \sum_{i=1}^{G} \sum_{t=1}^{L} \frac{\mathbf{1}\{\mathbf{h}_t^{(i)} = \mathbf{h}, \mathbf{y}_t^{(i)} = a\}}{\pi_{\text{old}}(a \mid \mathbf{h})} A_i \right]}_{\tilde{A}(\mathbf{h}, a)}.$$

Using mirror descent (MD), for each iteration, one solves

$$\max_\theta \quad \hat{J}(\theta) - \frac{1}{\eta} \sum_{\mathbf{h}} D_{\text{KL}}(\pi_\theta(\cdot \mid \mathbf{h}) \,\|\, \pi_{\text{old}}(\cdot \mid \mathbf{h}))$$

which is equivalent to solving for each fixed $\mathbf{h}$,

$$\max_{\pi(\cdot|\mathbf{h})} \quad \sum_a \pi(a \mid \mathbf{h})\tilde{A}(\mathbf{h}, a) - \frac{1}{\eta}\sum_a \pi(a \mid \mathbf{h})\log\frac{\pi(a|\mathbf{h})}{\pi_{\text{old}}(a|\mathbf{h})}.$$

Introducing a Lagrangian multiplier $\lambda_{\mathbf{h}}$ for the probability simplex constraint $\sum_a \pi(a \mid \mathbf{h}) = 1$, by first order condition

$$\tilde{A}(\mathbf{h}, a) - \frac{1}{\eta}\left(\log\frac{\pi(a|\mathbf{h})}{\pi_{\text{old}}(a|\mathbf{h})} + 1\right) + \lambda_{\mathbf{h}} = 0.$$

Solving the above equation for $\pi_\theta$ yields

$$\pi_\theta(a \mid \mathbf{h}) = \frac{\pi_{\theta_{\text{old}}}(a|\mathbf{h})\exp(\eta\tilde{A}(\mathbf{h},a))}{\sum_{a' \in \mathcal{V}}\pi_{\theta_{\text{old}}}(a'|\mathbf{h})\exp(\eta\tilde{A}(\mathbf{h},a'))},$$

and evaluate at the realized pairs $(a, \mathbf{h}) = (\mathbf{y}_t^{(i)}, \mathbf{h}_t^{(i)})$ in training. Note that the above GRPO update can be analyzed by interpreting it as one natural policy gradient (NPG) step under softmax tabular parametrization (Agarwal et al., 2021).

---

**Algorithm 1** Iterative Group Relative Policy Optimization

---

1: **Input:** model parameters $\theta_{\text{init}}$, reward models $r_\varphi$, prompts $\mathcal{X}$, and hyperparameters $\varepsilon, \beta, \mu$.
2: **Initialization:** $\theta \leftarrow \theta_{\text{init}}$.
3: **for** iteration $= 1, \cdots, I$ **do**
4:     $\pi_{\text{ref}} \leftarrow \pi_\theta$.
5:     **for** $j = 1, \cdots, M$ **do**
6:         Sample a batch $\mathcal{X}_j$ from $\mathcal{X}$.
7:         Update the old policy model $\pi_{\theta_{\text{old}}} \leftarrow \pi_\theta$.
8:         Sample $G$ outputs $\{o_i\}_{i=1}^G \sim \pi_{\theta_{\text{old}}}(\cdot \mid \mathbf{x})$ for each question $\mathbf{x} \in \mathcal{X}_j$.
9:         Compute rewards $\{r_i\}_{i=1}^G$ for each sampled output $o_i$ using the reward model $r_\varphi$.
10:       Compute $\hat{A}_{i,t}$ for the $t$-th token of $o_i$ via group-relative advantage estimation.
11:       Update the policy model $\pi_\theta$ using GRPO.
12:     **end for**
13:     Update $r_\varphi$ through continuous training using a replay mechanism.
14: **end for**
15: **Output:** $\pi_\theta$.

---

**GRPO analysis.** Interpreting the GRPO update as a natural policy gradient (NPG) step has been widely adopted to study entropy dynamics throughout training (Cui et al., 2025). Here, we summarize the key components of GRPO in Algorithm 1, which motivate our reduction to an NPG-style update for analyzing the effect of clipping in GRPO. We note that Algorithm 1 should be viewed as an abstraction of GRPO implementations used in practice (Shao et al., 2024).

In the outer loop, a reference policy is fixed once per iteration, and the per-step objective may include a KL penalty that constrains the updated policy $\pi_\theta$ to remain close to $\pi_{\text{ref}}$, thereby controlling the effective step size and preventing excessive policy drift. Recent "zero-RL" setups (see Yu et al., 2025), which are also adopted in the empirical evaluation of Shao et al. (2025), set the KL coefficient to zero, effectively removing the explicit KL term from the objective. As such, we likewise omit the KL term in our analysis. Under this regime, the outer loop does not affect the subsequent analysis.

In the middle loop, which is for GRPO training, the model samples each batch, which is update-style agnostic. The key difference between exact-GRPO- and NGP-style update happens in the inner loop (line 10). First, $\mu$ is a constant hyperparameter for the number of actual updates per macro batch, used to improve sample efficiency and better optimize the surrogate while clipping limits drift from $\pi_{\text{old}}$. Therefore, the statement *for GRPO iteration* $= 1, \ldots, \mu$ performs $\mu$ optimizer steps on the *same* mini-batch to maximize the clipped GRPO surrogate. At each step, importance ratio $r_t^{(i)} = \frac{\pi_\theta(\mathbf{y}_t^{(i)}|\mathbf{x})}{\pi_{\text{old}}(\mathbf{y}_t^{(i)}|\mathbf{x})}$ are recomputed and the loss $\frac{1}{G}\sum_{i,t}\min\{r_t^{(i)}\tilde{A}, \text{clip}(r_t^{(i)}, 1-\varepsilon, 1+\varepsilon)\tilde{A}\}$ is backpropagated.

In GRPO, the $\mu$-step inner loop produces a chain of micro-updates whose importance ratios $r$ evolve across steps, making the *expected* contribution of clipping analytically intractable unless one specifies the per-step *clip-activation rate* (the expected fraction of tokens/micro-batches with $r \notin [1-\varepsilon, 1+\varepsilon]$).

This rate is model- and dataset-dependent and is only available empirically. Conditioning on the empirically measured activation rate, we collapse the $\mu$ clipped micro-steps into a single NPG-update with actual model-specific token-level expected clipping activation ratio. This surrogate preserves the first-order effect of clipping and enables tractable bounds for our theoretical results. Comparing to recent works that directly used NPG for GRPO analysis, our setup for clipping analysis is validly justified, facilitating the later theoretical derivation and without unjustified oversimplification.

### C.1 MISSING PROOFS IN §2

**Proof of Theorem 2.2.** Fixing $\mathbf{h}$, we rewrite Eq. (2) as

$$\pi_{\text{new}}(a \mid \mathbf{h}) = \frac{\pi_{\text{old}}(a|\mathbf{h}) \exp(\eta \tilde{A}(\mathbf{h},a))}{Z_{\mathbf{h}}(\eta)},$$

where

$$Z_{\mathbf{h}}(\eta) = \sum_{a \in \mathcal{V}} \pi_{\text{old}}(a \mid \mathbf{h}) \exp(\eta \tilde{A}(\mathbf{h}, a)) = \mathbb{E}_{a \sim \pi_{\text{old}}(\cdot|\mathbf{h})}[\exp(\eta \tilde{A}(\mathbf{h}, a))].$$

Taking the logarithm of both sides yields

$$\log(\pi_{\text{new}}(a \mid \mathbf{h})) = \log(\pi_{\text{old}}(a \mid \mathbf{h})) + \eta \tilde{A}(\mathbf{h}, a) - \log(Z_{\mathbf{h}}(\eta)). \tag{10}$$

We define $\psi_{\mathbf{h}}(\eta) = \log(Z_{\mathbf{h}}(\eta))$. Then, we have $\psi_{\mathbf{h}}(0) = 0$, $\psi'_{\mathbf{h}}(0) = \mu(\mathbf{h})$, and $\psi''_{\mathbf{h}}(0) = \sigma^2(\mathbf{h})$. Fixing $(\mathbf{h}, a)$, we define $I_i(\mathbf{h}, a) := \sum_{t=1}^{|\mathbf{y}^{(i)}|} \mathbf{1}\{\mathbf{h}_t^{(i)} = \mathbf{h}, \mathbf{y}_t^{(i)} = a\} \in \{0, 1\}$ and $N := \sum_{i=1}^{G} I_i(\mathbf{h}, a)$. Then, we have

$$\tilde{A}(\mathbf{h}, a) = \frac{1}{\pi_{\text{old}}(a|\mathbf{h})G} \left( \sum_{i=1}^{G} I_i(\mathbf{h}, a)A_i \right).$$

By using $\sum_{i=1}^{G} A_i = 0$ and $\sum_{i=1}^{G} A_i^2 \leq G$, we have

$$\left| \sum_{i=1}^{G} I_i(\mathbf{h}, a)A_i \right| \leq \sqrt{N(G - N)} \leq \frac{G}{2}.$$

This implies $|\tilde{A}(\mathbf{h}, a)| \leq \frac{1}{2\pi_{\min}}$ for all $a \in \mathcal{V}$. By using Taylor's theorem with Lagrange remainder, we have

$$\psi_{\mathbf{h}}(\eta) = \mu(\mathbf{h})\eta + \frac{1}{2}\sigma^2(\mathbf{h})\eta^2 + \frac{1}{6}\psi'''_{\mathbf{h}}(\xi)\eta^3 \text{ for some } \xi \in (0, \eta).$$

Since $\psi'''_{\mathbf{h}}(\eta)$ is the third central moment of $\tilde{A}(\mathbf{h}, a)$ under the exponentially tilted distribution and $\tilde{A}(\mathbf{h}, a) \in [-M, M]$, the sharp bound $|\mathbb{E}[(X - \mathbb{E}[X])^3]| \leq \frac{(b-a)^3}{6\sqrt{3}}$ for $X \in [a, b]$ yields

$$|\psi'''_{\mathbf{h}}(\eta)| \leq \frac{1}{6\sqrt{3}(\pi_{\min})^3}.$$

Therefore, we conclude that

$$\left| \psi_{\mathbf{h}}(\eta) - \mu(\mathbf{h})\eta - \frac{1}{2}\sigma^2(\mathbf{h})\eta^2 \right| \leq \frac{\eta^3}{36\sqrt{3}(\pi_{\min})^3}.$$

Combining this with Eq. (10) yields the claimed inequality with $C = \frac{1}{36\sqrt{3}(\pi_{\min})^3}$. The statements for $\log(r(\mathbf{h}, a))$ and the standardized case follow immediately. $\square$

**Proof of Theorem 2.4.** We prove three statements one by one as follows.

(i) Let $\tau : \{0,1\}^G \to \{0,1\}^G$ be $\tau(\mathbf{r}_1, \ldots, \mathbf{r}_G) = (1 - \mathbf{r}_1, \ldots, 1 - \mathbf{r}_G)$. If $(\mathbf{r}'_1, \ldots, \mathbf{r}'_G) = \tau(\mathbf{r}_1, \ldots, \mathbf{r}_G)$, then we can also define $A'_i$ and $\bar{\mathbf{r}}'$ similar to $A_i$ and $\bar{\mathbf{r}}$. With the notation above, we have $\bar{\mathbf{r}}' = 1 - \bar{\mathbf{r}}$ and

$$\mathbf{r}'_j - \bar{\mathbf{r}}' = (1 - \mathbf{r}_j) - (1 - \bar{\mathbf{r}}) = -(\mathbf{r}_j - \bar{\mathbf{r}}).$$

Hence $\mathbf{S}_{\mathbf{r}'} = \mathbf{S}_{\mathbf{r}}$ and $A'_i = -A_i$. Since $(\mathbf{r}_1, \ldots, \mathbf{r}_G)$ is i.i.d. Bernoulli$(\frac{1}{2})$, its law is invariant under $\tau$. Thus, we know $(\mathbf{r}'_1, \ldots, \mathbf{r}'_G) \overset{d}{=} (\mathbf{r}_1, \ldots, \mathbf{r}_G)$ and $A'_i \overset{d}{=} A_i$. Combining the above two facts, we obtain $A_i \overset{d}{=} -A_i$ and thus $\mathbb{E}[A_i^{2k-1}] = 0$.

(ii) Let $K := \sum_{j=1}^{G} \mathbf{r}_j$, then $\bar{\mathbf{r}} = \frac{K}{G}$ and $\mathbf{S}_{\mathbf{r}}^2 = \frac{K(G-K)}{G(G-1)}$. Thus,

$$|A_i| = \begin{cases} \sqrt{G-1}\sqrt{\frac{G-K}{GK}} & \text{if } \mathbf{r}_i = 1, \\ \sqrt{G-1}\sqrt{\frac{K}{G(G-K)}} & \text{if } \mathbf{r}_i = 0. \end{cases}$$

Thus, it is easy to see $|A_i| \leq \sqrt{G} - \frac{1}{\sqrt{G}}$.

(iii) Let $K := \sum_{j=1}^{G} \mathbf{r}_j \sim \text{Binomial}(G, \frac{1}{2})$ and $p := \frac{K}{G}$. On $\{1 \leq K \leq G-1\}$, we have

$$\mathbf{S}_{\mathbf{r}} = \sqrt{p(1-p)}, \qquad A_i = \begin{cases} \sqrt{\frac{1-p}{p}}, & \mathbf{r}_i = 1, \\ -\sqrt{\frac{p}{1-p}}, & \mathbf{r}_i = 0. \end{cases}$$

Hence for $k \in \mathbb{N}^+$,

$$\mathbb{E}[|A_i|^k \mid K] = p\left(\frac{1-p}{p}\right)^{k/2} + (1-p)\left(\frac{p}{1-p}\right)^{k/2} = \frac{x^{k/2} + x^{1-k/2}}{1+x},$$

with $x := \frac{p}{1-p} > 0$. Define $h_k(x) := x^{k/2} + x^{1-k/2} - x - 1$. Then

$$h_k''(x) = \frac{k}{2}\left(\frac{k}{2} - 1\right)x^{k/2-2} + \frac{k}{2}\left(\frac{k}{2} - 1\right)x^{-k/2-1} \geq 0, \quad \forall\, x > 0 \text{ if } k \geq 2,$$

and $h_k(1) = h_k'(1) = 0$. By convexity, $h_k(x) \geq 0$ for all $x > 0$, hence $\mathbb{E}[|A_i|^k \mid K] \geq 1$ whenever $1 \leq K \leq G-1$. Taking the expectation and using the fact that $A_i = 0$ on $\{K \in \{0, G\}\}$ yields

$$\mathbb{E}[|A_i|^k] = \sum_{K=1}^{G-1}\binom{G}{K}2^{-G}\mathbb{E}[|A_i|^k \mid K] \geq \sum_{k=1}^{G-1}\binom{G}{k}2^{-G} = 1 - 2^{1-G}, \quad \text{if } k \geq 2.$$

Finally, it is trivial to write down $\mathbb{E}[|A_i|]$ with the above information.

This completes the proof. $\qquad\square$

**Failure of Eq. (5) under random reward.** We have

$$\text{Cov}_{\mathbf{y}\sim\pi_{\text{old}}(\cdot|\mathbf{x})}\left(\log(\pi_{\text{old}}(\mathbf{y} \mid \mathbf{x})), A(\mathbf{x}, \mathbf{y})\right)$$
$$= \mathbb{E}_{\mathbf{y}\sim\pi_{\text{old}}(\cdot|\mathbf{x})}\left[\log(\pi_{\text{old}}(\mathbf{y} \mid \mathbf{x}))A(\mathbf{x}, \mathbf{y})\right] - \mathbb{E}_{\mathbf{y}\sim\pi_{\text{old}}(\cdot|\mathbf{x})}\left[\log(\pi_{\text{old}}(\mathbf{y} \mid \mathbf{x}))\right]\underbrace{\mathbb{E}_{\mathbf{y}\sim\pi_{\text{old}}(\cdot|\mathbf{x})}[A(\mathbf{x}, \mathbf{y})]}_{=0}$$
$$= \mathbb{E}_{\mathbf{y}\sim\pi_{\text{old}}(\cdot|\mathbf{x})}\left[\log(\pi_{\text{old}}(\mathbf{y} \mid \mathbf{x}))\right]\underbrace{\mathbb{E}_{\mathbf{y}\sim\pi_{\text{old}}(\cdot|\mathbf{x})}[A(\mathbf{x}, \mathbf{y})]}_{=0} = 0.$$

In other words, the co-variance between $A(\mathbf{x}, \mathbf{y})$ and $\pi_{\text{old}}(\mathbf{y} \mid \mathbf{x})$ is uninformative under random reward since these two terms are independent from each other in this specific setting. Thus, a more accurate estimation of $\mathcal{H}(\pi_{\text{new}}) - \mathcal{H}(\pi_{\text{old}})$ beyond Eq. (5) is desirable.

### C.2 MISSING PROOFS IN §3

**Proof of Theorem 3.2.** By definition, we have $C_{\text{tot}}^+ = \sum_{t=1}^{L} D_t^+ A$. Thus, we have

$$\mathbb{E}[|C_{\text{tot}}^+|^2] = \sum_{s=1}^{L}\sum_{t=1}^{L}\mathbb{E}[D_s^+ D_t^+ A^2] = \sum_{t=1}^{L}\mathbb{E}[(D_t^+)^2 A^2] + \sum_{s\neq t}\mathbb{E}[D_s^+ D_t^+ A^2]. \qquad (11)$$

Recall in the proof of Theorem 2.2, we have shown that $|\tilde{A}(\mathbf{h}, a)| \leq \frac{1}{2\pi_{\min}}$. We also have

$$r(\mathbf{h}, a) = \frac{e^{\eta\tilde{A}(\mathbf{h},a)}}{Z_{\mathbf{h}}(\eta)}, \qquad Z_{\mathbf{h}}(\eta) = \sum_{a}\pi_{\text{old}}(a \mid \mathbf{h})e^{\eta\tilde{A}(\mathbf{h},a)}.$$

Since $\tilde{A}(\mathbf{h}, a) \geq -\frac{1}{2\pi_{\min}}$, we have $Z_{\mathbf{h}}(\eta) \geq e^{-\eta/(2\pi_{\min})}$ and $e^{\eta\tilde{A}(\mathbf{h},a)} \leq e^{\eta/(2\pi_{\min})}$. Thus, we have $r(\mathbf{h}, a) \leq e^{\eta/\pi_{\min}} =: R_{\eta}^{\max}$. This implies $r_t \leq R_{\eta}^{\max}$ for all $t$. In what follows, we bound the diagonal and off-diagonal terms in the right-hand side of Eq. (11).

**Diagonal term.** On $\{I_t^+ = 1\}$, we have $r_t > 1 + \varepsilon \geq 1$ and $\bar{r}_t = 1 + \varepsilon$. Thus, we have

$$|D_t^+| = |(1 + \varepsilon - r_t)I_t^+| = (r_t - 1 - \varepsilon)I_t^+ \leq (r_t - 1)I_t^+.$$

By using the fact that $(x - 1)^2 \leq 2x\phi(x)$ for all $x \geq 1$, we have

$$(D_t^+)^2 \leq (r_t - 1)^2 I_t^+ \leq 2r_t\phi(r_t)I_t^+.$$

Since $|A| \leq M$ (c.f. Theorem 2.4), we have

$$\mathbb{E}[(D_t^+)^2 A^2] \leq M^2 \mathbb{E}[(D_t^+)^2] \leq 2M^2 \mathbb{E}[r_t\phi(r_t)I_t^+].$$

Since $\phi$ is strictly increasing and $r_t \leq R_\eta^{\max}$, we have $r_t\phi(r_t)I_t^+ \leq R_\eta^{\max}\phi(R_\eta^{\max})I_t^+$. Putting these pieces together yields

$$\sum_{t=1}^{L} \mathbb{E}[(D_t^+)^2 A^2] \leq 2LM^2 R_\eta^{\max}\phi(R_\eta^{\max})\mathbb{E}[I_t^+] = 2LM^2 R_\eta^{\max}\phi(R_\eta^{\max})p_+. \tag{12}$$

**Off-diagonal term.** We define $Z_t := D_t^+ A$ and have $|Z_t| \leq |D_t^+||A| \leq M|D_t^+|$. Then, we define $\Delta_\eta^+ := (R_\eta^{\max} - 1 - \varepsilon)_+$ and have

$$|D_t^+| = (r_t - 1 - \varepsilon)I_t^+ \leq (R_\eta^{\max} - 1 - \varepsilon)_+ I_t^+ = \Delta_\eta^+ I_t^+.$$

Putting these pieces together yields $|Z_t| \leq M\Delta_\eta^+ I_t$ and

$$\sum_{s \neq t} \mathbb{E}[|Z_s Z_t|] \leq M^2 (\Delta_\eta^+)^2 \sum_{s \neq t} \mathbb{E}[I_s^+ I_t^+] \leq M^2 (\Delta_\eta^+)^2 \mathbb{E}\left[\left(\sum_{t=1}^{L} I_t\right)^2\right]. \tag{13}$$

Since $\sum_{t=1}^{L} I_t^+ \leq L$, we have $(\sum_t I_t^+)^2 \leq L(\sum_t I_t^+)$. Thus, we have

$$\mathbb{E}\left[\left(\sum_{t=1}^{L} I_t^+\right)^2\right] \leq L\mathbb{E}\left[\sum_{t=1}^{L} I_t^+\right] = L^2 p.$$

On $\{I_t^+ = 1\}$, we have $r_t \geq 1 + \varepsilon$ and $\phi(r_t) \geq \phi(1 + \varepsilon)$. This implies $I_t^+ \leq \frac{\phi(r_t)}{\phi(1+\varepsilon)}$ and

$$\sum_{t=1}^{L} I_t^+ \leq \frac{1}{\phi(1+\varepsilon)}\left(\sum_{t=1}^{L} \phi(r_t)I_t^+\right) \leq \frac{L\phi(R_\eta^{\max})}{\phi(1+\varepsilon)},$$

Putting these pieces together yields

$$\mathbb{E}\left[\left(\sum_{t=1}^{L} I_t^+\right)^2\right] \leq L^2 \min\left\{p, \left(\frac{\phi(R_\eta^{\max})}{\phi(1+\varepsilon)}\right)^2\right\}. \tag{14}$$

Plugging Eq. (14) into Eq. (13) gives

$$\sum_{s \neq t} \mathbb{E}[|Z_s Z_t|] \leq M^2 (\Delta_\eta^+)^2 L^2 \min\left\{p, \left(\frac{\phi(R_\eta^{\max})}{\phi(1+\varepsilon)}\right)^2\right\}. \tag{15}$$

**Conclusion.** Using $\mathbb{E}[|C_{\text{tot}}^+|] \leq \sqrt{\mathbb{E}[|C_{\text{tot}}^+|^2]}$ and $\sqrt{x + y} \leq \sqrt{x} + \sqrt{y}$ for $x, y \geq 0$ together with Eq. (11), Eq. (12), and Eq. (15) yields

$$\mathbb{E}[|C_{\text{tot}}^+|] \leq M\sqrt{2p_+ L R_\eta^{\max}\phi(R_\eta^{\max})} + ML\Delta_\eta^+ \min\left\{\sqrt{p_+}, \frac{\phi(R_\eta^{\max})}{\phi(1+\varepsilon)}\right\}, \tag{16}$$

Let $u = \eta/\pi_{\min} \leq 1$, so $R_\eta^{\max} = e^u \leq e$. For $u \in [0, 1]$, we have $e^u - 1 \leq (e - 1)u$ and $\phi(e^u) \leq u^2$. Thus, we have $R_\eta^{\max}\phi(R_\eta^{\max}) \leq eu^2$ and $\Delta_\eta^+ \leq e^u - 1 \leq (e - 1)u$. This together with Eq. (16) yields $\mathbb{E}[|C_{\text{tot}}^+|] \leq c_1\eta\sqrt{L} + \min\{c_2\eta\sqrt{p}L, c_3\eta^3 L\}$ where $c_1 = M\sqrt{2e}\pi_{\min}^{-1}$, $c_2 = M(e - 1)\pi_{\min}^{-1}$, and $c_3 = M(e - 1)\phi(1 + \epsilon)^{-1}\pi_{\min}^{-3}$. This completes the proof. $\qquad \square$

**Proof of Theorem 3.4.** We recall that $|\tilde{A}| \leq \frac{1}{2\pi_{\min}} =: M$. By Theorem 2.4, $A_i$ is symmetric, and thus $\tilde{A}(\mathbf{h}, \cdot)$ is also symmetric. Thus, we have $r_t(\eta) \stackrel{d}{=} r_t(-\eta)$ and

$$\mathbb{E}\left[|A| \sum_{t=1}^{L} r_t(\eta)\right] = \mathbb{E}\left[|A| \sum_{t=1}^{L} r_t(-\eta)\right].$$

This implies

$$\mathbb{E}[|N_{\text{raw}}|] = \mathbb{E}\left[|A| \left(\sum_{t=1}^{L} \frac{r_t(\eta)+r_t(-\eta)}{2}\right)\right].$$

We write $r_\eta := r_\eta(\mathbf{h}, a) = \frac{e^{\eta\tilde{A}(\mathbf{h},a)}}{Z_{\mathbf{h}}(\eta)}$ where $Z_{\mathbf{h}}(\eta) = \mathbb{E}_{a\sim\pi_{\text{old}}(\cdot|\mathbf{h})}[\exp(\eta\tilde{A}(\mathbf{h},a))]$. Then, we have

$$\frac{r_\eta+r_{-\eta}}{2} \geq \sqrt{r_\eta r_{-\eta}} = \frac{1}{\sqrt{Z_{\mathbf{h}}(\eta)Z_{\mathbf{h}}(-\eta)}}.$$

By using the convexity of $x \mapsto e^{\eta x}$ on $[-\widetilde{M}, \widetilde{M}]$, we have

$$e^{\eta x} \leq \frac{M+x}{2M}e^{\eta M} + \frac{M-x}{2M}e^{-\eta M} = \cosh(\eta M) + \frac{x}{M}\sinh(\eta M), \quad \text{for all } x \in [-M, M].$$

Averaging under $\pi_{\text{old}}(\cdot \mid \mathbf{h})$ yields

$$Z_{\mathbf{h}}(\eta) \leq \cosh(\eta M) + \frac{\mu(\mathbf{h})}{M}\sinh(\eta M), \quad Z_{\mathbf{h}}(-\eta) \leq \cosh(\eta M) - \frac{\mu(\mathbf{h})}{M}\sinh(\eta M),$$

where $\mu(\mathbf{h}) := \mathbb{E}_{a\sim\pi_{\text{old}}(\cdot|\mathbf{h})}[\tilde{A}(\mathbf{h}, a))]$. Then, we have

$$Z_{\mathbf{h}}(\eta)Z_{\mathbf{h}}(-\eta) \leq \cosh^2(\eta M) - \left(\frac{\mu(\mathbf{h})}{M}\right)^2 \sinh^2(\eta M) \leq \cosh^2(\eta M).$$

Putting these pieces together yields

$$\frac{r_\eta+r_{-\eta}}{2} \geq \frac{1}{\cosh(\eta M)} = \frac{1}{\cosh(\eta/(2\pi_{\min}))}.$$

Applying this to $(\mathbf{h}_t, \mathbf{y}_t)$, using $\cosh(x) \leq \exp(x^2/2)$, and summing over $t$ yields

$$\sum_{t=1}^{L} \frac{r_t(\eta)+r_t(-\eta)}{2} \geq \frac{L}{\cosh(\eta/(2\pi_{\min}))} \geq Le^{-\eta^2/(8\pi_{\min}^2)} = Le^{-C\eta^2}.$$

Taking the expectations yields Eq. (7). This together with Theorem 3.2 yields the desired bound. □

## C.3 MISSING PROOFS IN §4

We first summarize the setup and notations used in this section. We only consider $L = 1$ (bandit case) for illustration. Denote $\pi_{\text{new}}^u$ as the new policy obtained from Eq. (2) (unclipped case) and the new policy obtained from

$$\pi_{\text{new}}^c := \max_{\pi\in\Delta_{|\mathcal{V}|}} \left\{F(\pi) := \hat{J}(\pi) - \frac{1}{\eta}D_{\text{KL}}(\pi\|\pi_{\text{old}})\right\},$$

where we only consider upper clipping in the surrogate function as follows,

$$\hat{J}(\pi) := \frac{1}{G} \sum_{i=1}^{G} \min\{r(\mathbf{y}^{(i)})A_i, \min\{r(\mathbf{y}^{(i)}, 1+\varepsilon\}A_i\}.$$

We define $r_u(a) := \frac{\pi_{\text{new}}^u(a)}{\pi_{\text{old}}(a)}$, $r_c(a) := \frac{\pi_{\text{new}}^c(a)}{\pi_{\text{old}}(a)}$, and

$$S_+(a) := \sum_{i=1}^{G} A_i\mathbf{1}\{\mathbf{y}^{(i)} = a, A_i > 0\}, \quad S_-(a) := \sum_{i=1}^{G} A_i\mathbf{1}\{\mathbf{y}^{(i)} = a, A_i < 0\}.$$

**Lemma C.1.** *Let $\mathcal{V}$ be the vocabulary and $\tilde{A}$ be defined in Eq. (3) with $L = 1$. Then for $\eta > 0$,*

$$\mathbb{E}[\mathcal{H}(\pi_\eta) - \mathcal{H}(\pi_{\text{old}})] = -\frac{\eta^2}{2}\mathbb{E}[\text{Var}_{\pi_{\text{old}}}(\tilde{A}) + \text{Cov}_{\pi_{\text{old}}}(\tilde{A}^2, \log\pi_{\text{old}})] + \mathbb{E}[R(\eta)], \quad (17)$$

*Whenever there exists $\hat{\pi} \in (0, 1]$ such that $|\tilde{A}(a)| \leq \frac{1}{2\hat{\pi}}$ for all $a \in \mathcal{V}$, the remainder satisfies the explicit bound*

$$|\mathbb{E}[R(\eta)]| \leq \frac{e^{\eta/(2\hat{\pi})}}{24}\left(192 + 176\log(\frac{1}{\pi_{\min}}) + \frac{176\eta}{\hat{\pi}}\right)\eta^4\mathbb{E}\left[\mathbb{E}_{\pi_{\text{old}}}[\tilde{A}^4]\right]. \quad (18)$$

*Proof.* Write $L(a) := \log \pi_{\text{old}}(a)$ and $\psi(\eta) := \log Z(\eta)$. Then $\log \pi_\eta(a) = L(a) + t\tilde{A}(a) - \psi(\eta)$ and hence

$$\mathcal{H}(\pi_\eta) = -\mathbb{E}_{\pi_\eta}[\log \pi_\eta] = -\mathbb{E}_{\pi_\eta}[L] - \eta\mathbb{E}_{\pi_\eta}[\tilde{A}] + \psi(\eta). \tag{19}$$

Differentiate $\pi_\eta(a) = \pi_{\text{old}}(a)e^{\eta\tilde{A}(a)}/Z(\eta)$:

$$\frac{\mathrm{d}}{\mathrm{d}\eta}\pi_\eta(a) = \pi_\eta(a)(\tilde{A}(a) - \psi'(\eta)), \quad \psi'(\eta) = \mathbb{E}_{\pi_\eta}[\tilde{A}].$$

Thus, for any $g : \mathcal{V} \to \mathbb{R}$,

$$\frac{\mathrm{d}}{\mathrm{d}\eta}\mathbb{E}_{\pi_\eta}[g] = \sum_a g(a)\frac{\mathrm{d}}{\mathrm{d}\eta}\pi_\eta(a) = \mathrm{Cov}_{\pi_\eta}(g, \tilde{A}). \tag{20}$$

From Eq. (19) and $\psi'(\eta) = \mathbb{E}_{\pi_\eta}[\tilde{A}]$, we can rewrite

$$\mathcal{H}(\pi_\eta) = -\mathbb{E}_{\pi_\eta}[L] - \eta\psi'(\eta) + \psi(\eta).$$

Differentiate once and use Eq. (20) and $\psi''(\eta) = \mathrm{Var}_{\pi_\eta}(\tilde{A})$:

$$\mathcal{H}'(\eta) = -\mathrm{Cov}_{\pi_\eta}(L, \tilde{A}) - \eta\mathrm{Var}_{\pi_\eta}(\tilde{A}).$$

Differentiate again and evaluate at $\eta = 0$:

$$\mathcal{H}''(0) = -\left(\frac{\mathrm{d}}{\mathrm{d}\eta}\mathrm{Cov}_{\pi_\eta}(L, \tilde{A})\right)\bigg|_{\eta=0} - \mathrm{Var}_{\pi_{\text{old}}}(\tilde{A}).$$

Now expand $\mathrm{Cov}_{\pi_\eta}(L, \tilde{A}) = \mathbb{E}_{\pi_\eta}[L\tilde{A}] - \mathbb{E}_{\pi_\eta}[L]\mathbb{E}_{\pi_\eta}[\tilde{A}]$ and differentiate each expectation using Eq. (20). A direct calculation gives

$$\frac{\mathrm{d}}{\mathrm{d}\eta}\mathrm{Cov}_{\pi_\eta}(L, \tilde{A}) = \mathrm{Cov}_{\pi_\eta}(L, \tilde{A}^2) - 2\,\mathbb{E}_{\pi_\eta}[\tilde{A}]\,\mathrm{Cov}_{\pi_\eta}(L, \tilde{A}).$$

At $\eta = 0$, since $\mathbb{E}_{\pi_{\text{old}}}[\tilde{A}] = 0$, we have

$$\left(\frac{\mathrm{d}}{\mathrm{d}\eta}\mathrm{Cov}_{\pi_\eta}(L, \tilde{A})\right)\bigg|_{\eta=0} = \mathrm{Cov}_{\pi_{\text{old}}}(L, \tilde{A}^2). \tag{21}$$

Therefore

$$\mathcal{H}''(0) = -\mathrm{Var}_{\pi_{\text{old}}}(\tilde{A}) - \mathrm{Cov}_{\pi_{\text{old}}}(\tilde{A}^2, L), \tag{22}$$

which, with expectation, is the coefficient of $\eta^2$ in Eq. (17).

Define $f(\eta) := \mathbb{E}[\mathcal{H}(\pi_\eta) - \mathcal{H}(\pi_{\text{old}})]$ where the outer $\mathbb{E}[\cdot]$ averages over the randomness of $\tilde{A}$. By symmetry, $\pi_{-\eta}(\cdot; \tilde{A}) = \pi_\eta(\cdot; -\tilde{A})$, hence $\mathcal{H}(\pi_{-\eta}; \tilde{A}) = \mathcal{H}(\pi_\eta; -\tilde{A})$ and $f(\eta) = f(-\eta)$. Therefore $f'(0) = f^{(3)}(0) = 0$, and Taylor's theorem yields that for any $\eta > 0$ there exists $\xi \in (0, \eta)$ such that

$$f(\eta) = \frac{f''(0)}{2}\eta^2 + \frac{f^{(4)}(\xi)}{24}\eta^4.$$

Combining with Eq. (22) gives Eq. (17) with $\mathbb{E}[R(\eta)] = \frac{f^{(4)}(\xi)}{24}\eta^4$ and the bound

$$\left|\mathbb{E}[R(\eta)]\right| \leq \frac{\eta^4}{24}\sup_{t\in[0,\eta]}|f^{(4)}(\eta)| \leq \frac{\eta^4}{24}\sup_{t\in[0,\eta]}\mathbb{E}\left[|\mathcal{H}^{(4)}(\eta)|\right]. \tag{23}$$

Thus, it remains to bound $|\mathcal{H}^{(4)}(\eta)|$ by $\mathbb{E}_{\pi_\eta}[\tilde{A}^4]$. Let $m_\eta := \mathbb{E}_{\pi_\eta}[\tilde{A}]$, $X_\eta := \tilde{A} - m_\eta$, and $L_\eta := L - \mathbb{E}_{\pi_\eta}[L]$. A direct four-times differentiation of Eq. (19) using Eq. (20) yields the explicit identity

$$\mathcal{H}^{(4)}(\eta) = -\mathbb{E}_{\pi_\eta}[L_\eta X_\eta^4] + 4\mathbb{E}_{\pi_\eta}[X_\eta^3]\mathbb{E}_{\pi_\eta}[L_\eta X_\eta] + 6\mathbb{E}_{\pi_\eta}[X_\eta^2]\mathbb{E}_{\pi_\eta}[L_\eta X_\eta^2]$$
$$- 3\mathbb{E}_{\pi_\eta}[X_\eta^4] + 9\left(\mathbb{E}_{\pi_\eta}[X_\eta^2]\right)^2 + \eta\left(10\,\mathbb{E}_{\pi_\eta}[X_\eta^2]\mathbb{E}_{\pi_\eta}[X_\eta^3] - \mathbb{E}_{\pi_\eta}[X_\eta^5]\right). \tag{24}$$

Now $\|L_\eta\|_\infty \leq \log(1/\pi_{\min})$ because $L(a) \in [\log\pi_{\min}, 0]$. Also, by Cauchy-Schwarz and Hölder,

$$(\mathbb{E}[X_\eta^2])^2 \leq \mathbb{E}[X_\eta^4], \quad \mathbb{E}[|X_\eta|^3]\mathbb{E}[|X_\eta|] \leq \mathbb{E}[X_\eta^4].$$

Therefore the first line of Eq. (24) is bounded by $11 \log(1/\pi_{\min})\mathbb{E}_{\pi_\eta}[X_\eta^4]$, and the first two terms of the second line by $12\mathbb{E}_{\pi_\eta}[X_\eta^4]$. For the last term, $|m_\eta| \leq \mathbb{E}[|\tilde{A}|] \leq 1/(2\hat{\pi})$ and hence $|X_\eta| \leq |\tilde{A}| + |m_\eta| \leq 1/\hat{\pi}$ pointwise. Thus

$$\mathbb{E}[|X_\eta|^3] \leq \frac{1}{\hat{\pi}}\mathbb{E}[X_\eta^2] \leq \frac{1}{\hat{\pi}}\sqrt{\mathbb{E}[X_\eta^4]}, \quad \mathbb{E}[|X_\eta|^5] \leq \frac{1}{\hat{\pi}}\mathbb{E}[X_\eta^4],$$

which gives

$$\left|10\mathbb{E}[X_\eta^2]\mathbb{E}[X_\eta^3] - \mathbb{E}[X_\eta^5]\right| \leq 10\mathbb{E}[X_\eta^2]\mathbb{E}[|X_\eta|^3] + \mathbb{E}[|X_\eta|^5] \leq \frac{11}{\hat{\pi}}\mathbb{E}[X_\eta^4].$$

Plugging these bounds into Eq. (24) yields

$$|\mathcal{H}^{(4)}(\eta)| \leq \left(12 + 11\log(\tfrac{1}{\pi_{\min}}) + \tfrac{11t}{\hat{\pi}}\right)\mathbb{E}_{\pi_\eta}[X_\eta^4]. \tag{25}$$

Finally, $(u-v)^4 \leq 8(u^4 + v^4)$ and Jensen's inequality imply $\mathbb{E}_{\pi_\eta}[X_\eta^4] \leq 16\mathbb{E}_{\pi_\eta}[\tilde{A}^4]$, hence

$$|\mathcal{H}^{(4)}(\eta)| \leq \left(192 + 176\log(\tfrac{1}{\pi_{\min}}) + \tfrac{176t}{\hat{\pi}}\right)\mathbb{E}_{\pi_\eta}[\tilde{A}^4]. \tag{26}$$

By Jensen's inequality and the fact that $\mathbb{E}_{\pi_{\text{old}}}[\tilde{A}] = 0$, we have $Z(\eta) \geq 1$. Hence for $\eta \geq 0$,

$$\mathbb{E}_{\pi_\eta}[\tilde{A}^4] = \frac{1}{Z(\eta)}\mathbb{E}_{\pi_{\text{old}}}[e^{t\tilde{A}}\tilde{A}^4] \leq \mathbb{E}_{\pi_{\text{old}}}[e^{t\tilde{A}}\tilde{A}^4] \leq e^{t/(2\hat{\pi})}\mathbb{E}_{\pi_{\text{old}}}[\tilde{A}^4],$$

Combining with Eq. (26) gives, for $t \in [0, \eta]$,

$$\mathbb{E}[|\mathcal{H}^{(4)}(\eta)|] \leq e^{t/(2\hat{\pi})}\left(192 + 176\log(\tfrac{1}{\pi_{\min}}) + \tfrac{176t}{\hat{\pi}}\right)\mathbb{E}\left[\mathbb{E}_{\pi_{\text{old}}}[\tilde{A}^4]\right].$$

Taking the supremum over $t \in [0, \eta]$ yields

$$\sup_{t \in [0,\eta]}\mathbb{E}[|\mathcal{H}^{(4)}(\eta)|] \leq e^{\eta/(2\hat{\pi})}\left(192 + 176\log(\tfrac{1}{\pi_{\min}}) + \tfrac{176\eta}{\hat{\pi}}\right)\mathbb{E}\left[\mathbb{E}_{\pi_{\text{old}}}[\tilde{A}^4]\right].$$

Plug this into Eq. (23) to obtain Eq. (18). □

**Corollary C.2.** *Under the same setting as in Theorem C.1, if we know that conditioned on an event $B$, $|\tilde{A}(a)| \leq \frac{1}{2\hat{\pi}}$, then we have a refined bound*

$$|\mathbb{E}[R(\eta)]| \leq \mathbb{P}(B)C(\hat{\pi}) + (1 - \mathbb{P}(B))C(\pi_{\min}), \tag{27}$$

*where*

$$C(\pi) := \frac{e^{\eta/(2\pi)}}{24}\left(192 + 176\log(\tfrac{1}{\pi_{\min}}) + \tfrac{176\eta}{\pi}\right)\mathbb{E}\left[\mathbb{E}_{\pi_{\text{old}}}[\tilde{A}^4]\right]\eta^4.$$

*Proof.* Let $q := \mathbb{P}(B)$. By the tower property,

$$\mathbb{E}[R(\eta)] = q\mathbb{E}[R(\eta) \mid B] + (1-q)\mathbb{E}[R(\eta) \mid B^c],$$

hence by the triangle inequality,

$$|\mathbb{E}[R(\eta)]| \leq q|\mathbb{E}[R(\eta) \mid B]| + (1-q)|\mathbb{E}[R(\eta) \mid B^c]|.$$

Conditioned on $B$ we have the envelope $|\tilde{A}(a)| \leq \frac{1}{2\hat{\pi}}$ for all $a$, so applying the proof of Theorem C.1 to the conditional probability space yields $|\mathbb{E}[R(\eta) \mid B]| \leq C(\hat{\pi})$. On the other hand, $\pi_{\text{old}}(a) \geq \pi_{\min}$ and $|\tilde{A}(a)| \leq \frac{1}{2\pi_{\text{old}}(a)} \leq \frac{1}{2\pi_{\min}}$ for all $a$, so Theorem C.1 also gives $|\mathbb{E}[R(\eta) \mid B^c]| \leq C(\pi_{\min})$. Substituting these bounds back proves Eq. (27). □

**Lemma C.3.** *Let $\mathcal{V}$ be the vocabulary and $\tilde{A}$ be defined in Eq. (3) with $L = 1$, then*

$$\mathbb{E}[\text{Var}_{\pi_{\text{old}}}(\tilde{A})] = \frac{1 - 2^{1-G}}{G}(|\mathcal{V}| - 1),$$

$$\mathbb{E}[\text{Cov}_{\pi_{\text{old}}}(\tilde{A}^2, \log(\pi_{\text{old}}))] = \frac{1 - 2^{1-G}}{G}\left(\sum_a L(a) - |\mathcal{V}|\mathbb{E}_{\pi_{\text{old}}}[L]\right)$$

$$\mathbb{E}\left[\mathbb{E}_{\pi_{\text{old}}}[\tilde{A}^4]\right] \leq \frac{\mathbb{E}[A_1^4]}{G^3}(S_2 - 7S_1 + 12|\mathcal{V}| - 6) + \frac{3(\mathbb{E}[A_1^4] + (G-1)\mathbb{E}[A_1^2 A_2^2])}{G^3}(S_1 - 2|\mathcal{V}| + 1),$$

*where $S_1, S_2 > 0$ are*

$$S_1 = \frac{|\mathcal{V}| - 1}{\pi_{\min}} + \frac{1}{1 - (|\mathcal{V}| - 1)\pi_{\min}}, \quad S_2 = \frac{|\mathcal{V}| - 1}{(\pi_{\min})^2} + \frac{1}{(1 - (|\mathcal{V}| - 1)\pi_{\min})^2}.$$

*Proof.* We first compute $\mathbb{E}[\text{Var}_{\pi_{\text{old}}}(\tilde{A})]$. Indeed, we have

$$
\begin{aligned}
\text{Var}_{\pi_{\text{old}}}(\tilde{A}) &= \sum_a \pi_{\text{old}}(a)(\tilde{A}(a))^2 - \left(\sum_a \pi_{\text{old}}(a)\tilde{A}(a)\right)^2 \\
&= \sum_a \pi_{\text{old}}(a)\left(\frac{1}{\pi_{\text{old}}(a)G}\sum_{i=1}^{G} A_i \mathbf{1}_{\{\mathbf{y}^{(i)}=a\}}\right)^2 - \underbrace{\left(\sum_a \frac{\pi_{\text{old}}(a)}{\pi_{\text{old}}(a)G}\sum_{i=1}^{G} A_i \mathbf{1}_{\{\mathbf{y}^{(i)}=a\}}\right)^2}_{=0} \\
&= \frac{1}{G^2}\left(\sum_{i,j} A_i A_j \left(\sum_a \frac{1}{\pi_{\text{old}}(a)}\mathbf{1}_{\{\mathbf{y}^{(i)}=a\}}\mathbf{1}_{\{\mathbf{y}^{(j)}=a\}}\right)\right).
\end{aligned}
$$

Using the fact that $A_i$ is independent of $\mathbf{y}^{(i)}$, we have

$$
\begin{aligned}
\mathbb{E}[\text{Var}_{\pi_{\text{old}}}(\tilde{A})] &= \frac{1}{G^2}\sum_{i=1}^{G}\mathbb{E}[A_i^2]\left(\mathbb{E}\left[\sum_a \frac{1}{\pi_{\text{old}}(a)}\mathbf{1}_{\{\mathbf{y}^{(i)}=a\}}\right]\right) + \frac{1}{G^2}\sum_{i\neq j}\mathbb{E}[A_i A_j]\left(\mathbb{E}\left[\sum_a \frac{1}{\pi_{\text{old}}(a)}\mathbf{1}_{\{\mathbf{y}^{(i)}=\mathbf{y}^{(j)}=a\}}\right]\right) \\
&= \frac{1}{G^2}\left(|\mathcal{V}|\sum_{i=1}^{G}\mathbb{E}[A_i^2] + \sum_{i\neq j}\mathbb{E}[A_i A_j]\right).
\end{aligned}
$$

Since $\sum_{i=1}^{G} A_i = 0$, we have $\sum_{i=1}^{G} A_i^2 = -\sum_{i\neq j} A_i A_j$. In addition, by Theorem 2.4, we have $\mathbb{E}[A_i^2] = 1 - 2^{1-G}$. Thus, we have

$$
\mathbb{E}[\text{Var}_{\pi_{\text{old}}}(\tilde{A})] = \frac{1-2^{1-G}}{G}(|\mathcal{V}|-1).
$$

We then compute $\mathbb{E}[\text{Cov}_{\pi_{\text{old}}}(\tilde{A}^2, \log(\pi_{\text{old}}))]$. Indeed, we have

$$
\text{Cov}_{\pi_{\text{old}}}(\tilde{A}^2, \log(\pi_{\text{old}})) = \sum_a \pi_{\text{old}}(a)\log(\pi_{\text{old}}(a))\tilde{A}(a)^2 - \text{Var}_{\pi_{\text{old}}}(\tilde{A})\left(\sum_a \pi_{\text{old}}(a)\log\pi_{\text{old}}(a)\right).
$$

Then, we have

$$
\mathbb{E}[\text{Cov}_{\pi_{\text{old}}}(\tilde{A}^2, \log(\pi_{\text{old}}))] = \frac{1-2^{1-G}}{G}\left(\sum_a \log(\pi_{\text{old}}(a)) - |\mathcal{V}|\sum_a \pi_{\text{old}}(a)\log(\pi_{\text{old}}(a))\right).
$$

Finally, we compute $\mathbb{E}[\mathbb{E}_{\pi_{\text{old}}}[\tilde{A}^4]]$. We define $I_i(a) = \mathbf{1}\{\mathbf{y}^{(i)} = a\}$. Thus, we have $I_i(a) \sim$ Bernoulli$(p)$ i.i.d. and it is independent of $\{A_i\}_{i=1}^{G}$. We write $S(a) = \sum_{i=1}^{G} A_i I_i(a)$ and

$$
\sum_a \pi_{\text{old}}(a)(\tilde{A}(a))^4 = \sum_a \pi_{\text{old}}(a)\left(\frac{1}{\pi_{\text{old}}(a)G}\sum_{i=1}^{G} A_i \mathbf{1}\{\mathbf{y}^{(i)}=a\}\right)^4 = \sum_a \frac{(S(a))^4}{(\pi_{\text{old}}(a))^3 G^4}.
$$

Conditioning on $\{A_i\}_{i=1}^{G}$ and using $\sum_{i=1}^{G} A_i = 0$, a direct fourth-moment expansion gives

$$
\begin{aligned}
\mathbb{E}_{\pi_{\text{old}}}\left[(S(a))^4 \mid \{A_i\}_{i=1}^{G}\right] &= (\pi_{\text{old}}(a) - 7(\pi_{\text{old}}(a))^2 + 24(\pi_{\text{old}}(a))^3 - 6(\pi_{\text{old}}(a))^4)\left(\sum_{i=1}^{G} A_i^4\right) \\
&\quad + (3(\pi_{\text{old}}(a))^2 - 12(\pi_{\text{old}}(a))^3 + 3(\pi_{\text{old}}(a))^4)\left(\sum_{i=1}^{G} A_i^2\right)^2.
\end{aligned}
$$

This implies

$$
\begin{aligned}
\mathbb{E}_{\pi_{\text{old}}}\left[\mathbb{E}_{\pi_{\text{old}}}[\tilde{A}^4] \mid \{A_i\}_{i=1}^{G}\right] &= \frac{1}{G^4}\left(\left(\sum_a \frac{1}{(\pi_{\text{old}}(a))^2}\right) - 7\left(\sum_a \frac{1}{\pi_{\text{old}}(a)}\right) + 24|\mathcal{V}| - 6\right)\left(\sum_{i=1}^{G} A_i^4\right) \\
&\quad + \frac{1}{G^4}\left(3\left(\sum_a \frac{1}{\pi_{\text{old}}(a)}\right) - 12|\mathcal{V}| + 3\right)\left(\sum_{i=1}^{G} A_i^2\right)^2.
\end{aligned}
$$

In addition, we have

$$\mathbb{E}\left[\sum_{i=1}^{G} A_i^4\right] = G\mathbb{E}[A_1^4], \quad \mathbb{E}\left[\left(\sum_{i=1}^{G} A_i^2\right)^2\right] = G\mathbb{E}[A_1^4] + G(G-1)\mathbb{E}[A_1^2 A_2^2],$$

and

$$\sum_a \frac{1}{\pi_{\text{old}}(a)} \leq \frac{|\mathcal{V}|-1}{\pi_{\min}} + \frac{1}{1-(|\mathcal{V}|-1)\pi_{\min}}, \quad \sum_a \frac{1}{(\pi_{\text{old}}(a))^2} \leq \frac{|\mathcal{V}|-1}{\pi_{\min}^2} + \frac{1}{[1-(|\mathcal{V}|-1)\pi_{\min}]^2}.$$

Putting these pieces together yields

$$\mathbb{E}\left[\mathbb{E}_{\pi_{\text{old}}}[\tilde{A}^4]\right] \leq \frac{\mathbb{E}[A_1^4]}{G^3}(S_2 - 7S_1 + 12|\mathcal{V}| - 6) + \frac{3(\mathbb{E}[A_1^4]+(G-1)\mathbb{E}[A_1^2 A_2^2])}{G^3}(S_1 - 2|\mathcal{V}| + 1),$$

where $S_1, S_2 > 0$ are

$$S_1 = \frac{|\mathcal{V}|-1}{\pi_{\min}} + \frac{1}{1-(|\mathcal{V}|-1)\pi_{\min}}, \quad S_2 = \frac{|\mathcal{V}|-1}{(\pi_{\min})^2} + \frac{1}{(1-(|\mathcal{V}|-1)\pi_{\min})^2}.$$

$\square$

In what follows, we provide a detailed version of Theorem 4.1 and its proof.

**Theorem 4.1 (restated).** *If $L = 1$, then without clipping, for any $\eta > 0$, we have*

$$\mathbb{E}[\mathcal{H}(\pi_{\text{new}}) - \mathcal{H}(\pi_{\text{old}})] = -c_G \Phi(\pi_{\text{old}})\eta^2 + \mathbb{E}[R(\eta)], \quad |R(\eta)| \leq C\eta^4.$$

*where $c_G := \frac{1-2^{1-G}}{2G}$ and*

$$\begin{aligned}
\Phi(\pi) &= |\mathcal{V}| - 1 + \sum_{a \in \mathcal{V}} \log(\pi(a)) - |\mathcal{V}|\left(\sum_{a \in \mathcal{V}} \pi(a)\log(\pi(a))\right), \\
C(\pi_{\min}) &= \frac{e^{\eta/\pi_{\min}}}{24}\left(192 + 176\log(\frac{1}{\pi_{\min}}) + \frac{176\eta}{\pi_{\min}}\right)M, \\
M(\pi_{\min}) &= \frac{\mathbb{E}[A_1^4]}{G^3}(S_2(\pi_{\min}) - 7S_1(\pi_{\min}) + 12|\mathcal{V}| - 6) \\
&\quad + \frac{3(\mathbb{E}[A_1^4]+(G-1)\mathbb{E}[A_1^2 A_2^2])}{G^3}(S_1(\pi_{\min}) - 2|\mathcal{V}| + 1),
\end{aligned}$$

*with*

$$\begin{aligned}
S_1(\pi_{\min}) &= \frac{|\mathcal{V}|-1}{\pi_{\min}} + \frac{1}{1-(|\mathcal{V}|-1)\pi_{\min}}, \\
S_2(\pi_{\min}) &= \frac{|\mathcal{V}|-1}{(\pi_{\min})^2} + \frac{1}{(1-(|\mathcal{V}|-1)\pi_{\min})^2}.
\end{aligned}$$

*Proof.* If $L = 1$, there is only one $\mathbf{h}$ and every rollout has the same $\mathbf{h}$, so we ignore $\mathbf{h}$ for simplicity. In this regard, we abbreviate $\pi(a \mid \mathbf{h})$ as $\pi(a)$, $\mathbf{y}_t^{(i)}$ as $\mathbf{y}^{(i)}$, and

$$\tilde{A}(a) := \frac{1}{G}\left(\sum_{i=1}^{G} \frac{\mathbf{1}\{\mathbf{y}^{(i)}=a\}}{\pi_{\text{old}}(a)} A_i\right).$$

We rewrite the update rule as follows,

$$\pi_\eta(a) := \pi_{\text{new}}(a) = \frac{\pi_{\text{old}}(a)e^{\eta\tilde{A}(a)}}{Z(\eta)}, \quad Z(\eta) = \sum_{a \in \mathcal{V}} \pi_{\text{old}}(a)e^{\eta\tilde{A}(a)}.$$

We define $\psi(\eta) := \log Z(\eta)$ and $u(a) := \eta\tilde{A}(a) - \psi(\eta)$. In Theorem 2.2, we have shown $|\tilde{A}(a)| \leq 1/(2\pi_{\min})$ for all $a$, so by taking $\hat{\pi} := \pi_{\min}$, we can apply Theorem C.1 to obtain

$$\mathbb{E}[\mathcal{H}(\pi_\eta) - \mathcal{H}(\pi_{\text{old}})] = -\frac{\eta^2}{2}\mathbb{E}[\text{Var}_{\pi_{\text{old}}}(\tilde{A}) + \text{Cov}_{\pi_{\text{old}}}(\tilde{A}^2, \log\pi_{\text{old}})] + \mathbb{E}[R(\eta)].$$

Finally, we apply Theorem C.3 and define

$$\Phi(\pi) = |\mathcal{V}| - 1 + \sum_{a \in \mathcal{V}} \log(\pi(a)) - |\mathcal{V}|\left(\sum_{a \in \mathcal{V}} \pi(a)\log(\pi(a))\right),$$

to obtain the desired result. $\square$

We provide a numerical example in Figure 9 to illustrate the above theoretical result. Building on the two-armed setting in Remark 4.2, we conduct additional numerical experiments under unclipped GRPO training. As shown in Figure 9, entropy growth occurs only when the policy is initialized in a sufficiently skewed regime. This observation underscores that injecting spurious rewards without clipping can help preserve or restore entropy in GRPO training, particularly when the policy entropy has already collapsed or degraded toward a highly skewed distribution.

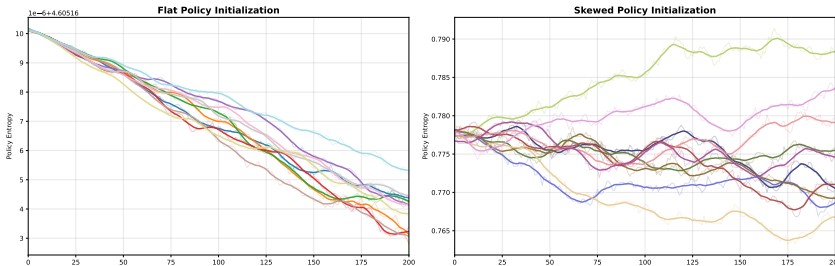

Figure 9: Simulation of policy entropy evolution over unclipped GRPO training. Each panel includes the result with 10 independent trails. Flat (relatively less-skewed) policy $\pi$ initialization (**Left**); Skewed policy $\pi$ initialization (**Right**).

Then, we proceed to the entropy dynamics with upper clipping. Before proving Theorem 4.3, we present some useful lemmas.

**Lemma C.4.** *The surrogate objective $\hat{J}(\pi)$ with only upper clipping can be rewritten as*

$$\hat{J}(\pi) = \tfrac{1}{G}\sum_{a\in\mathcal{V}}(S_-(a)r(a) + S_+(a)\min\{r(a), 1+\varepsilon\}).$$

*There exist $\lambda \in \mathbb{R}$ and $\{\mu_a\}_{a\in\mathcal{V}}$ with $\mu_a \geq 0$ and $\mu_a r_c(a) = 0$ such that,*

$$\tfrac{1}{G}(S_-(a) + S_+(a)\xi_a) - \tfrac{\pi_{\text{old}}(a)}{\eta}(\log r_c(a) + 1) - \lambda\pi_{\text{old}}(a) + \mu_a = 0, \quad \text{for every } a,$$

*where*

$$\xi_a = \begin{cases} 1, & \text{if } r_c(a) < 1+\varepsilon, \\ 0, & \text{if } r_c(a) > 1+\varepsilon, \\ [0,1], & \text{otherwise.} \end{cases}$$

*In particular, if $r_c(a) > 1+\varepsilon$, we have $\xi_a = 0$ and $\log(r_c(a)) = \frac{\eta S_-(a)}{\pi_{\text{old}}(a)G} - \eta\lambda - 1$.*

*Proof.* Since $\pi(a) = \pi_{\text{old}}(a)r(a)$, we have $\sum_a \pi_{\text{old}}(a)r(a) = 1$, $r(a) \geq 0$ and

$$D_{\text{KL}}(\pi \odot r\|\pi) = \sum_a \pi(a)r(a)\log(r(a)).$$

By definition, we have

$$\pi_{\text{new}}^c := \underset{\substack{r\geq 0 \\ \pi\odot r\in\Delta_{|\mathcal{V}|}}}{\arg\max}\left\{\tfrac{1}{G}\sum_a(S_-(a)r(a) + S_+(a)\min\{r(a), 1+\varepsilon\}) - \tfrac{1}{\eta}\sum_a \pi(a)r(a)\log(r(a))\right\}.$$

Since the objective is concave in $r$ and the constraint qualification holds, the KKT conditions are necessary and sufficient. We define $g(r) := \min\{r, 1+\varepsilon\}$ and derive its subdifferential as follows,

$$\partial g(r) = \begin{cases} \{1\}, & \text{if } r < 1+\varepsilon, \\ \{0\}, & \text{if } r > 1+\varepsilon, \\ [0,1], & \text{otherwise.} \end{cases}$$

We introduce the Lagrangian function with $\lambda$ for the equality constraint and $\mu_a \geq 0$ for $r(a) \geq 0$:

$$\mathcal{L}(r,\lambda,\mu) = \tfrac{1}{G}\sum_a(S_-(a)r(a) + S_+(a)\min\{r(a), \text{cap}\})$$

$$-\tfrac{1}{\eta}\sum_a\pi(a)r(a)\log(r(a)) - \lambda\left(\sum_a\pi(a)r(a) - 1\right) + \sum_a\mu_a r(a).$$

For $\xi_a \in \partial \min\{r(a), 1 + \varepsilon\}$, we have

$$0 \in \partial_{r(a)} \mathcal{L} = \tfrac{1}{G}(S_-(a) + S_+(a)\xi_a) - \tfrac{\pi(a)}{\eta}(\log(r(a)) + 1) - \lambda\pi(a) + \mu_a.$$

We also have $\mu_a r(a) = 0$. This implies $\mu_a = 0$ for any $a$ with $r_c(a) > 0$ and

$$\tfrac{\pi(a)}{\eta}(\log(r_c(a)) + 1) = \tfrac{1}{G}(S_-(a) + S_+(a)\xi_a) - \lambda\pi(a).$$

If $r_c(a) > 1 + \varepsilon$, we have $\xi_a = 0$. Putting these pieces together yields the desired result. $\quad\square$

**Lemma C.5.** *For any group of samples $\{\mathbf{y}^{(i)}\}_{i=1}^G$, we define $U := \mathcal{V} \setminus \{\mathbf{y}^{(1)}, \ldots, \mathbf{y}^{(G)}\}$. Then, we have*

$$\pi_{\mathrm{old}}(U) = \sum_{a \in U} \pi_{\mathrm{old}}(a) \geq (|\mathcal{V}| - G)\pi_{\min} =: M_0.$$

*Suppose that*

$$\tfrac{(1+\varepsilon)\eta}{2} < \left(M_0 - \tfrac{1}{2}\sqrt{\eta(1+\varepsilon)}\right)\log(1 + \varepsilon). \tag{28}$$

*Then, we have $r_c(a) \leq 1 + \varepsilon$ for all $a \in \mathcal{V}$.*

*Proof.* First, we show $D_{\mathrm{KL}}(\pi_{\mathrm{new}}^c \| \pi_{\mathrm{old}}) \leq \tfrac{1}{2}\eta(1 + \varepsilon)$. By definition, we have $F(\pi_{\mathrm{new}}^c) \geq F(\pi_{\mathrm{old}})$. Since $r \equiv 1$ and $\hat{J}(\pi_{\mathrm{old}}) = \sum_i A_i = 0$, we have $F(\pi_{\mathrm{old}}) = 0$. Thus, we obtain from $F(\pi_{\mathrm{new}}^c) \geq 0$ that $D_{\mathrm{KL}}(\pi_{\mathrm{new}}^c \| \pi_{\mathrm{old}}) \leq \eta\hat{J}(\pi_{\mathrm{new}}^c)$. Due to the upper clipping, given the number of samples with reward $+1$ in the group $1 \leq K \leq G - 1$, we have

$$\hat{J}(\pi_{\mathrm{new}}^c) \leq (1 + \varepsilon)\sum_{i=1}^G A_i \mathbf{1}\{A_i > 0\} = (1 + \varepsilon)\tfrac{K}{G}\sqrt{\tfrac{1 - K/G}{K/G}} = (1 + \varepsilon)\sqrt{\tfrac{K}{G}\left(1 - \tfrac{K}{G}\right)} \leq \tfrac{1+\varepsilon}{2}.$$

Putting these pieces together yields the desired result.

Then, we show $\pi_{\mathrm{new}}^c(U) \geq M_0 - \tfrac{\sqrt{\eta(1+\varepsilon)}}{2}$. Indeed, this can be derived from the Pinsker's inequality as follows,

$$|\pi_{\mathrm{new}}^c(U) - \pi_{\mathrm{old}}(U)| \leq \|\pi_{\mathrm{new}}^c - \pi_{\mathrm{old}}\|_{\mathrm{TV}} \leq \sqrt{\tfrac{1}{2}D_{\mathrm{KL}}(\pi_{\mathrm{new}}^c \| \pi_{\mathrm{old}})} = \tfrac{1}{2}\sqrt{\eta(1 + \varepsilon)}.$$

This together with $\pi_{\mathrm{old}}(U) \geq M_0$ yields the desired result.

Finally, we show that $r_c(a) \leq 1 + \varepsilon$ for all $a \in \mathcal{V}$ given Eq. (28). Suppose that there are some $a^\star$ so that $r_c(a^\star) > 1 + \varepsilon$. Then, by Theorem C.4, we have $\xi_{a^\star} = 0$ and

$$\log(r_c(a^\star)) = \tfrac{\eta S_-(a^\star)}{\pi_{\mathrm{old}}(a^\star)G} - \eta\lambda - 1 \overset{S_-(a^\star) \leq 0}{\leq} -\eta\lambda - 1,$$

By definition, $S_+(b) = S_-(b) = 0$ for all $b \in U$. Thus, we have $\log(r_c(b)) = -\eta\lambda - 1$ and $r_c(b) \geq r_c(a^\star) > 1 + \varepsilon$ for all $b \in U$. This implies

$$D_{\mathrm{KL}}(\pi_{\mathrm{new}}^c \| \pi_{\mathrm{old}}) = \sum_a \pi_{\mathrm{new}}^c(a)\log(r_c(a)) > \log(1 + \varepsilon)\pi_{\mathrm{new}}^c(U).$$

Putting these pieces together yields a direct contradiction to Eq. (28) as follows,

$$\tfrac{1}{2}\eta(1 + \varepsilon) \geq D_{\mathrm{KL}}(\pi_{\mathrm{new}}^c \| \pi_{\mathrm{old}}) > \log(1 + \varepsilon)\left(M_0 - \tfrac{\sqrt{(1+\varepsilon)\eta}}{2}\right).$$

This completes the proof. $\quad\square$

**Theorem 4.3 (restated).** *Define $C_i := \{A_i > 0, r_u(\mathbf{y}^{(i)}) > 1 + \varepsilon\}$. Let $\rho := \mathbb{P}(C_1)$ and $\delta = \mathbb{E}[r_u(\mathbf{y}^{(1)}) - (1 + \varepsilon) \mid C_1]$. Then, for $\eta > 0$ satisfying Eq. (28) and any $p \in (\pi_{\min}, 1)$, we have*

$$\mathbb{E}[\mathcal{H}(\pi_{\mathrm{new}}^c) - \mathcal{H}(\pi_{\mathrm{old}})] \leq -c_G \Phi(\pi_{\mathrm{old}})\eta^2 + \mathbb{E}[R(\eta)] + c(p)G\left(\rho\delta_{\mathrm{eff}} - \tfrac{X_{\max}}{2}(G - 1)p\right),$$

*where $c_G$, $\Phi$ and $R(\eta)$ are defined as the same as in Theorem 4.1, and*

$$c(p) := -\pi_{\min}\left(\log pe^{\eta/(2\pi_{\min})}\right)_-, \quad \delta_{\mathrm{eff}} := \tfrac{X_{\max}(\delta - M(p))_+}{X_{\max} - M(p)},$$
$$X_{\max} := \exp(\eta/(2\pi_{\min})) - (1 + \varepsilon), \quad M(p) := [\exp(\eta/(2p)) - (1 + \varepsilon)]_+.$$

*Proof.* For simplicity, define $\Delta\mathcal{H}^c := \mathcal{H}(\pi_{\text{new}}^c) - \mathcal{H}(\pi_{\text{old}})$ and $\Delta\mathcal{H}^u := \mathcal{H}(\pi_{\text{new}}^u) - \mathcal{H}(\pi_{\text{old}})$. Then, we have

$$\mathbb{E}[\Delta\mathcal{H}^c] = \mathbb{E}[\mathcal{H}(\pi_{\text{new}}^c) - \mathcal{H}(\pi_{\text{new}}^u)] + \mathbb{E}[\Delta\mathcal{H}^u].$$

By first part of Theorem C.1 and Theorem C.3, we have

$$\mathbb{E}[\Delta\mathcal{H}^u] = -\frac{1-2^{1-G}}{2G}\Phi(\pi_{\text{old}})\eta^2 + \mathbb{E}[R(\eta)]. \tag{29}$$

It suffices to consider $\mathbb{E}[\mathcal{H}(\pi_{\text{new}}^c) - \mathcal{H}(\pi_{\text{new}}^u)]$. Indeed, we consider

$$\mathcal{H}(\pi_{\text{new}}^c) = -\sum_a \pi_{\text{new}}^c(a)\log(\pi_{\text{new}}^c(a)) = -\sum_a \pi_{\text{new}}^c(a)\log(\pi_{\text{new}}^u(a)) - D_{\text{KL}}(\pi_{\text{new}}^c\|\pi_{\text{new}}^u).$$

Since $D_{\text{KL}}(\pi_{\text{new}}^c\|\pi_{\text{new}}^u) \geq 0$, we have

$$\begin{aligned}
\mathcal{H}(\pi_{\text{new}}^c) - \mathcal{H}(\pi_{\text{new}}^u) &\leq& -\sum_a \pi_{\text{new}}^c(a)\log(\pi_{\text{new}}^u(a)) + \sum_a \pi_{\text{new}}^u(a)\log(\pi_{\text{new}}^u(a)) \\
&\leq& \sum_{a\in\mathcal{V}} (\pi_{\text{new}}^u(a) - \pi_{\text{new}}^c(a))\log(\pi_{\text{new}}^u(a)).
\end{aligned}$$

By Theorem C.5, we have $\pi_{\text{new}}^c(a) \leq (1+\varepsilon)\pi_{\text{old}}(a)$ for every $a$. If $r_u(a) > 1 + \varepsilon$, we have

$$\begin{aligned}
\pi_{\text{new}}^u(a) - \pi_{\text{new}}^c(a) &=& \pi_{\text{old}}(a)r_u(a) - \pi_{\text{new}}^c(a) \geq \pi_{\text{old}}(a)(r_u(a) - (1+\varepsilon)) \\
&\geq& \pi_{\min}(r_u(a) - (1+\varepsilon)).
\end{aligned}$$

Thus, we have

$$\mathcal{H}(\pi_{\text{new}}^c) - \mathcal{H}(\pi_{\text{new}}^u) \leq \sum_{a\in\mathcal{V}} \pi_{\min}(r_u(a) - (1+\varepsilon))\log(\pi_{\text{new}}^u(a)). \tag{30}$$

For any $p \in (0, \frac{1}{|\mathcal{V}|}]$, on the set $\{\pi_{\text{old}}(\mathbf{y}^{(i)}) \leq p\}$, we have

$$\log(\pi_{\text{new}}^u(\mathbf{y}^{(i)})) = \log(\pi_{\text{old}}(\mathbf{y}^{(i)})r_u(\mathbf{y}^{(i)})) \leq \min\{0, \log(pe^{\eta/(2\pi_{\min})})\}.$$

Using Eq. (30), restricting to indices $i$ where $C_i$ occurs, we have

$$\begin{aligned}
\mathcal{H}(\pi_{\text{new}}^c) - \mathcal{H}(\pi_{\text{new}}^u) &\leq& \min\{0, \log(pe^{\eta/(2\pi_{\min})})\} \sum_{a:r_u(a)>1+\varepsilon} \pi_{\min}(r_u(a) - (1+\varepsilon)) \\
&\leq& \min\{0, \log(pe^{\eta/(2\pi_{\min})})\} \sum_{a:r_u(a)>1+\varepsilon, \pi_{\text{old}}(a)\leq p} \pi_{\min}(r_u(a) - (1+\varepsilon)).
\end{aligned}$$

We let $X_i := (r_u(a) - (1+\varepsilon))\mathbf{1}_{C_i}$ and $n_a := \sum_{j=1}^G \mathbf{1}\{\mathbf{y}^{(j)} = a\}\mathbf{1}_{C_j}$. Then, we have

$$\sum_{a:r_u(a)>1+\varepsilon, \pi_{\text{old}}(a)\leq p} \pi_{\min}(r_u(a) - (1+\varepsilon)) = \pi_{\min}\left(\sum_{i=1}^G \frac{X_i\mathbf{1}\{\pi_{\text{old}}(\mathbf{y}^{(i)})\leq p\}}{n_{\mathbf{y}^{(i)}}}\right).$$

Define $K := n_{\mathbf{y}^{(i)}} - 1$. Conditioned on $\mathbf{y}^{(i)} = a$ and $\pi_{\text{old}}(a) \leq p$, we have

$$K \leq \sum_{j=1}^G \mathbf{1}\{\mathbf{y}^{(j)} = a\} - 1 \sim \text{Binomial}(G-1, \pi_{\text{old}}(a)).$$

Thus, we have

$$\begin{aligned}
\mathbb{E}[K\mathbf{1}\{\pi_{\text{old}}(\mathbf{y}^{(i)}) \leq p\}] &=& \mathbb{E}[\mathbb{E}[K \mid \pi_{\text{old}}(\mathbf{y}^{(i)}) \leq p]\mathbf{1}\{\pi_{\text{old}}(\mathbf{y}^{(i)}) \leq p\}] \\
&=& \mathbb{E}[(G-1)\pi_{\text{old}}(\mathbf{y}^{(i)})\mathbf{1}\{\pi_{\text{old}}(\mathbf{y}^{(i)}) \leq p\}] \\
&=& (G-1)\sum_a (\pi_{\text{old}}(a))^2\mathbf{1}\{\pi_{\text{old}}(a) \leq p\} \\
&\leq& (G-1)p.
\end{aligned}$$

Since for any $K \geq 0$, we always have $\frac{1}{K+1} \geq 1 - \frac{K}{2}$,

$$\frac{X_i\mathbf{1}\{\pi_{\text{old}}(\mathbf{y}^{(i)})\leq p\}}{n_{\mathbf{y}^{(i)}}} \geq X_i\mathbf{1}\{\pi_{\text{old}}(\mathbf{y}^{(i)}) \leq p\} - \frac{1}{2}X_i\mathbf{1}\{\pi_{\text{old}}(\mathbf{y}^{(i)}) \leq p\}(n_{\mathbf{y}^{(i)}} - 1).$$

Thus, we have

$$\mathbb{E}\left[\frac{X_i \mathbf{1}\{\pi_{\text{old}}(\mathbf{y}^{(i)}) \leq p\}}{n_{\mathbf{y}^{(i)}}}\right] \geq \mathbb{E}[X_i \mathbf{1}\{\pi_{\text{old}}(\mathbf{y}^{(i)}) \leq p\}] - \frac{X_{\max}}{2}(G-1)p,$$

where $X_i \leq X_{\max} := \exp(\eta/(2\pi_{\min})) - (1 + \epsilon)$. This $X_{\max}$ is derived by recalling that $|\tilde{A}(a)| \leq 1/(2\pi_{\min})$ and noticing that Jensen's inequality implies $Z(\eta) := \mathbb{E}_{\pi_{\text{old}}}[e^{\eta \tilde{A}(a)}] \geq e^{\eta \mathbb{E}_{\pi_{\text{old}}}[\tilde{A}(a)]} = 1$. Taking expectation of both sides of Eq. (30) yields

$$\mathbb{E}[\mathcal{H}(\pi_{\text{new}}^c) - \mathcal{H}(\pi_{\text{new}}^u)] \leq c(p)G\left(\mathbb{E}[X_i \mathbf{1}\{\pi_{\text{old}}(\mathbf{y}^{(i)}) \leq p\}] - \frac{X_{\max}}{2}(G-1)p\right), \qquad (31)$$

where we denote $c(p) := \min\{0, \log(pe^{\eta/(2\pi_{\min})})\}\pi_{\min}$. Finally, notice that on $\{\pi_{\text{old}}(\mathbf{y}^{(i)}) > p\}$, we have another bound $X_i \leq M(p) := [\exp(\eta/(2p)) - (1+\varepsilon)]_+$. By symmetry, for any $i = 1, \ldots, G$, we have

$$\begin{aligned}
\delta &= \mathbb{E}[r(\mathbf{y}^{(i)}) - (1 + \varepsilon) \mid C_i] = \mathbb{E}[X_i \mid C_i] \\
&= \mathbb{E}[X_i \mathbf{1}\{\pi_{\text{old}}(\mathbf{y}^{(i)}) \leq p\} \mid C_i] + \mathbb{E}[X_i \mathbf{1}\{\pi_{\text{old}}(\mathbf{y}^{(i)}) > p\} \mid C_i] \\
&\leq X_{\max}\mathbb{P}(\pi_{\text{old}}(\mathbf{y}^{(i)}) \leq p \mid C_i) + M(p)(1 - \mathbb{P}(\pi_{\text{old}}(\mathbf{y}^{(i)}) \leq p \mid C_i)).
\end{aligned}$$

This implies

$$\mathbb{P}(\pi_{\text{old}}(\mathbf{y}^{(i)}) \leq p \mid C_i) \leq \frac{X_{\max} - \delta}{X_{\max} - M(p)},$$

which further guarantees

$$\begin{aligned}
\mathbb{E}[X_i \mathbf{1}\{\pi_{\text{old}}(\mathbf{y}^{(i)}) \leq p\} \mid C_i] &= \delta - \mathbb{E}[X_i \mathbf{1}\{\pi_{\text{old}}(\mathbf{y}^{(i)}) > p\} \mid C_i] \\
&\geq \delta - M(p)\mathbb{P}(\pi_{\text{old}}(\mathbf{y}^{(i)}) \leq p \mid C_i) \\
&\geq \delta - M(p)\frac{X_{\max} - \delta}{X_{\max} - M(p)} = \frac{X_{\max}(\delta - M(p))}{X_{\max} - M(p)}.
\end{aligned}$$

Notice that $\mathbb{E}[X_i \mathbf{1}\{\pi_{\text{old}}(\mathbf{y}^{(i)}) \leq p\} \mid C_i]$ is nonnegative. Then, we have

$$\mathbb{E}[X_i \mathbf{1}\{\pi_{\text{old}}(\mathbf{y}^{(i)}) \leq p\} \mid C_i] \geq \frac{X_{\max}(\delta - M(p))_+}{X_{\max} - M(p)} := \delta_{\text{eff}}.$$

By using the symmetry again, we have

$$\mathbb{E}[X_i \mathbf{1}\{\pi_{\text{old}}(\mathbf{y}^{(i)}) \leq p\}] = \mathbb{P}(C_i)\mathbb{E}[X_i \mathbf{1}\{\pi_{\text{old}}(\mathbf{y}^{(i)}) \leq p\} \mid C_i] \geq \rho\delta_{\text{eff}}.$$

Therefore, Eq. (31) becomes

$$\mathbb{E}[\mathcal{H}(\pi_{\text{new}}^c) - \mathcal{H}(\pi_{\text{new}}^u)] \leq c(p)G\left(\rho\delta_{\text{eff}} - \frac{X_{\max}}{2}(G-1)p\right). \qquad (32)$$

Combining Eq. (29) and Eq. (32), we obtain the desired result

$$\begin{aligned}
&\mathbb{E}[\mathcal{H}(\pi_c) - \mathcal{H}(\pi_u)] \\
\leq\ & -\frac{1 - 2^{1-G}}{2G}\Phi(\pi_{\text{old}})\eta^2 + \mathbb{E}[R(\eta)] + c(p)G\left(\rho\delta_{\text{eff}} - \frac{X_{\max}}{2}(G-1)p\right) \\
=\ & -\frac{1 - 2^{1-G}}{2G}\Phi(\pi_{\text{old}})\eta^2 + \mathbb{E}[R(\eta)] - G\pi_{\min}\left(\log(pe^{\eta/(2\pi_{\min})})\right)_-\left(\rho\delta_{\text{eff}} - \frac{X_{\max}}{2}(G-1)p\right).
\end{aligned}$$
$$(33)$$

This completes the proof. $\qquad\square$

**Lemma C.6.** *Let $\mathcal{V}_{\leq \hat{\pi}} := \{a \in \mathcal{V} : \pi_{\text{old}}(a) \leq \hat{\pi}\}$ with some $\hat{\pi} \in [\pi_{\min}, 1)$. Define $\text{Coll}_{\leq \hat{\pi}} := \{\exists a \in \mathcal{V}_{\leq \hat{\pi}} : N_a \geq 2\}$ where $N_a := \sum_{i=1}^{G} \mathbf{1}\{\mathbf{y}^{(i)} = a\}$. Then $\mathbb{P}(\text{Coll}_{\leq \hat{\pi}}) \leq \binom{G}{2}|\mathcal{V}|\hat{\pi}^2$. Furthermore, if $\hat{\pi} \leq \frac{G}{2(\sqrt{G} - 1/\sqrt{G})}\pi_{\min}$, then on $\text{Coll}_{\leq \hat{\pi}}^c$, we have $|\tilde{A}(a)| \leq \frac{1}{2\hat{\pi}}$ for all $a$.*

*Proof.* Let $\mu_{\leq \hat{\pi}} := \pi_{\text{old}}(\mathcal{V}_{\leq \hat{\pi}})$. For any fixed $a \in \mathcal{V}_{\leq \hat{\pi}}$, we know $N_a \sim \text{Binomial}(G, \pi(a))$ and $\mathbb{P}(N_a \geq 2) \leq \binom{G}{2}\pi(a)^2$. Union bound over $a \in \mathcal{V}_{\leq \hat{\pi}}$ yields

$$\mathbb{P}(\text{Coll}_{\leq \hat{\pi}}) \leq \binom{G}{2}\sum_{a:\pi(a)\leq\hat{\pi}}\pi(a)^2 \leq \binom{G}{2}\hat{\pi}\sum_{a:\pi(a)\leq\hat{\pi}}\pi(a) = \binom{G}{2}\hat{\pi}\mu_{\leq\hat{\pi}},$$

and using $\mu_{\leq\hat{\pi}} \leq |\mathcal{V}|\hat{\pi}$ gives the final bound $\mathbb{P}(\text{Coll}_{\leq\hat{\pi}}) \leq \binom{G}{2}|\mathcal{V}|\hat{\pi}^2$. On $\text{Coll}_{\leq\hat{\pi}}^c$, if $\pi_{\text{old}}(a) \leq \hat{\pi}$, then the token $a$ appears at most once in the batch, hence $\tilde{A}(a) = \frac{A_i}{G\pi(a)}$ or $\tilde{A}(a) = 0$. For the former one, by Theorem 2.4, we have $|\tilde{A}(a)| \leq \frac{\sqrt{G} - 1/\sqrt{G}}{G\pi(a)} \leq \frac{\sqrt{G} - 1/\sqrt{G}}{G\pi_{\min}}$. By assumption $\hat{\pi} \leq \frac{G}{2(\sqrt{G} - 1/\sqrt{G})}\pi_{\min}$, one concludes that $|\tilde{A}(a)| \leq \frac{1}{2\hat{\pi}}$. $\qquad\square$

**Remark C.7.** *In this remark, we will show that under practical settings, the assumption, i.e., Eq. (28), in Theorem 4.3 is indeed satisfied and the extra term ensure the entropy is decreasing in expectation. Recall the parameters we used or observed in experiments:* $G = 16$, $\eta = 5 \times 10^{-7}$, $|\mathcal{V}| \approx 150000$, $\pi_{\min} = 10^{-7}$, $\epsilon = 0.2$, $\rho \approx 0.001$, $\delta \approx 10$, *and we choose the threshold* $p = \hat{\pi} = 2\pi_{\min} = 2 \times 10^{-7}$. *By Theorem 4.3, we compute* $X_{\max} = 10.98$, $M(p) = 2.29$, $\delta_{\text{eff}} = 9.74$, *and* $c(p) = -1.29 \times 10^{-6}$. *Thus, the clipping correction term (the third term) in the upper bound of* $\mathbb{E}[\Delta\mathcal{H}]$ *is* $-2.01 \times 10^{-7}$. *Furthermore,* $c_G\eta^2 = 7.81 \times 10^{-15}$ *and* $\Phi(\pi) \geq \Phi_{\min} = -2.23 \times 10^6$. *Finally, define* $B := \text{Coll}_{\leq \hat{\pi}}$ *in Theorem C.6, applying Theorem C.2, we have*

$$|\mathbb{E}[R(\eta)]| \leq (1 - q_{\hat{\pi}})C(\hat{\pi}) + q_{\hat{\pi}}C(\pi_{\min}),$$

*where* $q_{\hat{\pi}} \leq 7.2 \times 10^{-7}$ *by Theorem C.6,* $C(\hat{\pi}) = 3.43 \times 10^{-8}$, *and* $C(\pi_{\min}) = 5.31 \times 10^{-7}$. *Hence,* $|\mathbb{E}[R(\eta)]| \leq 3.43 \times 10^{-8}$. *Combining all numbers, we have*

$$\mathbb{E}[\Delta\mathcal{H}] \leq (-2.23 \times 10^6) \times (-7.81 \times 10^{-15}) + 3.43 \times 10^{-8} - 2.01 \times 10^{-7} < -1.49 \times 10^{-7} < 0.$$

*Thus, we can conclude* $\mathbb{E}[\Delta\mathcal{H}] < 0$.

## C.4 MISSING PROOFS IN §5

**Proof of Theorem 5.2.** Since $f \sim \text{Binomial}(n_i, \frac{1}{2})$ and $g \sim \text{Binomial}(n_c, \frac{1}{2})$, we have $\mathbb{E}[f] = \frac{n_i}{2}$ and $\mathbb{E}[g] = \frac{n_c}{2}$. We rewrite Eq. (8) as $\Delta = \frac{n_c(f-g)}{G} + g$ and have

$$\mathbb{E}[\Delta] = \frac{n_c(n_i - n_c)}{2G} + \frac{n_c}{2} = \frac{n_c(G - n_c)}{G}.$$

In addition, we have $\text{Var}(f) = \frac{n_i}{4}$ and $\text{Var}(g) = \frac{n_c}{4}$. Since $f$ and $g$ are independence, we have

$$\text{Var}(\Delta) = \left(\frac{n_c}{G}\right)^2 \text{Var}(f) + \left(\frac{n_i}{G}\right)^2 \text{Var}(g) = \left(\frac{n_c}{G}\right)^2 \frac{n_i}{4} + \left(\frac{n_i}{G}\right)^2 \frac{n_c}{4} = \frac{n_c(G - n_c)}{4G}.$$

This completes the proof. $\qquad\square$

**Proof of Theorem 5.3.** We define $X = f$ and $Y = n_c - g$. Then, we have $Y \sim \text{Binomial}(n_c, \frac{1}{2})$ and

$$\Delta = \frac{n_c X}{G} + \frac{n_i(n_c - Y)}{G} = \frac{n_i n_c}{G} + \frac{n_c X}{G} - \frac{n_i Y}{G}. \tag{34}$$

By definition, $Z = X + Y \sim \text{Binomial}(G, \frac{1}{2})$ and

$$f > g \iff Z > n_c, \quad f < g \iff Z < n_c.$$

By definition, we have $\Pr(\mathbf{r}_j = 1 \mid Z = z) = \frac{z}{G}$ for each $j \in \mathcal{I}$ and each $j \in \mathcal{C}$. Thus, we have

$$\mathbb{E}[X \mid Z = z] = \frac{n_c z}{G}, \quad \mathbb{E}[Y \mid Z = z] = \frac{n_i z}{G}.$$

Taking the conditional expectation of both sides of Eq. (34) yields

$$\mathbb{E}[\Delta \mid Z = z] = \frac{n_i n_c}{G} + \frac{n_c}{G}\mathbb{E}[X \mid Z = z] - \frac{n_i}{G}\mathbb{E}[Y \mid Z = z] = \frac{n_i n_c}{G}.$$

By using the tower property, we have

$$\mathbb{E}[\Delta \mid f > g] = \mathbb{E}[\Delta \mid g > f] = \frac{n_i n_c}{G}.$$

Note that $\mathbb{E}[\Delta\mathbf{1}_{\{f>g\}}] = \mathbb{E}[\Delta \mid f > g]\Pr(f > g)$ and $\mathbb{E}[\Delta\mathbf{1}_{\{g<f\}}] = \mathbb{E}[\Delta \mid g > f]\Pr(g > f)$. Thus, it suffices to prove that $\Pr(f > g) < \Pr(f < g)$. Indeed, we write them in wedge form as

$$\Pr(f > g) = \frac{1}{2^G} \sum_{k > \ell} \binom{n_i}{k}\binom{n_c}{\ell}, \quad \Pr(g > f) = \frac{1}{2^G} \sum_{k > \ell} \binom{n_c}{k}\binom{n_i}{\ell}. \tag{35}$$

Fixing the integers $k > \ell \geq 0$, we define the function as follows,

$$\Psi(n) = \frac{\binom{n}{k}}{\binom{n}{\ell}}, \text{ for all } n \geq k.$$

We claim that $\Psi(n)$ is strictly increasing in $n$. Indeed, we have

$$\Psi(n) = \frac{\ell!}{k!}\frac{(n-\ell)!}{(n-k)!} = \frac{\ell!}{k!}\left(\Pi_{j=0}^{k-\ell-1}(n - \ell - j)\right).$$

Each term in the product is strictly increasing in $n$. Thus, this yields the desired result.

Since $n_c > n_i \geq k$, we have

$$\frac{\binom{n_c}{k}}{\binom{n_c}{\ell}} = \Phi(n_c) > \Phi(n_i) = \frac{\binom{n_i}{k}}{\binom{n_i}{\ell}}.$$

This implies

$$\binom{n_c}{k}\binom{n_i}{\ell} > \binom{n_i}{k}\binom{n_c}{\ell}, \text{ for all } k > \ell.$$

This together with Eq. (35) yields the desired result. $\qquad\square$

**Conditional variance analysis.** Let $f \sim \text{Binomial}(n_i, \frac{1}{2})$ and $g \sim \text{Binomial}(n_c, \frac{1}{2})$ be independent, and let $\Delta$ be defined in Eq. (8). We write $X = f$, $Y = n_c - g$ and let $Z = X + Y \sim \text{Binomial}(G, \frac{1}{2})$. Then, we have

$$\mathbb{E}[\Delta \mid Z = z] = \frac{n_i n_c}{G}, \quad \text{Var}(\Delta \mid Z = z) = \frac{n_i(G - n_i)}{G - 1} \frac{z(G - z)}{G^2}.$$

We let $C = \frac{n_i(G - n_i)}{G^2(G - 1)}$ and define $h(z) = z(G - z)$. Then, we have

$$\text{Var}(\Delta \mid f > g) = C\mathbb{E}[h(Z) \mid Z > n_c], \quad \text{Var}(\Delta \mid g > f) = C\mathbb{E}[h(Z) \mid Z < n_c].$$

If $n_c > n_i$, we have $\text{Var}(\Delta \mid f > g) < \text{Var}(\Delta \mid g > f)$.

**Proof.** Conditional on $Z = z$, the $z$ positive labels are uniformly scattered among $G$ positions. Then, the count $X$ of positives falling inside the $n_i$ indices of $\mathcal{I}$ is

$$X \mid Z = z \sim \text{Hypergeometric}(G, z, n_i).$$

Thus, we have

$$\mathbb{E}[X \mid Z = z] = \frac{n_i z}{G}, \quad \text{Var}(X \mid Z = z) = \frac{n_i z}{G}\left(1 - \frac{z}{G}\right)\frac{G - n_i}{G - 1}.$$

By definition, we have

$$\Delta = \frac{n_c X}{G} + \frac{n_i(n_c - Y)}{G} = X - \frac{n_i Z}{G} + \frac{n_i n_c}{G}.$$

and

$$\mathbb{E}[\Delta \mid Z = z] = \mathbb{E}[X \mid Z = z] - \frac{n_i z}{G} + \frac{n_i n_c}{G} = \frac{n_i n_c}{G},$$

and

$$\text{Var}(\Delta \mid Z = z) = \text{Var}(X \mid Z = z) = \frac{n_i(G - n_i)}{G - 1}\frac{z(G - z)}{G^2}.$$

We let $C = \frac{n_i(G - n_i)}{G^2(G - 1)}$ and define $h(z) = z(G - z)$. Since $\mathbb{E}[\Delta \mid Z]$ is independent of $Z$, we have

$$\text{Var}(\Delta \mid A) = \mathbb{E}[\text{Var}(\Delta \mid Z) \mid A] = C\mathbb{E}[h(Z) \mid A], \quad \text{for an event } A \text{ measurable w.r.t. } Z.$$

Since $f > g \iff Z > n_c$ and $g > f \iff Z < n_c$, we have

$$\text{Var}(\Delta \mid f > g) = C\mathbb{E}[h(Z) \mid Z > n_c], \quad \text{Var}(\Delta \mid g > f) = C\mathbb{E}[h(Z) \mid Z < n_c].$$

Since $Z \sim \text{Binomial}(G, \frac{1}{2})$, $h(G - z) = h(z)$ and $h(z)$ is strictly increasing on $\{0, 1, 2, 3, \ldots, \lfloor \frac{G}{2} \rfloor\}$, we have

$$\mathbb{E}[h(Z) \mid Z > n_c] = \mathbb{E}[h(Z) \mid Z < G - n_c].$$

Since $n_c > \frac{G}{2}$, we have $0 \leq G - n_c < n_c \leq G$. Thus, we have

$$\mathbb{E}[h(Z) \mid Z < G - n_c] < \mathbb{E}[h(Z) \mid Z < n_c].$$

Multiplying both sides of the above inequality by $C > 0$ yields

$$\text{Var}(\Delta \mid f > g) < \text{Var}(\Delta \mid g > f).$$

This completes the proof. $\qquad \square$

