# OpenReview forum: "Exploration vs Exploitation: Rethinking RLVR through Clipping, Entropy, and Spurious Reward"
_ICLR.cc/2026/Conference — ICLR 2026 Poster_

### Official Review · Reviewer_tUmE · 2025-10-28

**Soundness:** 3
**Presentation:** 2
**Contribution:** 3
**Rating:** 6
**Confidence:** 2

**Summary:**

This paper investigates the exploration-exploitation trade-off in RLVR for LLMs. It examine the paradoxical finding that random rewards can improve LLM reasoning performance. The core contributions are a theoretical and empirical reframing of the underlying mechanisms. This paper argues that clipping bias, previously thought to be the driver of these gains, provides a negligible learning signal. Instead, it demonstrates that clipping acts as a regularizer that deterministically reduces policy entropy. Through experiments on multiple model series, it shows that these gains are not exclusive to potentially contaminated models. Finally, this paper proposes a reward-misalignment model to explain why stronger models are more likely to benefit from this training regime.

**Strengths:**

- This paper provides a  theoretical argument that refutes the prevailing view that upper-clipping bias is the primary driver of performance gains under random rewards.
- This paper establishes a relationship between the application of clipping and the reduction of policy entropy, effectively reframing clipping as an entropy-minimizing regularizer rather than a source of learning signal.
- This paper introduces a  probabilistic model  to explain which models are likely to benefit from random reward training. This framework provides an explanation for the empirical observation that stronger models tend to improve more.

**Weaknesses:**

- Several key figures presenting experimental results are difficult to interpret due to missing legends and insufficient detail in captions. Figures 1, 2, 3 (Middle), and 4 all contain plots with multiple colored lines, but none include a legend to explain what each line represents. The review is left to assume they correspond to independent training runs, but this is not stated. The captions often lack sufficient detail. For example, the caption for Figure 1 does not explain that the faint lines in the Left and Middle panels are likely individual runs.
- The paper's mathematical exposition is dense, and the notation is not always clear or consistent.  The definitions for the clipping analysis in Theorem 3.3  are imprecise. $N_t^{clip} := \bar{r}_t \hat{A}_t$ is defined, but $\bar{r}_t$ itself is not. The proof of Proposition 3.2  appears to rely on unstated assumptions. The expansion $log~Z(h) = \frac{1}{2}\eta^2 + \mathcal{O}(\eta^3)$ requires $\mathbb{E}[\hat{A}] = 0$ and $\mathbb{E}[\hat{A}^2] = 1$ .
- The paper's refutation of contamination does not address a key alternative. The low clipping activation rate ($\approx 0.001$) for Qwen2.5-Math-7B (Remark 3.5) is used to show the clipping *signal* is negligible. However, this low rate could *itself* be an *effect* of contamination (i.e., the model is already so confident on the benchmark that $r_t$ rarely exceeds $1+\epsilon$). This possibility is not discussed.

**Questions:**

see Weaknesses and:

- The explanation for the entropy *increase* in unclipped training (Fig 2 Left) relies on the policy being "skewed" (Remark 4.3 , Fig 5). Can this paper provide any empirical measure of the skewness of the initial Qwen2.5-Math-7B policy to confirm it is in the regime described by Remark B.7?

- The paper implies (via Thm 4.2 and Fig 3 Right) that clipping should be detrimental on harder datasets like AIME because it forces entropy minimization. Did this paper tests the performance of *unclipped* training on the AIME dataset? This seems like a critical experiment to connect the paper's theoretical sections.

---

> ### Author Response · Authors · 2025-11-21
> **Author response to reviewer tUme (part 1)**
>
> Thank you for your positive evaluation and time to provide these detailed and very useful suggestions. We revise the manuscript and conduct more experiments following your points. Below, we address them one-by-one.
>
> 1. **Several key figures presenting experimental results are difficult to interpret due to missing legends and insufficient detail in captions.**
>
> We apologize for the lack of clarity. All lines with different colors in each subpanel indeed correspond to independent runs. For each color, the faint line shows the raw trajectory of an individual run, and the bold line is the smoothed average to make the underlying trend easier to observe despite oscillations. We have added explanations in the revised manuscript.
>
> 2. **The paper's mathematical exposition is dense, and the notation is not always clear or consistent.**
>
> Thank you for pointing this out. We have reorganized Section 2 (Preliminaries) to introduce the full setup and explicitly define all notations before they appear in the later theoretical analysis. We also re-checked all theorems and proofs to ensure consistency throughout.
>
> - **The definitions for the clipping analysis in Theorem 3.3 are imprecise. $\bar{r_{t}}$ itself is not defined...**
>
> We agree. In the revision, we have added Definition 3.1 that formally introduces the notations and related quantities used in later theorems.
>
> - **The proof of Proposition 3.2 appears to rely on unstated assumptions. The expansion...**
>
> We agree and restructured the original Proposition 3.2 into the new Lemma 2.2, where we now explicitly state the required conditions in the lemma statement and provide additional explanation in Remark C.1 to make the underlying assumptions and their role in the expansion fully transparent.
>
> 3. **However, this low rate could *itself* be an *effect* of contamination. This possibility is not discussed.**
>
> Thanks for bringing this out. We first want to clarify that the **existing evidence of Qwen-Math contamination only appears within the validation set**, i.e., MATH500. The training set used in our experiments, DeepScaleR, according to Luo et al. (2025), only consists of selected questions from AMC, AIME, Omni-Math, and Still. According to the Qwen2.5-Math technical report (arXiv 2409.12122), their training data is composed of two parts: (i) CoT data synthesis, which they state includes only GSM8K, MATH, and NuminaMath (page 6, Section 3.1.1), and (ii) a tool-integrated reasoning dataset, which includes GSM8K, MATH, CollegeMath, NuminaMath, MuggleMath, and DotaMath (page 6, Section 3.1.2).
>
> Thus, there is no overlap between our training set and the datasets used to train Qwen2.5-Math. Moreover, the clipping activation ratio for Qwen-Math-7B on our difficult training set of past AIME series also never exceeds 0.002, with a mean around 0.001. Similarly, under $\varepsilon=0.2$, DeepSeek-Distill-Llama-8B's clipping activation rate never exceeds 0.006 (slightly larger, probably due to longer CoT rollout and larger parameter size). We therefore consider it's unlikely that the low clipping rate on the training set is an effect of contamination.
>
> We added a new paragraph in Appendix B.1 over this point to provide further context regarding model contamination to our readers.
>
> 4. **The explanation for the entropy *increase* in unclipped training (Fig 2 Left) relies on the policy being "skewed" (Remark 4.3 , Fig 5). Can this paper provide any empirical measure of the skewness of the initial Qwen2.5-Math-7B policy to confirm it is in the regime described by Remark B.7?**
>
> Insightful point. We conducted additional experiments to empirically evaluate Remark B.7 (now Remark 4.3). To quantify the skewness of the policy, we take the first 500 questions from the DeepScaleR training set and, for each question $x$, we sample 64 rollouts $y$ from the initial Qwen2.5-Math-7B policy to estimate $\Phi(\pi(\cdot \mid x))$. We then visualize the corresponding log-probabilities in Figure~8 (Appendix A.3, page 18) for both cases $\Phi(\pi(\cdot \mid x)) > 0$ and $\Phi(\pi(\cdot \mid x)) < 0$ to provide a clearer sense of skewness under different $\Phi$ values. Recall from Remark B.7 (now Remark 4.3), entropy increase under unclipped training can occur only when $\Phi(\pi(\cdot \mid x)) < 0$, which corresponds to more skewed policies. Among the 500 sampled questions, 358 satisfy $\Phi(\pi(\cdot \mid x)) < 0$.

---

> > ### Author Response · Authors · 2025-11-21
> > **Author response to reviewer tUme (part 2)**
> >
> > 5. **Did this paper tests the performance of *unclipped* training on the AIME dataset? This seems like a critical experiment to connect the paper's theoretical sections.**
> >
> > Great point. We have conducted additional experiments with unclipped training on AIME and added the results of multiple independent runs, together with their entropy evolution, to Figure 7 (Appendix A.3, page 17), along with a detailed discussion. We observe qualitatively similar behavior to the clipped setting, but with different entropy dynamics, as some runs exhibit increasing entropy during training. We also present specific trial in Figure 7 (right), where both performance and entropy increase over training.
> >
> > Taken together with our previous counterexample, where entropy decreases while performance degrades, these results further support our claim that there is no direct causal relationship between policy entropy and policy performance. Again, thank you for raising this important point.
> >
> > ---
> >
> > We again appreciate your time to provide key suggestions to enhance the empirical results. In the global rebuttal box, we summarized all requested changes from reviewers. We hope that our responses have addressed your concerns, and we look forward to an open-minded discussion if any such concerns remain.
> >
> >
> >
> > Sincerely,
> >
> >
> > Authors of Paper 35

---

### Official Review · Reviewer_1ufZ · 2025-10-30

**Soundness:** 3
**Presentation:** 3
**Contribution:** 2
**Rating:** 4
**Confidence:** 2

**Summary:**

This paper investigates the exploration–exploitation dynamics in reinforcement learning with verifiable rewards (RLVR), a method used for training reasoning-focused large language models. The authors analyze the roles of clipping, policy entropy, and spurious (random) rewards, arguing that prior explanations of performance gains under random rewards are incomplete. They propose theoretical analyses and empirical studies suggesting that clipping primarily reduces policy entropy, and that observed gains arise from reward–entropy interactions and model–data regime effects rather than contamination or genuine learning.

**Strengths:**

•	Interesting theoretical framing: The paper offers a formal treatment linking clipping bias and policy entropy, contributing to a better conceptual understanding of RLVR optimization dynamics.
•	Relevance to ongoing discourse: Given recent debates about “spurious reward” effects and entropy minimization in reasoning LLMs, this paper addresses a timely and relevant topic for the ICLR community.
•	Clarity of theoretical results: The analytical sections (Theorems 3.3–4.2) are clearly presented and mathematically rigorous, providing useful insight into when and why entropy changes during GRPO updates.

**Weaknesses:**

•	Experimental design inconsistency: Although the paper claims “extensive experiments across multiple model families and sizes,” not all experiments are conducted uniformly; rather, different subsets of experiments use different models and data. This fragmented setup makes it difficult to assess the generality of the conclusions.
•	Limited empirical depth: The experiments do not convincingly support the claim of “reconciling conflicting reports” in the literature. The evaluation scope remains narrow, and results on random rewards or entropy minimization are mostly qualitative or limited to a few benchmarks (e.g., MATH500, AIME).
•	Unclear novelty of findings: The conclusion that entropy minimization acts as a regularizer, or that clipping primarily affects entropy, is intuitive and has been suggested in prior work. The contribution beyond existing studies remains unclear.
•	Incomplete closure of central questions: While the paper lists three core research questions, the final sections do not directly or conclusively answer them.

**Questions:**

1.	You mention that experiments were run across several model families and sizes, but not all settings overlap. Could you clarify how these choices were made and whether comparable setups were used across models?

---

> ### Author Response · Authors · 2025-11-21
> **Author response to reviewer 1ufZ (part 1)**
>
> We appreciate your time to provide these constructive feedbacks in detail. We follow your suggestions to improve the manuscript and conduct further experiments. Below, we address your questions one-by-one.
>
> 1. **Although the paper claims “extensive experiments across multiple model families and sizes,” not all experiments are conducted uniformly; rather, different subsets of experiments use different models and data. This fragmented setup makes it difficult to assess the generality of the conclusions.** & **You mention that experiments were run across several model families and sizes, but not all settings overlap. Could you clarify how these choices were made and whether comparable setups were used across models?**
>
> We clarify the intent behind our experimental design and introduce additional experiments we have conducted to enhance comparability and generality.
>
> **Clarification regarding experimental design.**
>
> Our experiments are organized around two goals: (i) understanding the interaction between clipping, policy entropy, and model performance under spurious rewards, and (ii) validating whether the benefits of spurious rewards extend beyond Qwen-Math} to a broader class of model families.
>
> For goal (i), we apologize for any confusions and hope to clarify that our experiment design is consistent, which we sticked to Qwen-Math-7B when studying all the dynamics between clipping, entropy, and performance. This choice is guided both by prior empirical work on random rewards and by practical considerations: Qwen-Math-7B has a moderate parameter count and a relatively short context window (4K tokens per response), which makes training more stable and less susceptible to uncontrollable factors such as gradient explosion during unclipped training, which can otherwise obscure the behavior of unclipped training (recall the discussion in Section 4.2).
>
> Moreover, RLVR results often require validation over multiple independent runs. A single Qwen-Math-7B run on the full dataset already requires about 15 hours on 8xH100 GPUs. In contrast, reproducing the full set of experiments with larger models would incur prohibitive computational cost (e.g., a single DeepSeek-R1-Distill-Llama-8B run would take roughly 68 hours on 8xH200, and QwQ-32B would take 32xH200 for training). For this reason, we employ larger models primarily for goal (ii): to demonstrate that the phenomena under spurious rewards are not specific to the potentially contaminated Qwen-Math family. We believe this design strikes a balance between consistency (a fixed, stable Qwen-Math-7B backbone for detailed analysis and ablations) and generality (representative experiments on distinct, stronger model families).
>
> We added a detailed explanation of experiment design rationale and results overview in Appendix A.1 for further elaboration and better clarity.
>
> **Additional experiments to enhance comparability and generality.**
>
> To strengthen cross-model comparison, we have added another DeepSeek-R1-Distill-Llama-8B experiments and additional QwQ-32B trials in Figure 3, using the same harder AIME training setup as for Qwen-Math-7B. These additions provide a more direct, side-by-side comparison across weaker and stronger models under identical data and training conditions, and further support the generality of our conclusions across different model families and capacities.

---

> ### Author Response · Authors · 2025-11-21
> **Author response to reviewer 1ufZ (part 2)**
>
> 2. **The experiments do not convincingly support the claim of “reconciling conflicting reports” in the literature. The evaluation scope remains narrow, and results on random rewards or entropy minimization are mostly qualitative or limited to a few benchmarks (e.g., MATH500, AIME).**
>
> Thanks for raising this point. Our claim is specifically about reconciling the conflicting reports on random-reward RLVR for math reasoning, which arise on MATH500 with Qwen-Math in prior works. Our benchmark and model choices therefore follow these literatures by design, MATH500 is exactly where contamination and clipping-bias explanations were proposed, and AIME is introduced as a harder dataset in the same domain to test whether the same mechanisms still hold and to further study entropy dynamics. To obtain comparable empirical results and isolate the underlying dynamics, we thus adopt the same setup as previous works, where there is uniform evidence of contamination on MATH500.
>
> Regarding the concern that our results are "mostly qualitative", in all settings we present full trajectories over multiple independent runs. To further strengthen the quantitative support for our claims, we have added: (i) a direct quantitative comparison between weaker and stronger model improvement in Figure 4 (Right); (ii) additional quantitative analysis of policy skewness (Figure 8), directly characterizing the skewness behavior predicted by our theory; and (iii) more systematic ablations over the clipping ratio (Figure 5), further analyzing that our claim about clipping and performance.
>
> 3. **The conclusion that entropy minimization acts as a regularizer, or that clipping primarily affects entropy, is intuitive and has been suggested in prior work. The contribution beyond existing studies remains unclear.**
>
> We agree that higher-level ideas such as "entropy minimization can behave like a regularizer" may have appeared in prior work. However, these ideas have mostly been empirical insights, not embedded in a provable and principled framework. Our contribution is precisely to move beyond informal intuition and provide a rigorous analysis that links clipping, entropy, and performance to random rewards, reward misalignment, and model strength, yielding a principled framework supported by both theoretical and empirical results.
>
> Moreover, for entropy section, we emphasize that our contribution is not a generic statement that "entropy affects performance", but a specific, provable characterization of when entropy will increase or decrease. In particular, our analysis identifies concrete conditions, such as the skewness of the policy distribution, under which entropy dynamics behave differently, going beyond high-level intuitions.
>
> To the best of our knowledge, this level of formalization and unification has not been achieved in previous work, and we believe the ongoing debate on RLVR learning dynamics (spurious rewards, entropy control, etc.) will benefit from this more balanced theoretrical-empirical perspective.
>
> 4. **While the paper lists three core research questions, the final sections do not directly or conclusively answer them.**
>
> Thanks for pointing it out. We have revised the conclusion section accordingly.
>
> ---
>
> In the global rebuttal box, we summarized all requested changes raised by all reviewers. We again appreciate your time to provide these insightful feedbacks. We hope that our responses have addressed your concerns, and we look forward to an open-minded discussion if any such concerns remain.
>
>
> Sincerely,
>
>
> Authors of Paper 35

---

> > ### Comment · Reviewer_1ufZ · 2025-11-24
> >
> > Thank you for the clarification and modifying the paper accordingly. I think the revised version will be improved and therefore increase my score to 6 to reflect this.

---

> > > ### Author Response · Authors · 2025-11-24
> > > **Thank you for your time and effort**
> > >
> > > Dear reviewer 1ufz,
> > >
> > > Again, we sincerely appreciate for your time and effort in reviewing the manuscript and suggestions to guide us to improve the manuscript.
> > >
> > > In the end, thank you for your support and valuable input towards our work.
> > >
> > > ---
> > >
> > > Sincerely,
> > >
> > > Authors of Paper 35

---

### Official Review · Reviewer_PiCo · 2025-11-01

**Soundness:** 3
**Presentation:** 3
**Contribution:** 3
**Rating:** 6
**Confidence:** 3

**Summary:**

This paper investigates how policy entropy influences reinforcement learning with verifiable rewards and whether spurious rewards can yield gains through clipping bias and model contamination. Through rigorous mathematical analysis supported by empirical experiments, the authors establish a direct connection between clipping and entropy, showing that clipping primarily reduces policy entropy. They argue that clipping does not provide a genuine learning signal and discuss when reduced entropy is beneficial. The authors introduce a reward-misalignment model to explain under which conditions spurious rewards enhance performance.

**Strengths:**

- The paper tackles an important and timely question in reinforcement learning for large language models.
- Rigorous theoretical analysis linking clipping and entropy, extending prior accounts.
- The reward-misalignment model offers a probabilistic explanation for the benefits of random rewards.
- The paper is well motivated, interesting, and clearly presented.

**Weaknesses:**

- The evaluations are concentrated on MATH500, and ablations on hyperparameters (e.g., clipping ratio, group size) are missing.
- Some findings, while formalized, may only confirm intuitively expected behaviors (e.g., entropy minimization failing when incorrect trajectories are the peak of the distribution).

**Questions:**

- Why did you not stick to the same model for the experiments or use them all?
- How do empirical results change with different clipping thresholds?

---

> ### Author Response · Authors · 2025-11-21
> **Author response to reviewer PiCo (part 1)**
>
> Thank you for your positive assessment and valuable input. We follow your suggestions and conduct further experiments.  We answer your questions below one-by-one.
>
> 1. **The evaluations are concentrated on MATH500, and ablations on hyperparameters (e.g., clipping ratio, group size) are missing.**
>
> Thanks raising this point. In the revised manuscript, we include additional ablations over the clipping ratio and the group size. For the clipping ratio $\varepsilon$, the original draft presented results for $\varepsilon = 0.2$ and the unclipped case (i.e., $\varepsilon = \infty$), which demonstrates the effect of clipping becomes looser. To further demonstrate the robustness of our conclusions under stricter clipping, we conducted additional experiments with $\varepsilon \in \\\{0.15, 0.1\\\}$, each with six independent trials. The detailed results and discussion are provided in Figure 5 (page 16). In summary, we find that the qualitative behavior is consistent: the final performance is largely unaffected by the specific choice of clipping ratio within this range.
>
> For the group size $G$, from the reward-misalignment view, larger groups ($G = 16$) yield more balanced random-reward assignments and reduce the chance of extreme cases (e.g., assigning 0 to all correct rollouts in a group). Conversely, smaller groups increase the probability of such extreme assignments. To empirically validate this, we repeated the experiments under the same setup with a reduced group size of $G = 8$, and present the results in Figure 6 (page 17). In short, while performance can still improve, the training becomes noticeably less stable, with larger variance across trials and inducing instability.
>
>
> Regarding the evaluation benchmark, our work aims to clarify the open questions raised from the previous RLVR empirical studies. Recent studies provide uniform evidence of Qwen-Math contamination on MATH500, which directly motivated the contamination–clipping-bias hypothesis as an explanation for its improvements under random rewards. To align with their setups, we focus on MATH500 to ensure comparable empirical setting.
>
>
> 2. **Some findings, while formalized, may only confirm intuitively expected behaviors (e.g., entropy minimization failing when incorrect trajectories are the peak of the distribution).**
>
> We agree that some expected behaviors sound intuitive, but we want to stress that most current RLVR works discuss these insights in a largely empirical way, leading to many different and sometimes conflicting viewpoints on these open questions (e.g., as discussed in preliminary section, after Shao et al. (2025) proposed that spurious rewards can improve Qwen-Math, Oertell et al. (2025) soon argued that spurious heuristics may be harmful). These discrepancies arise precisely because there is no principled framework that connects theory to empirical performance. Our contribution is to replace informal intuition with a **provable and principled** framework that links clipping, entropy, and performance, supported by extensive empirical experiments.

---

> > ### Author Response · Authors · 2025-11-21
> > **Author response to reviewer PiCo (part 2)**
> >
> > 3. **Why did you not stick to the same model for the experiments or use them all?**
> >
> > In Appendix A.1, we add a detailed overview of our experimental design rationale, and we also add more experiments to ensure consistency.
> >
> > At higher level, our experiments are organized around two goals: (i) understanding the interaction between clipping, policy entropy, and model performance under spurious rewards, and (ii) validating whether the benefits of spurious rewards extend beyond Qwen-Math to a broader class of model families.
> >
> > For goal (i), we fix the model to Qwen-Math-7B to ensure a consistent setup when studying the interaction between clipping, entropy, and performance. This choice is guided both by prior empirical work on random rewards and by practical considerations: Qwen-Math-7B has a moderate parameter count and a 4K-token context window, which makes training more stable and less susceptible to uncontrollable factors such as gradient explosion, which can otherwise obscure the behavior of unclipped training (recall the discussion in Section 4.2).
> >
> > Moreover, given the necessity of multiple independent runs for empirical verification, running a single Qwen-Math-7B trial on the full dataset already requires 15 hours on 8xH100 GPUs. In contrast, reproducing the full battery of experiments with stronger models would induce prohibitive computational cost (e.g., a single DeepSeek-R1-Distill-Llama-8B run would take 68 hours on 8xH200, and QwQ-32B takes 32xH200 for training). For this reason, we use larger models only for goal (ii), demonstrating that the phenomena under spurious rewards are not specific to the potentially contaminated Qwen-Math model. We believe this design strikes a balance between consistency (via a fixed, stable Qwen-Math-7B backbone for detailed analysis) and generality (via representative experiments on distinct, stronger model families).
> >
> > Still, to strengthen the comparison experiment across models of different strengths, we have added DeepSeek-R1-Distill-Llama-8B experiments and additional QwQ-32B trials in Figure 3, enabling a more direct comparison with Qwen-Math-7B on the harder AIME training dataset.
> >
> > ---
> >
> > In the global rebuttal box, we summarized requested changes raised by all reviewers. We again appreciate your time to provide these insightful feedbacks, guiding us to conduct further experiments. We hope that our responses have addressed your concerns, and we look forward to an open-minded discussion if any such concerns remain.
> >
> >
> > Sincerely,
> >
> > Authors of Paper 35

---

### Official Review · Reviewer_9GLK · 2025-11-02

**Soundness:** 3
**Presentation:** 2
**Contribution:** 3
**Rating:** 6
**Confidence:** 2

**Summary:**

The authors investigate seemingly paradoxical phenomena with RLVR: spurious rewards, which should decrease exploitation, and entropy minimization, which should increase exploitation, seem to both yield improvements for LLM's trained on MATH 500. The authors present a series of theoretical and empirical analyses to explain this phenomena.

First, following previous hypotheses that clipping induces a bias towards high probability tokens under the initial model, they derive a bound on the magnitude of the effect of clipping on the gradient and perform empirical experiments to show that improvement still occurs under spurious rewards without clipping.

Second, they analyze the effect of clipping on policy entropy and present empirical results that Qwen2.5-Math-7B undergoes entropy increase with spurious rewards without clipping and entropy decrease with clipping on MATH 500, while both improve in validation performance. They discuss how entropy minimization may not always be useful, depending on the performance of the initial model, and perform empirical experiments with models of different sizes.

Third, they analyze when spurious rewards can be useful in terms of the success rate of the true correctness rate of the initial policy. The basic intuition is that stronger base models will tend to have correct responses reinforced by spurious rewards, and they show experiments with a fine-tuned Llama3 model to show that improvement under spurious rewards becomes more likely as the model gets stronger.

**Strengths:**

The authors carefully investigate open questions related to RLVR and present interesting insights. The combination of empirical and theoretical analysis is compelling.

**Weaknesses:**

1. The presentation of the motivation for each analysis and key takeaways could be more clear. A few general suggestion would be to explicitly state the goal of each analysis at the beginning of each section, to lead with the empirical results that the theory aims to explain, and to use more descriptive figure captions that discuss the key conclusion from each figure. At a few points reading through the paper, it was not clear to me why an analysis was being conducted and what conclusion the authors were justifying with the theoretical result.

2. Notation is not always clear (e.g. r is used to represent importance ratio before defining in section 3)

3. Overall, could do a better job of guiding the reader through the analysis as a story. Current flow seems to be 1) clipping does not explain learning from spurious rewards 2) clipping leads to entropy reduction and acts as a regularizer 3) entropy decrease from clipping does not always lead to improvements 4) models that have high success rates before RL are more likely to improve with spurious rewards. Motivating each of these points and connecting them back to the initial questions would be helpful

**Questions:**

Section 3 questions / comments:
* what is the main takeaway from the upper bound on the clipping bias? The implication seems to be that the clip correction having a small magnitude makes it insignificant, but the later results show that it has meaningful effects (e.g. on entropy). The empirical results are much more clear to me (clipping not required for learning under spurious rewards)
* r as the importance ratio is used in remark 3.1 without defining it (r up to this point had been reward?)
* would be helpful to state why the law of clipping is important
* would be helpful to state the key conclusion from the figure in the caption

Section 4 questions / comments
* it would be helpful to lead with the results in figure 2 to motivate the theoretical analysis (these are actually mentioned in the theory section motivation before they are presented, which is a bit confusing)
* it would be helpful to discuss the implications of the derived expression for entropy change more explicitly (e.g. after theorem 4.2, explain how this indicates that training with clipping leads to entropy decrease). The notation is very dense and proof is in the appendix, so it is not obvious how to interpret this analysis as one reads through. Currently the paragraph after the proof refers to the empirical results, rather than the implications of the theorem
* Remark 4.3 does a better job of discussing implications, but again it refers to the empirical results that haven't been introduced
* section 4.3 felt a bit out of place given that the whole section up to this point had been about the relationship between clipping and entropy, not entropy and performance. Useful analysis, but it would be helpful to set it up a bit more for the reader (e.g. mention this question in the intro paragraph for this section)

Section 5
* Figure could be made more clear. If the key point is that initial model performance explains how likely a model is to improve with spurious rewards, a plot comparing initial model performance to improvement would be helpful

---

> ### Author Response · Authors · 2025-11-21
> **Author response to reviewer 9GLK (part 1)**
>
> We appreciate your encouraging remarks and constructive suggestions, which is very helpful for us to improve the manuscript. We follow your suggestions and address each points below:
>
> 1. **The presentation of the motivation for each analysis and key takeaways could be more clear.** & 3. **Overall, could do a better job of guiding the reader through the analysis as a story.**
>
>
> We highly appreciate these helpful, section-specific suggestions. In the revised version, we have revised Section 2, 3, 4, and 5, and added guiding text to more clearly connect the theoretical and empirical results. At the beginning of each section, we include a short, reader-friendly overview that states the goal of the section, and how it relates to the central questions of the paper. We have also revise the conclusion section to more explicitly summarize the key messages and how they answer the open questions proposed Section 2.
>
>
> 2. **Notation is not always clear. (e.g. r is used to represent importance ratio before defining in section 3)**
>
> We agree and have revised the notation to improve clarity. In particular, we have restructured Section 2 to introduce the full setup and explicitly define all notations before they appear in the later theoretical analysis. All the theorems and proofs are also re-checked to ensure the consistency.
>
>
> ---
>
> *Section-specific questions - section 3*
>
> 4. **What is the main takeaway from the upper bound on the clipping bias?**
>
> We added Remark 3.3 after Theorem 3.2 to further explain this. The main purpose of the upper bound is to provide a model-agnostic analysis on clipping effects: bounding the clipping bias in terms of different hyperparameters along with the empirical token-level clipping activation rate $p$ (varies across model families), which can be traced during training. For takeaway, larger $p$ directly implies a larger upper bound on the bias, and the bound grows with the rate of $\mathcal{O}(\eta\sqrt{p})$ ($\eta$ as learning rate). We added a sentence at the end of Theorem 3.2 to make this implication explicit.
>
> 5. **Would be helpful to state why the law of clipping is important**
>
> We agree. Recall that Corollary 3.6 elaborates the law of clipping by plugging in the actual hyperparameters used in training, yielding a realistic estimate of how much clipping bias can affect the gradient.
>
> We revised a new paragraph (line 318-323) below Corollary 3.6 to highlight the key conclusion and its significance: clipping bias itself does not provide a meaningful learning signal. Within this paragraph, we also add a natural bridge to Section 4, where we analyze the actual influence of clipping through its effect on policy entropy.
>
> 6. **Would be helpful to state the key conclusion from the figure in the caption**
>
> We provided the explanation to Figure 1 in the paragraph immediately below it (lines 306–309).

---

> ### Author Response · Authors · 2025-11-21
> **Author response to reviewer 9GLK (part 2)**
>
> *Section-specific questions - section 4*
>
> 7. **It would be helpful to lead with the results in figure 2 to motivate the theoretical analysis (these are actually mentioned in the theory section motivation before they are presented, which is a bit confusing).** & 8. **it would be helpful to discuss the implications of the derived expression for entropy change more explicitly (e.g. after theorem 4.2, explain how this indicates that training with clipping leads to entropy decrease).**
>
> We agree. In the revised version, we added a new Remark 4.2 that first introduces the empirical results from Figure 2 and then uses them to interpret the mentioned theorem. This makes the theory easier to follow and helps readers better understand the meaning and relevance of the formal result.
>
> (we have slightly reordered Section 4 for better readability: the original Definition 4.1 (policy entropy) has been moved to the preliminaries so that readers are introduced to the definition of entropy at beginning, and the former Theorem 4.2 is now renumbered as Theorem 4.1)
>
>
> 9. **Remark 4.3 does a better job of discussing implications, but again it refers to the empirical results that haven't been introduced.**
>
> We agree. We restructured the original Remark 4.3 by splitting it into a technical statement (now Theorem 4.3) and a follow-up explanation with a numerical simulation example (now Remark 4.4), which uses a simple two-armed policy to illustrate Theorem 4.3.
>
> We also added an additional experiment to evaluate Qwen-Math-7B's $\Phi(\pi)$, so that readers can see the actual skewness of the model and better relate it to Theorem 4.3. This new experiment is described in the paragraph immediately following Remark 4.4, and the resuls are presented in Figure 8.
>
> 10. **section 4.3 felt a bit out of place given that the whole section up to this point had been about the relationship between clipping and entropy, not entropy and performance. Useful analysis, but it would be helpful to set it up a bit more for the reader (e.g. mention this question in the intro paragraph for this section).**
>
> Thanks for the suggestion. We added an intro paragraph at the beginning of Section 4 to setup these results. We also revised Section 4.3 to make it more smoothly flow from the previous part.
>
> ---
>
> *Section-specific questions - section 5*
>
> 11. **Figure could be made more clear. If the key point is that initial model performance explains how likely a model is to improve with spurious rewards, a plot comparing initial model performance to improvement would be helpful.**
>
> Great point. In Figure 4 (page 10), we present the average percentage performance improvement across multiple independent runs for models with different base strengths (i.e., initial model performance). We additionally conducted experiments for Qwen-Math-1.5B model under exactly the same training setup, which makes the contrast between weaker and stronger base models more explicit.
>
> ---
>
> In the global rebuttal box, we summarized all requested changes (both to the paper content and additional experiments) raised by all reviewers. We again appreciate your close reading and valuable feedback, which we believe has significantly improved the readability and flow of the revised manuscript.  We hope that our responses have addressed your concerns, and we look forward to an open-minded discussion if any such concerns remain.
>
>
>
> Sincerely,
>
>
> Authors of Paper 35

---

### Author Response · Authors · 2025-11-21
**Summary of requested changes**

Again, we thank reviewers for your time to provide constructive suggestions. We provide a summary of requested changes from all reviewers across paper sections and additional experiments. Non-trivial changes are reflected in green highlights.

---

*Paper*

- Section 2: we reorganize the preliminary section by (i) clearly define and introduce all the notations and setup and (ii) moving important lemmas and definitions that used in the later analysis to this section, for better clarity.

- Section 3: we add Definition 3.1 for a clearer definition and Remark 3.3 for better explanation of theorem to the readers.


- Section 4: we add Remark 4.2 and Remark 4.4 to better explain their correspdoning theorems.


- Appendix A.1: a systematic review of experiment results and rationale.


- Appendix A.2: discussions over additional ablation analysis.


- Appendix A.3: discussions over additional unclipped experiment and model skewness.

---

*Experiments*

- Page 16-17, Figure 5-6: ablation on clipping ratio threshold $\varepsilon$ and GRPO group size $G$.

- Page 17, Figure 7: additional experiment on Qwen2.5-Math-7B with unclipped training over AIME set.

- Page 8, Figure 3: additional experiments of DeepSeek-R1-Distill-Llama-8B over AIME set & more trials over QwQ-32B.

- Page 18, Figure 8, further demonstration of policy skewness.

- Page 10, Figure 4: additional experiment and figure presentation on the base models' initial strength and their corresponding final average improvement.

---


Sincerely,


Authors of Paper 35

---

### Author Response · Authors · 2025-12-01
**Summary of Rebuttal & Manuscript Improvement to Area Chair**

Dear Area Chair,

Following the recent ICLR security incident and the resulting reassignment of papers, we first want to sincerely thank you for the additional time and effort you are investing in handling our submission.

To help make your assessment easier, we provide a concise summary of the reviews, our rebuttal, and the main changes to the paper.

---

***Post-rebuttal overview***

- Current reviewer scores: 6 (9GLK), 6 (PiCo), 6 (tUmE), and 4$\rightarrow$6 (1ufZ, updated on November 24 after reading the rebuttal, though the system restored it back to 4).
- Reviewers consistently describe the work as timely for RL for LLMs, theoretically rigorous, and clearly presented.
- The main concerns focus on:
  1. Sharpening the key takeaways and novelty.
  2. Improving clarity of notation, mathematical expressions, figures, and story flow.
  3. Strengthening and stress-testing the empirical evaluation (ablations, consistency, and alternative explanations).
- We addressed these via substantial clarifications and new experiments: restructured preliminaries and notation, added theorem remarks and clearer conclusions, included additional ablations, and ran new experiments (DeepSeek-R1-Distill-Llama-8B, QwQ-32B, policy skewness, and unclipped AIME training).

---

**Reviewer 9GLK (Score: 6)**

Reviewer 9GLK appreciates our careful investigation of open questions in RLVR, the interesting insights, and the combination of theoretical and empirical analysis. Their main suggestions concern the clarity of (i) key takeaways, (ii) notation, and (iii) story flow.

- For (i) key takeaways, we added additional remarks to the main theorem and expanded the discussion to better highlight its significance and to more clearly connect the theoretical and empirical results.
- For (ii) notation, we restructured Section 2 (Preliminaries) so that the setup and all notation are introduced clearly before they appear later in the analysis.
- For (iii) story flow, we followed the reviewer’s section-specific questions, revising and reordering parts of the manuscript to make the narrative more coherent.

---

**Reviewer PiCo (Score: 6)**

Reviewer PiCo finds the topic **important and timely** for RL with LLMs, and characterizes the paper as **well-motivated, interesting, and clearly presented**, with **rigorous theoretical analysis**. Their main concerns are: (i) missing ablations, (ii) some findings appearing intuitive, and (iii) clarity of the experimental setup.

- For (i), we followed the reviewer’s suggestions, adding the requested ablations and including them with discussion in Appendix A.2.
- For (ii), we clarified that the core contribution is not that individual empirical observations are surprising, but that we provide a **provable and principled** framework that links the elements of RLVR, helping to reconcile recent controversial empirical findings in the community.
- For (iii), we expanded the description of the experimental rationale and clarified the consistency of the setup, and we incorporated this explanation directly into the manuscript.

---

**Reviewer 1ufZ (Score: 4$\longrightarrow$6 after rebuttal, now restored to 4)**

Reviewer 1ufZ commends our theoretical framing as **interesting**, **clearly presented**, and **mathematically rigorous**, and notes that it addresses a **timely and relevant topic** for the ICLR community. Their concerns focus on (i) experimental consistency, (ii) limitations of the empirical evaluation, and (iii) clarity regarding novelty.

- For (i), we clarified the experimental goals and rationale (using Qwen-Math-7B to examine detailed training dynamics and larger models for generality), and we added additional runs with DeepSeek-R1-Distill-Llama-8B and QwQ-32B under a shared setup.
- For (ii), we emphasized that our empirical focus is on reconciling conflicting results on random-reward RLVR for math reasoning and strengthened this section with additional quantitative analyses, including cross-model comparisons, policy skewness, and clipping ablations.
- For (iii), we clarified that our main contribution is a **provable and principled** framework linking clipping, entropy, and spurious rewards, providing precise conditions for entropy changes. We revised the conclusion to explicitly answer the three core research questions and make the novelty more transparent.

---

> ### Author Response · Authors · 2025-12-01
> **Summary to Area Chair -- Continued**
>
> **Reviewer tUmE (Score: 6)**
>
> Reviewer tUmE acknowledges the technical strength of the work. Their concerns focus on (i) clarity of figures and captions, (ii) clarity of mathematical expressions and assumptions, (iii) alternative contamination explanations, and (iv) additional experiments on skewness and unclipped AIME training.
>
> - For (i), we clarified that different colors represent independent runs, explained the meaning of faint vs. bold lines, and revised all relevant figure captions accordingly.
> - For (ii), we reorganized the preliminaries, explicitly defining all notation (new Definition 3.1), and restated the earlier proposition as Lemma 2.2 with clearly stated assumptions and an explanatory remark.
> - For (iii), we clarified that contamination is unlikely based on non-overlapping training data and consistently low clipping activation rates across datasets and models, now discussed in Appendix B.1.
> - For (iv), we added new experiments measuring policy skewness (Fig. 8) and testing unclipped training on AIME (Fig. 7), directly addressing the reviewer’s requests and further supporting our theoretical claims about the relationship between entropy and performance.
>
> ---
>
> In closing, we again thank all four reviewers for their efforts and for the constructive suggestions that helped us strengthen the paper. We are also very grateful to you for the additional time and care involved in assessing our submission and coordinating the discussion under the current circumstances. We hope that this summary of the rebuttal and the resulting changes helps to reduce your workload.
>
>
>
> Sincerely,
>
>
>
> Authors of Paper 35

---

### Meta-Review · Area_Chair_gk62 · 2026-01-06

**Summary:**

Most of the reviewers’ concerns with the initial submission fell into three categories: questions about the thoroughness and consistency of the experimental settings, writing/clarity concerns, and questions about the level of insight given by the overall analysis. Having read the discussion and looked at the revised paper, I am happy that the first two groups of concerns have been addressed; while of course it would be “better” to run every possible combination of experiment on every model, for a reasonable computational budget I think the set of experiments in the current version are well justified and more than reasonable. The remaining group of concern is about whether the results are sufficiently surprising or insightful to be interesting: I agree that some of the insights are unsurprising, but think that the current version of the paper gives sufficiently valuable on a highly relevant and timely problem to be more than worth sharing with the ICLR community.

**Reviewer Concerns:**

Discussed above.

**Reviewer Scores:**

One reviewer did explicitly upgrade from a 4 to a 6 in a comment that has been maintained below; for the others, I would think the scores would either have remained 6 or increased slightly given a full discussion period.

---

### Decision · Program_Chairs · 2026-01-26

Accept (Poster)